# ShuffleDetect: Detecting Adversarial Images against Convolutional Neural Networks

Raluca Chitic [1,*] , Ali Osman Topal [2,*] and Franck Leprévost [2]

1 Robert Bosch GmbH, 013937 Bucharest, Romania
2 Faculty of Science, Technology and Medicine, University of Luxembourg,
L-4364 Esch-sur-Alzette, Luxembourg; franck.leprevost@uni.lu
* Correspondence: ioana-raluca.chitic@ro.bosch.com (R.C.); aliosman.topal@uni.lu (A.O.T.);
Tel.: +352-661-555-436 (A.O.T.)

**Abstract:** Recently, convolutional neural networks (CNNs) have become the main drivers in many image recognition applications. However, they are vulnerable to adversarial attacks, which can lead to disastrous consequences. This paper introduces ShuffleDetect as a new and efficient unsupervised method for the detection of adversarial images against trained convolutional neural networks. Its main feature is to split an input image into non-overlapping patches, then swap the patches according to permutations, and count the number of permutations for which the CNN classifies the unshuffled input image and the shuffled image into different categories. The image is declared adversarial if and only if the proportion of such permutations exceeds a certain threshold value. A series of 8 targeted or untargeted attacks was applied on 10 diverse and state-of-the-art ImageNet-trained CNNs, leading to 9500 relevant clean and adversarial images. We assessed the performance of ShuffleDetect intrinsically and compared it with another detector. Experiments show that ShuffleDetect is an easy-to-implement, very fast, and near memory-free detector that achieves high detection rates and low false positive rates.

**Keywords:** adversarial attacks; detection; evolutionary algorithms; convolutional neural networks; security

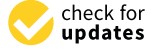



## 1. Introduction

Convolutional neural networks (CNNs) trained on large sets of examples are dominant tools for object recognition [1]. Although CNNs are capable of accurately classifying new images into object categories, they can nevertheless be deceived by adversarial attacks [2], whose strategies generally consist of altering inputs with perturbations that lead to classification errors.

These attacks can be classified in terms of the amount of information that the attackers have at their disposal. Gradient-based attacks (e.g., [3–6]) require information about the CNN's architecture and weights. Transfer-based attacks (e.g., [7–9]) require less insider knowledge about the CNN but query the CNN for a set of inputs, and the collected information is used to create a substitute model, similar to the targeted CNN. This substitute model is attacked by gradient-based methods, leading to adversarial images that also fool the target CNN. Score-based attacks (see [10]) are even less demanding. They do not have access to the training data, model architecture, or CNN parameters. They only make use of the CNN's predicted output probabilities for all or a subset of object classes.

Ideally, security issues posed by adversarial attacks are prevented by methods that detect malicious input images, potentially exclude them from further processing by the CNN, and alert the user. Such detectors may be tailor-made for a specific type of attack or applied efficiently to a large variety of attacks. Their performances are encompassed by a series of indicators that assess how far their outputs can be trusted, and the memory overhead, time, or complexity required to finish their tasks.



These detectors can be classified as supervised and unsupervised. On the one hand, supervised techniques have knowledge of adversarial images, and may attempt to reinforce CNNs by adding adversarial images to the training set [4]. These techniques are particularly effective when attacks are known in advance. On the other hand, unsupervised techniques [11–14] operate without prior access to adversarial images. Instead, they apply transformations to the input image and analyze the consistency of predictions between the input image and its transformed versions. These techniques operate on the premise that CNNs maintain consistent predictions for clean images.

This paper introduces ShuffleDetect as a new unsupervised method for the detection of adversarial images; it is simple to implement and works efficiently against adversarial images created by a series of 8 different attacks applied to 10 different ImageNet-trained CNNs.

To summarize, given image $\mathcal{I}$, ShuffleDetect assesses whether $\mathcal{I}$ (resized to a square $N \times N$ image, if necessary, to fit the CNN's input size) is adversarial or not for a given CNN $\mathcal{C}$. Firstly, the algorithm extracts the dominating category $\text{Dom}_{\mathcal{C}}(\mathcal{I})$ in which $\mathcal{C}$ classifies $\mathcal{I}$. Secondly, the algorithm essentially "splits" $\mathcal{I}$ into non-overlapping patches of an equal size $s \times s$. Thirdly, for each permutation $\sigma$ of a set of $t$ permutations of these patches, the algorithm creates a shuffled image $sh(\mathcal{I}, \sigma)$, and requires from $\mathcal{C}$ the dominating category $\text{Dom}_{\mathcal{C}}(sh(\mathcal{I}, \sigma))$ in which $\mathcal{C}$ classifies the shuffled image. Lastly, the algorithm compares the outcome with $\text{Dom}_{\mathcal{C}}(\mathcal{I})$. The detector classifies an input image $\mathcal{I}$ as "adversarial" if the proportion of permutations $\sigma$ among $t$ permutations is such that the dominant categories of $\mathcal{I}$ and $sh_{\sigma}(I, s)$ differ by more than a certain threshold value $R_{th}$.

The remainder of this paper is organized as follows. Section 2 provides an overview of how CNNs perform image classification, defines the attack scenarios and adversarial image requisites, and fixes some concepts and notations used throughout the article. Section 3 is devoted to related works, provides the topography of detection methods, and lists the main evaluation criteria used to assess their performances. The design of the ShuffleDetect method is detailed in Section 4, where the pseudo-code of the ShuffleDetect$^{\mathcal{C}, R_{th}, t}$ algorithm is also given explicitly.

To evaluate the reliability of our ShuffleDetect method, we tested it against a large set of adversarial attacks deceiving a significant series of CNNs. Section 5 lists the 10 selected CNNs trained on ImageNet, as well as the reasons for their choices, the 100 clean ancestor images, and the specific scenarios used in our experiments. Section 6 lists the 8 attacks that are considered in this paper, seven of which are "white-box", while one is "black-box". Whenever applicable, we performed both the targeted and untargeted versions of the attacks. A total of 15,000 attack runs led to 9580 relevant adversarial images: 2975 adversarial images for the targeted scenario and 6505 adversarial images for the untargeted scenario, as described in Section 7.

Section 8 specifies the parameters used by our detector for images handled by CNNs trained on ImageNet. This section essentially amounts to measuring the outcomes of ShuffleDetect$^{\mathcal{C}}_{\sigma}$ individually for each permutation $\sigma$, each CNN $\mathcal{C}$, each clean image, and each image adversarial for $\mathcal{C}$, obtained by each attack for each scenario. The results lead to the selection of candidates for the threshold value $R_{th}$. The performance of the detector ShuffleDetect$^{\mathcal{C}, R_{th}, t}$ is then assessed in Section 9 against the indicators given in Section 3 for the candidate values of $R_{th}$. Beyond this intrinsic performance assessment, ShuffleDetect is compared with the well-known detector Feature Squeezing in Section 10.

Section 11 summarizes our findings, specifies our recommendations for the values of the parameters relevant to ShuffleDetect, and indicates some directions for future work.

Additional figures, tables, and relevant data are provided in the Appendix, including the original clean images, the permutations used, and individual performances of ShuffleDetect per CNN per attack per scenario.

Algorithms and experiments were implemented using Python 3.8 [15] with NumPy 1.19 [16] and PyTorch 1.9 [17] (including in particular the Adversarial Robustness Toolbox Python library used in Section 6). In addition, we used Maple 2022 to create the permutations used in Sections 8 and 9. The main computations were performed on nodes using

Nvidia Tesla V100 GPUs, which are part of the IRIS HPC Cluster at the University of Luxembourg [18].

## 2. CNNs and Adversarial Images

A CNN, which is expected to perform image classification, is first trained on a large dataset $\mathcal{S}$ of images. Training consists of sorting the given images into a finite set of predefined categories. The categories $c_1, \ldots, c_\ell$, their number $\ell$, and the images used in the process are associated with $\mathcal{S}$, and are common to any CNN trained on $\mathcal{S}$. The training phase of a CNN consists of two phases. Firstly, the CNN is given the training images, and, for each training image, a vector of length $\ell$, where each real-value component assesses the probability that the training image represents an object in the corresponding category. Secondly, CNN is challenged against a validation set of images that assesses its ability to sort images accurately. Once trained, a CNN can be exposed to an arbitrary image, and perform its classification according to $\ell$ categories.

An important, albeit technical, issue involves the sizes of the images. While the sizes of the images of $\mathcal{S}$ are arbitrary and may vary from one image to another, a CNN handles images of a fixed input size. Therefore, a resizing process is usually necessary to adapt a given image to the input size of the CNN before classification. To simplify the notation, we consider that this resizing process has been performed, and the input size handled by the CNN is square (Section 5 specifies which resizing function is used in the experiments). We also often identify image $\mathcal{I}$ with its resized version, which fits the input size of the CNN.

**Image classification and label values.** Concretely, given an input image $\mathcal{I}$, the trained CNN produces a classification output vector

$$\mathbf{o}_\mathcal{I} = (\mathbf{o}_\mathcal{I}[1], \ldots, \mathbf{o}_\mathcal{I}[\ell]),$$

where $0 \le \mathbf{o}_\mathcal{I}[i] \le 1$ for $1 \le i \le \ell$, and $\sum_{i=1}^{\ell} \mathbf{o}_\mathcal{I}[i] = 1$. Each component $\mathbf{o}_\mathcal{I}[i]$ defines the $c_i$-label value measuring the probability that image $\mathcal{I}$ belongs to the category $c_i$. Consequently, the CNN classifies image $\mathcal{I}$ as belonging to the category $c_k$ if $k = \arg\max_{1 \le i \le \ell}(\mathbf{o}_\mathcal{I}[i])$. One denotes $(c_k, \mathbf{o}_\mathcal{I}[k])$ this outcome, and $\text{Dom}_\mathcal{C}(\mathcal{I}) = c_k$ the dominating category in which $\mathcal{C}$ classifies $\mathcal{I}$. The higher the label value $\mathbf{o}_\mathcal{I}[k]$, the higher the confidence that $\mathcal{I}$ represents an object of the category $c_k$.

**Adversarial image requisites.** Assume that we are given $\mathcal{C}$ a trained CNN, $c_a$ a category among the $\ell$ possible categories, and $\mathcal{A}$ an image classified by $\mathcal{C}$ as belonging to $c_a$, with $\tau_a$ its $c_a$-label value.

For any attack scenario that we consider in this paper (namely the *target* or the *untargeted scenario*, as made precise below), we assume that the attack aims at creating a new *adversarial image* $\mathcal{D}(\mathcal{A})$, which remains so close to the ancestor's *clean image* $\mathcal{A}$ that a human would not be able to distinguish between $\mathcal{D}(\mathcal{A})$ and $\mathcal{A}$. The quantity $\epsilon(\mathcal{A}, \mathcal{D}(\mathcal{A}))$, which controls (or restricts) the global maximum amplitude allowed for the value modification of each individual pixel of $\mathcal{A}$ to obtain $\mathcal{D}(\mathcal{A})$, numerically assesses this human perception.

In the *untargeted scenario*, $\mathcal{C}$ is only required to classify the adversarial image $\mathcal{D}(\mathcal{A})$ as any class $c \ne c_a$. In the *target scenario*, one selects, *a priori*, a target category $c_t \ne c_a$. One would expect the adversarial image $\mathcal{D}(\mathcal{A})$ to be classified by $\mathcal{C}$ as belonging to the target category $c_t$, without any requirements on the $c_t$-label value beyond it being strictly dominant among all label values (this coincides with the concept of a *good enough* adversarial image introduced in [19]; see [19] for variants of the *target scenario* involving $\tau$-*strong adversarial* images).

Throughout the remainder of this article, any attack leading to the creation of adversarial images will be referred to as *atk*.

## 3. Related Works and Evaluation Criteria

As pointed out in the Introduction (Section 1), addressing the security issues posed by adversarial attacks often requires some warning that an attack is indeed taking place. The role of detectors is key in this process because their principal role is to decide whether an

image is clean or not. Such detectors can be categorized into two groups: supervised and unsupervised detectors (see [20]).

Supervised detectors are designed and trained with images known to be adversarial and obtained from one or more attacks. In contrast, unsupervised detectors require no prior access to adversarial images and are, therefore, not limited to any particular type of attack. This suggests that unsupervised methods, which are more resource-efficient because they do not require any training for new attacks, may be more robust against new adversarial attacks than supervised attacks.

Numerous detection methods from both categories have been introduced (some of which aim at detecting adversarial images for ImageNet-trained CNNs). One can mention the following four detection methods referred to in [20]: the supervised LID [21], the unsupervised NIC [22], ANR [14], and FS [13].

The supervised **Local intrinsic dimensionality (LID)** method extracts intermediate layer activations from the CNN when fed with either clean or adversarial inputs. At each layer, the activations stemming from the image (clean or adversarial) and the activations stemming from a limited number of clean neighbors of the image are used to compute the local intrinsic dimensionality. The authors of [21] found that adversarial images tend to have higher local intrinsic dimensionality values. This property is exploited using the extracted values as features to train a binary classifier that declares an image as clean or adversarial.

The **network invariant approach (NIC)** is an unsupervised method that declares an image to be adversarial if it is out-of-distribution, and clean if it is in distribution. This notion refers to the distribution observed for the ImageNet training set, which consists of only clean images, for each CNN layer activation. For a given image, one obtains a collection of layer-level declarations, indicating whether the image is in distribution or not for that particular layer. The detector's final declaration is an aggregation of all the layer-level declarations.

The **adaptive noise reduction (ANR)** algorithm is an unsupervised method that uses scalar quantization and smoothing spatial image filters to squeeze input images. The detector compares the categories predicted by the CNN for an image and for its squeezed version. If these categories are not identical, the image is considered to be adversarial.

The **feature squeezing (FS)** algorithm is an unsupervised method that applies depth reduction to an image color bit, a median image filter for local smoothing, and a variant of the Gaussian kernel for non-local spatial smoothing, leading to a squeezed image. The detector compares the output vectors predicted by the CNN for squeezed and unsqueezed images. The $L_1$ distance between the two vectors is measured, and if it exceeds a certain threshold, the image is considered adversarial.

**Remark.** Ideally, we compare ShuffleDetect with well-known detectors, among which NIC, LID, ANR, and FS, are introduced above. However, our attempt to do so led us to face several highly challenging issues, among which the following: The codes of most of these detectors are not available, the claimed performances are on CNNs different from ours (Inception V3 trained on ImageNet for instance), or on CNNs trained on different datasets than ImageNet (such as CIFAR10 or MNIST for instance, which also implies that these CNNs use images of smaller sizes than ours), the used attacks are not systematically and clearly documented, the definitions of the used performance indicators vary from one paper to another. A thorough comparison would require implementing all relevant alternative detectors essentially from scratch, and challenging them under the same conditions as ShuffleDetect. We do not undertake this complete task here and keep it for future work. Nevertheless, we provide in Section 10 a limited comparison between ShuffleDetect and FS.

**Evaluation criteria.** In the present paper, the performance of the detector is evaluated with the following indicators [20]:

- Detection rate (DR) represents the percentage of adversarial images that are correctly identified as such by the detector.

- False positive rate (FPR) represents the percentage of clean ancestor images that are identified as adversarial by the detector.
- Complexity refers to the time required to train a supervised detector.
- Overhead refers to the overall memory and computation resources necessary to use the detector (supervised or not). It depends on the number of parameters and size of the architecture of the detector, when applicable.
- Inference time latency is the amount of time required by the detector to run on an image. If the method is supervised, the inference time latency does not take into account the time needed to train the detector (this part is already taken into account in the Complexity measurements).
- Precision, Recall, and F1 scores used to quantify the detection performance are defined by the following formulae:

$$\text{Precision} = \frac{\text{TP}}{\text{TP+FP}} \tag{1}$$

$$\text{Recall} = \frac{\text{TP}}{\text{TP+FN}} \tag{2}$$

$$\text{F1} = 2 \times \frac{\text{Recall} \times \text{Precision}}{\text{Recall} + \text{Precision}} \tag{3}$$

where TP (true positive) is the number of correctly detected adversarial images, FN (false negative) is the number of adversarial images that escaped the detector, and FP (false positive) is the number of clean images declared adversarial by the detector. These formulae are pertinent whenever the number of clean images is equal to the number of adversarial images created by a given attack for a given CNN. This aspect is taken into account in Section 9.

## 4. ShuffleDetect

The general goal of the shuffling process is to interchange different parts of an image. We noticed in [23] that if one shuffles a clean image, CNNs usually classify the shuffled image into the same category as the unshuffled clean image. We also noticed that the situation differs from an adversarial image because CNNs usually tend to classify the shuffled adversarial image that is no longer in the same category as the unshuffled adversarial image, at least for those created by the two attacks of [23] (which are considered again in Section 6).

These findings, valid for the two attacks, led to the detection method exposed below, which is based on the assumption that shuffling affects the adversarial noise more than it affects the image's original components, whichever the attack.

**Shuffling an image.** One is given image $\mathcal{I}$ of fixed (square) size $n \times n$ fitting the CNN's input size, and an integer $s$, such that patches of size $s \times s$ create a partition in the mathematical meaning of the term, or a grid in the more visual meaning of the term, of $\mathcal{I}$. This latter condition requires that $s$ divides $n$ since the number of patches is the integer $N_s = \left(\frac{n}{s}\right)^2$. It is convenient in practice to label the patch $P_{i,j}$, positioned in the $i$th column and $j$th row of the grid, as $P_k$, where $k = \frac{n}{s}(i-1) + j$ for $1 \le i, j \le \frac{n}{s}$ (see Table A2 in Appendix B for an example, which is used in our experiments actually).

The set of possible scrambles of an image of size $n \times n$ is essentially parametrized by the symmetric group $\mathfrak{S}_{N_s}$ of permutations of $N_s$ letters since $\mathfrak{S}_{N_s}$ operate on the set of $N_s$ patches. Indeed, a permutation $\sigma \in \mathfrak{S}_{N_s}$ is represented as a finite product of cycles, each of the form $(k_1, k_2, \ldots, k_M)$, these cycles having two-by-two disjoint supports. Each cycle symbolizes that the $M$ patches $P_{k_1}, P_{k_2}, \ldots, P_{k_M}$, associated to $k_1, k_2, \ldots, k_M$, respectively, are rotated in a circular way: $P_{k_1}$ takes the position of $P_{k_2}$, and so on until $P_{k_M}$ takes the position of $P_{k_1}$.

The group $\mathfrak{S}_{N_s}$ is of order $N_s!$, and is non-trivial provided $s$ is a strict divisor of $n$, which we assume from now on.



Given $\sigma \in \mathfrak{S}_{N_s}$, one denotes by $sh_\sigma(\mathcal{I}, s)$ the image obtained from $\mathcal{I}$ by swapping its patches according to $\sigma$. Both the unshuffled image $\mathcal{I}$ and the shuffled image $sh_\sigma(\mathcal{I}, s)$ are given to the CNN for classification. Figure 1 illustrates the process with the partition of an image into 4 patches $P_1, P_2, P_3, P_4$: The permutation $\sigma \in \mathfrak{S}_4$, selected among the altogether $4! = 24$ elements of $\mathfrak{S}_4$, is defined as the product of two cycles of length 2, which actually amounts to interchanging the patches on the diagonals.

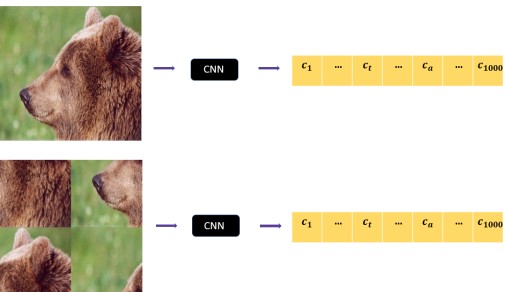

**Figure 1.** A $224 \times 224$ image $\mathcal{I}$ is divided into 4 patches of size $112 \times 112$ (top picture). The patches are shuffled around according to the permutation $\sigma = (1,4)(2,3) \in \mathfrak{S}_4$, leading to $sh_\sigma(\mathcal{I}, 112)$ (bottom picture). Both $\mathcal{I}$ and $sh_\sigma(\mathcal{I}, 112)$ are sent to the CNN to extract the output vector.

**ShuffleDetect.** To some extent, the global design of the algorithm ShuffleDetect mimics the design of classical probabilistic primality tests (such as those of Fermat, Solovay–Strassen, or Miller–Rabin, see [24], chapter 7 for instance), where the validity of an equation, which should be satisfied if a given integer $p$ is a prime, is checked for a series of rounds until either one has gained confidence (parameterized by the number of rounds) that $p$ is probably a prime or the equation is not satisfied for one of the rounds, in which case one knows that $p$ is not a prime. In our context, the equation, which assesses the detection of whether image $\mathcal{I}$ is adversarial or not, consists of comparing the dominating categories, given by a given CNN $\mathcal{C}$, before and after shuffling, for many permutations.

With consistent notations, one round of this detection method for image $\mathcal{I}$ is as follows. One picks at random a permutation $\sigma \in \mathfrak{S}_{N_s}$, with $\sigma \neq id$. Unless all of the patches of $\mathcal{I}$ addressed by $\sigma$ are identical (what happens if all $N_s$ patches of $\mathcal{I}$ are identical, which occurs, for instance, *a fortiori* if $\mathcal{I}$ is absolutely monochrome throughout all its pixels), $\sigma \neq id$ ensures that $\mathcal{I} \neq sh_\sigma(\mathcal{I}, s)$. The output of ShuffleDetect for $\mathcal{I}$ for the specific permutation $\sigma$, denoted by $\text{ShuffleDetect}^{\mathcal{C}}_\sigma(\mathcal{I})$, is:

$$\begin{array}{l} 1 \text{ if } \text{Dom}_\mathcal{C}(\mathcal{I}) \neq \text{Dom}_\mathcal{C}(sh_\sigma(\mathcal{I}, s)), \\ 0 \text{ if } \text{Dom}_\mathcal{C}(\mathcal{I}) = \text{Dom}_\mathcal{C}(sh_\sigma(\mathcal{I}, s)). \end{array} \qquad (4)$$

The image $\mathcal{I}$ is said $\sigma$-*adversarial* if $\text{ShuffleDetect}^{\mathcal{C}}_\sigma(\mathcal{I}) = 1$, and $\sigma$-*clean* if $\text{ShuffleDetect}^{\mathcal{C}}_\sigma(\mathcal{I}) = 0$.

For the full ShuffleDetect algorithm, written as $\text{ShuffleDetect}^{\mathcal{C}, R_{th}, t}(\mathcal{I})$ for the considered CNN $\mathcal{C}$ and image $\mathcal{I}$, one chooses a fixed number $t \in [1, N_s!]$ of rounds. For obvious practical reasons, $t$ should remain relatively small, in particular far smaller than $N_s!$. Then one selects at random $t$ two-by-two distinct permutations $\sigma_1, \ldots, \sigma_t \in \mathfrak{S}_{N_s}$, with $\sigma_r \neq id$ for all $1 \le r \le t$. One performs the successive $t$ rounds $\text{ShuffleDetect}^{\mathcal{C}}_{\sigma_1}(\mathcal{I}), \ldots, \text{ShuffleDetect}^{\mathcal{C}}_{\sigma_t}(\mathcal{I})$.

The threshold ratio $R_{th}$ is fixed as a percentage at will. For any number $t$ of permutations, the threshold ratio defines the integer $s_{th} = \lfloor t R_{th} \rfloor$, which is the number of permutations, such that $R_{th} \simeq \frac{s_{th}}{t}$.

The algorithm $\text{ShuffleDetect}^{\mathcal{C}, R_{th}, t}$ declares image $\mathcal{I}$:

- as "adversarial" for $\mathcal{C}$ if the output of $\text{ShuffleDetect}^{\mathcal{C}}_\sigma(\mathcal{I})$ is $\sigma$-adversarial for more than $s_{th}$ of the $t$ permutations $\sigma_1, \ldots, \sigma_t$,
- and as "clean" otherwise.

In more algorithmic terms, $\text{ShuffleDetect}^{\mathcal{C}, R_{th}, t}$ on image $\mathcal{I}$ works as described in the pseudo-code Algorithm 1. The user decides on the CNN $\mathcal{C}$, the degree of trust $R_{th}$, and

the number of permutations $t$ that index the rounds of the loop composing Steps (7) to (13). Once these parameters are chosen, Steps (1), (2), (3) are essentially setups, defined by the choices made for the parameters $\mathcal{C}, R_{th}, t$, while Steps (4), (5), (6) are essentially an initializing phase, depending only on $\mathcal{I}$ and $\mathcal{C}$. The choice of $R_{th}$ clearly determines the values of the indicators assessing the performance of the ShuffleDetect method (see Section 8 for a discussion on this issue and a recommended value).

---

**Algorithm 1** ShuffleDetect$^{\mathcal{C}, R_{th}, t}(\mathcal{I})$ pseudo-code

---

1: Compute and store $t$ permutations $\sigma_1, \dots, \sigma_t$
2: Select the size $s$ for the patches
3: Compute the integer $s_{th} = \lfloor t R_{th} \rfloor$
4: From $\mathcal{C}$, obtain the classification output vector $\mathbf{o}_{\mathcal{I}}$
5: Extract $\mathrm{Dom}_{\mathcal{C}}(\mathcal{I})$
6: Set $N = 0$
7: **For** $i$ from 1 to $t$ **run ShuffleDetect$_{\sigma_i}^{\mathcal{C}}(\mathcal{I})$ as follows:**
8:　　Create $sh_\sigma(\mathcal{I}, s)$.
9:　　From $\mathcal{C}$, obtain the classification output vector $\mathbf{o}_{sh_\sigma(\mathcal{I}, s)}$
10:　 Extract $\mathrm{Dom}_{\mathcal{C}}(sh_\sigma(\mathcal{I}, s))$.
11:　 Compare $\mathrm{Dom}_{\mathcal{C}}(\mathcal{I})$ and $\mathrm{Dom}_{\mathcal{C}}(sh_\sigma(\mathcal{I}, s))$.
12:　 Output 0 if they match, and 1 if they do not. In this latter case, $N := N + 1$.
13: **end**
14: Output "adversarial" if $N \geq s_{th}$, and "clean" otherwise.

---

**Remarks.** (1) Note that the process of comparing dominant categories does not require a precise assessment of their actual label values. Even in the case where an image is considered *σ-clean* for a given permutation $\sigma$, it is likely that, although the same category dominates both in the unshuffled and shuffled versions of the image, its label values differ strongly between both images.

(2) Although there is some flexibility *a priori* in setting the value of parameter $s$ at will, there are choices that turn out to be more appropriate for a given CNN's input size (see Section 8 for the choice of $s$ and its rationals for the experiments performed in this paper).

(3) When assessing many images of the same size, even if one fixes the number ($t$) of rounds once and for all, which is convenient in practice, there is still some flexibility in when to select the permutations. One option is to "reset" the random choice of $t$ permutations for each image to be tested. Another option is to proceed to the choice of the permutations at the same time as one chooses the value $t$, so that both $t$ and the set of $t$ random permutations $\sigma_1, \dots, \sigma_t$ are decided once for all images to test. There are pros and cons for both options, the former being (slightly) more time-consuming and (slightly) more memory-consuming but less biased, the latter saving time, allowing for an easier comparison and reproduction of the experiments, but providing a possible security leak because an attacker may ultimately guess what the $t$-selected permutations are and adapt to them accordingly. See Section 8 for the choices made in our experiments.

(4) Although there are theoretical measures and bounds of the proportion of composite numbers declared probably primes after $t$ rounds of a probabilistic test, there is no such thing regarding the proportion of adversarial images that are declared clean after $t$ rounds of ShuffleDetect. Therefore, for the time being, our choice of parameters is purely experimental.

(5) One can generalize the ShuffleDetect method thanks to the group of symmetries that preserve the square, namely the (non-abelian) dihedral group $D_8$ of order 8. Indeed, with consistent notations, and since each patch is a square, one could add to the action of a cycle $(k_1, k_2, \dots, k_M)$ of a permutation a randomly chosen sequence of elements $\gamma_{k_1}, \gamma_{k_2}, \dots, \gamma_{k_M} \in D_8$, which will act on the respective corresponding patches as well. We do not further explore this direction here, and stick to the exposed design of ShuffleDetect, which actually amounts to taking the identity for all symmetries $\gamma_{k_j} \in D_8$.

### 5. The CNNs, the Scenarios, the Ancestor Images

The selection of CNNs used in our experiments followed three criteria involving practicality, stability, and comparability. First, we required the availability of the pretrained versions of the CNNs in the PyTorch [17] library. Moreover, we required the CNNs to have stable architecture. Finally, to allow comparisons, despite their diversity in terms of architecture (number of layers, number of parameters, etc.), we required all CNNs to have the same image input size, and for this input size to be square (note that this later requirement is fulfilled by most CNNs in general).

This led us to select the following 10 well-known CNNs, trained on ImageNet [25], and with an input size of $224 \times 224$, namely $\mathcal{C}_1$ = VGG16 [26], $\mathcal{C}_2$ = VGG19 [26], $\mathcal{C}_3$ = ResNet50 [27], $\mathcal{C}_4$ = ResNet101 [27] and $\mathcal{C}_5$ = ResNet152 [27], $\mathcal{C}_6$ = DenseNet121 [28], $\mathcal{C}_7$ = DenseNet169 [28], $\mathcal{C}_8$ = DenseNet201 [28], $\mathcal{C}_9$ = MobileNet [29], and $\mathcal{C}_{10}$ = MNAS-Net [30].

Then, from the 1000 categories of ImageNet, we picked at random 10 ancestor classes and 10 corresponding target classes, as shown in Table 1.

**Table 1.** For $1 \le p \le 10$, the second column lists the ancestor category $c_{a_p}$ and its ordinal $1 \le a_p \le 1000$ among the categories of ImageNet. Mutatis mutandis in the third column with the target category $c_{t_p}$ and ordinal $t_p$.

| $p$ | $(c_{a_p}, a_p)$ | $(c_{t_p}, t_p)$ |
|---|---|---|
| 1 | (abacus, 398) | (bannister, 421) |
| 2 | (acorn, 988) | (rhinoceros beetle, 306) |
| 3 | (baseball, 429) | (ladle, 618) |
| 4 | (broom, 462) | (dingo, 273) |
| 5 | (brown bear, 294) | (pirate, 724) |
| 6 | (canoe, 472) | (saluki, 176) |
| 7 | (hippopotamus, 344) | (trifle, 927) |
| 8 | (llama, 355) | (agama, 42) |
| 9 | (maraca, 641) | (conch, 112) |
| 10 | (mountain bike, 671) | (strainer, 828) |

For each of the 10 ancestor classes ($1 \le p \le 10$), we randomly selected 10 ($1 \le q \le 10$) ancestor images $\mathcal{A}_q^p$ from the ImageNet validation set, classified as belonging to $c_{a_p}$ by the 10 CNNs. Whenever necessary, these ancestor images were resized to the CNNs common input size $224 \times 224$, thanks to the bilinear interpolation function [31]. Figure A1 and Table A1 in Appendix A present the 100 ancestor images $\mathcal{A}_q^p$ and their original sizes.

Starting with these 100 ancestor images, for each of the 10 CNNs listed above, the attacks, described in Section 6, were aimed at creating adversarial images either for the target scenario ($c_{a_p}, c_{t_p}$) of Table 1 (all CNNs produced negligible $c_{t_p}$-label values for the ancestors as a starting point) or for the untargeted scenario (in which case, it does not matter which category $c \ne c_a$ becomes dominant).

### 6. The 8 Attacks

This section presents the main features of the attacks employed in this paper and provides the chosen values for their parameters. Except for the EA attack, all attacks were applied using the Adversarial Robustness Toolbox (ART) [32], which is a Python library that includes several attack methods. ART functions and parameters used are specified in italics.

### 6.1. EA

Reference [19] is a black-box evolutionary algorithm-based attack that creates an initial population consisting of copies of the ancestor $X$ and modifies their pixels over generations. The goal of the EA is encoded in its fitness function, $fit(Ind) = o[c_t]_{Ind}$, where $Ind$ is a population individual and $o[c_t]_{Ind}$ is the individual's $c_t$ probability given by the CNN. The population size is set to 40, the magnitude by which a pixel could be mutated in one generation is $\alpha = 1/255$, the maximum mutation magnitude is $\epsilon = 8/255$, and the maximum number of generations is $N = 10{,}000$. We run both the targeted and untargeted versions of this attack. In the targeted case, for all CNNs, the threshold that dictates the adversarial image's minimum $c_t$ probability was set to meet the good enough requirements of [19].

### 6.2. FGSM

Reference [4], a white-box attack, is a one-step algorithm that calculates the gradient of the loss function $J(X, y)$ with respect to input $X$, to find the direction in which to modify $X$. In its untargeted version, the adversarial image is

$$X^{adv} = X + \epsilon sign(\Delta_X J(X, c_a)), \tag{5}$$

while in its targeted version, it is

$$X^{adv} = X - \epsilon sign(\Delta_X J(X, c_t)). \tag{6}$$

In the above equations, $\epsilon$ is the perturbation size, defined in the implementation by $eps = 2/255$, and $\Delta$ is the gradient function, as used in [4]. We use the *FastGradientMethod* function with the default value $eps\_step = 0.01$. We run both $targeted = True$ and $targeted = False$, corresponding to targeted and untargeted attacks, respectively.

### 6.3. BIM

Reference [3], a white-box attack, is an iterative version of FGSM. The adversarial image $X_0^{adv}$ is initialized with $X$ and is gradually updated for a given number of steps $N$, as follows:

$$X_{\ell+1}^{adv} = Clip_\epsilon \{X_\ell^{adv} + \alpha sign(\Delta_{\mathcal{A}}(J_C(X_\ell^{adv}, c_a)))\} \tag{7}$$

in its untargeted version and

$$X_{\ell+1}^{adv} = Clip_\epsilon \{X_\ell^{adv} - \alpha sign(\Delta_{\mathcal{A}}(J_C(X_\ell^{adv}, c_t)))\}, \tag{8}$$

in its targeted version, where $\alpha$ is the step size at each iteration and $\epsilon$ (which coincides with the ART function $eps$) is the maximum perturbation magnitude of $X^{adv} = X_N^{adv}$. We use the *BasicIterativeMethod* function with the default values $eps\_step = 0.01$, $max\_iter = int(eps \times 255 \times 1.25)$, and $eps = 1/255$. We run with both $targeted = True$ and $targeted = False$, corresponding to targeted and untargeted attacks, respectively.

### 6.4. PGD Inf

Reference [33], a white-box attack, is similar to the $BIM$ attack, with the difference that the image at the first attack iteration is not initialized with $X$, but rather with a random point situated within an $L_\infty$-ball around $X$. The distance between $X$ and $X^{adv}$ is measured using $L_\infty$ and the $\epsilon$ parameter represents the maximum perturbation magnitude. We use the *ProjectedGradientDescent* function with $norm = \inf$, and the default values $eps\_step = 0.01$, $batch\_size = 1$, and $eps = 1/255$. We run with both $targeted = True$ and $targeted = False$, corresponding to targeted and untargeted attacks, respectively.

### 6.5. PGD L1

Reference [33], a white-box attack, is similar to PGD Inf, with the difference that $L_\infty$ is replaced with $L_1$. We use the *ProjectedGradientDescent* function with $norm = 1$, and the

default values *eps_step* = 4, *batch_size* = 1, and *eps* = 30. We run with both *targeted = True* and *targeted = False*, corresponding to targeted and untargeted attacks, respectively.

### 6.6. PGD L2

Reference [33], a white-box attack, is similar to PGD Inf, with the difference that $L_\infty$ is replaced with $L_2$. We use the *ProjectedGradientDescent* function with *norm* = 2, and the default values *eps_step* = 0.1, *batch_size* = 1, and *eps* = 1. We run with both *targeted = True* and *targeted = False*, corresponding to targeted and untargeted attacks, respectively.

### 6.7. CW Inf

Reference [5], a white-box attack, solves the following optimization problem in its untargeted version:

$$\min_{\delta} \|\delta\| + cg(x'), \text{ such that } x' \in [0,1]^n, \tag{9}$$

$$\text{where } g(x') = \max\left(Z(x')_a - \max_{i \neq a} Z(x')_i, 0\right) \tag{10}$$

and $Z(x)$ is the pre-softmax classification output. The measure used to evaluate the difference between the ancestor $X$ and adversarial $X^{adv}$ is $L_\infty$. We use the *CarliniLInfMethod* function with the default values of the parameters. We ran with both *targeted = True* and *targeted = False*, corresponding to targeted and untargeted attacks, respectively.

### 6.8. DeepFool

Reference [34], a white-box attack, is an untargeted attack that calculates the minimum perturbation $\delta_*$ with which to modify $X$ such that its classification label changes, where $\delta_* = -f(X)w/\|w\|^2$, $f(X) = w^T x + b$, $F = \{x : f(x) = 0\}$. The attack solves the following optimization problem:

$$\underset{\delta_l}{\arg\min} \|\delta_l\|_2 \text{ such that } f(x_l) + \Delta f(x_l)^T \delta_l = 0. \tag{11}$$

The algorithm stops immediately after the label is changed, and $X^{adv} = X + \delta_*$. We use the *DeepFool* function with the default values of the parameters.

The seven attacks EA, FGSM, BIM, PGD Inf, PGD L1, PGD L2, and CW Inf are used both in the *untargeted* and in the *target* scenario, and the remaining DeepFool attack is used only in the context of the *untargeted* scenario. Apart from the black-box EA attack, all others are white-box attacks.

## 7. The Adversarial Images Obtained by the 8 Attacks

For each CNN $\mathcal{C}_k$ provided in Section 5, we run each of the 8 attacks *atk* given in Section 6, either for the *untargeted scenario* or for the *target scenario* whenever applicable, for the (potentially resized) 100 ancestor images $\mathcal{A}_q^p$, referred to in Section 5, and pictured in Figure A1, Appendix A. A successful attack for the *untargeted scenario* results in the image $\mathcal{D}_k^{atk,\text{untarget}}(\mathcal{A}_q^p)$, adversarial for $\mathcal{C}_k$ for that specific scenario. *Mutatis mutandis* with an adversarial image $\mathcal{D}_k^{atk,\text{target}}(\mathcal{A}_q^p)$ for the *target scenario*.

Since there are 8 untargeted and 7 targeted attacks, this amounts to $(8+7)$ attacks $\times$ 10 CNNs $\times$ 10 ancestor classes $\times$ 10 images per ancestor class. Out of these altogether 15,000 attack runs, 9746 were successful. More precisely, 6727 out of the 8000 untargeted attacks were successful, and there were 3019 successful targeted attacks out of the 7000 attempts, as detailed in Table 2.

Clearly, the number of successful attacks should be statistically relevant. We define this condition as satisfied if an attack succeeds in at least 35% of the cases for a given CNN (this value appears as a reasonable trade-off based on the experiments leading to Table 2). This leads us to disregard the targeted attacks performed by FGSM and CW Inf for all CNNs, as well as all attacks (untargeted and targeted) performed by PGD

L1 and the untargeted attack of FGSM on $\mathcal{C}_1$. The remaining 9580 statistically relevant successful attacks are listed in Table 3. The corresponding 2975 adversarial images for the target scenario and 6505 adversarial images for the untargeted scenario are considered in subsequent experiments.

**Table 2.** For each attack $atk$, and each $\mathcal{C}_k$, the number of successful runs performed on the 100 ancestors are presented. The results are given as a pair $(\alpha, \beta)$ or as a single value $\alpha$, depending on whether $atk$ is performed for both the untargeted and the targeted scenarios (assessed, respectively, by the values of $\alpha, \beta$ in the pair), or only the untargeted scenario (assessed by the single value of $\alpha$). The successful attacks on each individual CNN are given in the last row with obvious notations.

| $atk$ | $\mathcal{C}_1$ | $\mathcal{C}_2$ | $\mathcal{C}_3$ | $\mathcal{C}_4$ | $\mathcal{C}_5$ | $\mathcal{C}_6$ | $\mathcal{C}_7$ | $\mathcal{C}_8$ | $\mathcal{C}_9$ | $\mathcal{C}_{10}$ | Total |
|---|---|---|---|---|---|---|---|---|---|---|---|
| EA | (96, 91) | (97, 90) | (99, 88) | (98, 84) | (98, 79) | (99, 85) | (97, 89) | (98, 86) | (99, 97) | (99, 97) | (980, 886) |
| FGSM | (11, 0) | (83, 3) | (82, 2) | (81, 3) | (80, 2) | (86, 3) | (77, 4) | (80, 2) | (92, 13) | (89, 9) | (761, 41) |
| BIM | (93, 43) | (91, 38) | (96, 57) | (96, 52) | (93, 46) | (98, 56) | (95, 73) | (95, 50) | (95, 87) | (94, 78) | (946, 580) |
| PGD Inf | (93, 49) | (91, 38) | (96, 57) | (96, 52) | (93, 46) | (98, 56) | (95, 73) | (95, 50) | (95, 87) | (94, 78) | (946, 586) |
| PGD L1 | (26, 0) | (28, 1) | (19, 0) | (17, 1) | (12, 0) | (19, 1) | (15, 0) | (10, 0) | (33, 0) | (32, 0) | (211, 3) |
| PGD L2 | (93, 90) | (91, 88) | (97, 94) | (99, 92) | (96, 89) | (99, 94) | (98, 94) | (97, 86) | (96, 97) | (95, 99) | (961, 923) |
| CW Inf | (94, 0) | (95, 0) | (98, 0) | (99, 0) | (98, 0) | (100, 0) | (97, 0) | (99, 0) | (93, 0) | (94, 0) | (967, 0) |
| DeepFool | 94 | 97 | 92 | 97 | 94 | 100 | 94 | 97 | 96 | 94 | 955 |
| Total | (600, 273) | (673, 258) | (679, 298) | (683, 284) | (664, 262) | (699, 295) | (668, 333) | (671, 274) | (699, 381) | (691, 361) | (6727, 3019) |

**Table 3.** For each attack $atk$, and each $\mathcal{C}_k$, the number of successful runs performed on the 100 ancestors are presented, for which at least 35% were terminated successfully. The results are given as a pair $(\alpha, \beta)$ or as a single value $\alpha$, depending on whether $atk$ is performed for both the untargeted and the targeted scenarios (assessed, respectively, by the values of $\alpha, \beta$ in the pair), or only the untargeted scenario (assessed by the single value of $\alpha$). The statistically relevant successful attacks on each individual CNN are given in the last row with obvious notations.

| $atk$ | $\mathcal{C}_1$ | $\mathcal{C}_2$ | $\mathcal{C}_3$ | $\mathcal{C}_4$ | $\mathcal{C}_5$ | $\mathcal{C}_6$ | $\mathcal{C}_7$ | $\mathcal{C}_8$ | $\mathcal{C}_9$ | $\mathcal{C}_{10}$ | Total |
|---|---|---|---|---|---|---|---|---|---|---|---|
| EA | (96, 91) | (97, 90) | (99, 88) | (98, 84) | (98, 79) | (99, 85) | (97, 89) | (98, 86) | (99, 97) | (99, 97) | (980, 886) |
| FGSM | | 83 | 82 | 81 | 80 | 86 | 77 | 80 | 92 | 89 | 750 |
| BIM | (93, 43) | (91, 38) | (96, 57) | (96, 52) | (93, 46) | (98, 56) | (95, 73) | (95, 50) | (95, 87) | (94, 78) | (946, 580) |
| PGD Inf | (93, 49) | (91, 38) | (96, 57) | (96, 52) | (93, 46) | (98, 56) | (95, 73) | (95, 50) | (95, 87) | (94, 78) | (946, 586) |
| PGD L2 | (93, 90) | (91, 88) | (97, 94) | (99, 92) | (96, 89) | (99, 94) | (98, 94) | (97, 86) | (96, 97) | (95, 99) | (961, 923) |
| CW Inf | 94 | 95 | 98 | 99 | 98 | 100 | 97 | 99 | 93 | 94 | 967 |
| DeepFool | 94 | 97 | 92 | 97 | 94 | 100 | 94 | 97 | 96 | 94 | 955 |
| Total | (563, 273) | (645, 254) | (660, 296) | (666, 280) | (652, 260) | (680, 291) | (653, 329) | (661, 272) | (666, 368) | (659, 352) | (6505, 2975) |

## 8. Parameters and Experiments Performed on ShuffleDetect$_\sigma^{\mathcal{C}}$

In what follows, we essentially consider ShuffleDetect$_\sigma^{\mathcal{C}}$ for each individual permutation $\sigma$, each CNN $\mathcal{C}$, each clean image, and each image that is adversarial against $\mathcal{C}$. Altogether, the method is applied to all (resized if necessary) ancestors $\mathcal{A}_q^p$ on the one hand, as well as to all 2975 successful adversarials $\mathcal{D}_k^{atk,\text{target}}(\mathcal{A}_q^p)$ and all 6505 successful adversarials $\mathcal{D}_k^{atk,\text{untarget}}(\mathcal{A}_q^p)$ that compose Table 3 on the other hand. The ShuffleDetect parameters are specified below.

**Size of patches, number of permutations, and $\Psi_{\mathcal{C}}(t, s, \Omega)$.** Firstly, we selected $s = 56$ based on experiments detailed in [23]. Indeed, Table 4, extracted from [23], shows the average outcome for $2 \times 437$ adversarial images obtained from 84 common ancestor images (of size $224 \times 224$) for the same 10 CNNs considered here. The shuffling process was performed in [23] only for one permutation $\sigma$ per value of $s$ (hence $t = 1$ in this case) to obtain the values of Table 4.

**Table 4.** Percentages of shuffled images $sh_\sigma(\mathcal{A}_q^p, s)$ (first percentage), $sh_\sigma(D_k^{EA}(\mathcal{A}_q^p), s)$ (second percentage), and $sh_\sigma(D_k^{BIM}(\mathcal{A}_q^p), s)$ (third percentage) for which the predicted class is $c$.

| $s$ | Number of Patches | $c = c_a$ | $c \notin \{c_a, c_t\}$ | $c = c_t$ |
|-----|------------------|-----------|------------------------|-----------|
| 16 | 196 | 0.4, 0.1, 0.1 | 99.6, 99.9, 99.9 | 0.0, 0.0, 0.0 |
| 32 | 49 | 18.0, 9.2, 5.3 | 82.0, 90.8, 94.4 | 0.0, 0.0, 0.3 |
| 56 | 16 | 67.6, 39.3, 15.8 | 32.4, 60.3, 70.1 | 0.0, 0.4, 14.1 |
| 112 | 4 | 88.4, 62.3, 22.3 | 11.6, 33.2, 35.9 | 0.0, 4.5, 41.8 |

The experiments performed in [23] show that among the four considered possibilities, $s = 56$ provides an optimal balance between the proportion of clean ancestors that are correctly declared "clean" (67.6%), and the proportion of adversarial images that are correctly declared "adversarial" (99.6% for the adversarial images created by the EA, and 85.9% for those created by BIM) by our method. The choice of $s = 56$ being made, there are consequently $4^2 = 16$ patches of size $56 \times 56$, and the symmetric group $\mathfrak{S}_{16}$ has $16! > 2.10^{13}$ different permutations.

Secondly, to keep the computations manageable, we selected at random 100 permutations (they are given in Table A3 in Appendix B). For $1 \leq t \leq 100$, one defines $\mathcal{P}_t$ as the set of the first $t$ permutations. One has $\mathcal{P}_{t_1} \supset \mathcal{P}_{t_2}$ if $t_1 \geq t_2$. In particular, the first permutation $\sigma_1$ is common to all sets $\mathcal{P}_t$, the second permutation $\sigma_2$ is common to all sets $\mathcal{P}_t$ for $t \geq 2$, etc.

Given a set $\Omega$ of images and $\mathcal{C}$ a CNN, one defines the function $\Psi_{\mathcal{C}}(t, s, \Omega)$ as the proportion of images in $\Omega$ declared $\sigma$-adversarial for $s$ out of the first $t$ permutations. In other words, for $t$ and $s$ such that $1 \leq s \leq t \leq 100$, one has:

$$\Psi_{\mathcal{C}}(t, s, \Omega) = \frac{\#\{\mathcal{I} \in \Omega \text{ such that ShuffleDetect}_\sigma^{\mathcal{C}}(\mathcal{I}) = 1 \text{ for at least } s \text{ permutations } \sigma \in \mathcal{P}_t\}}{\#\{\mathcal{I} \in \Omega\}}.$$

Geometrically, $\mathcal{C}$ and $\Omega$ being fixed, $\Psi_{\mathcal{C}}(t, s, \Omega)$ defines a discrete surface. For a given $\mathcal{C}$, this function provides an assessment of the FPR value of the ShuffleDetect method for $\mathcal{C}$ by choosing for $\Omega$ a set of images known to be clean. This function also provides an assessment of the DR value by choosing for $\Omega$ a set of images known to be adversarial for $\mathcal{C}$.

As already stated in Section 4, the actual values of FPR and DR are determined by the choice of the threshold ratio $R_{th}$. Its value is fixed as a consequence of the experiments performed on clean images, on the one hand, and adversarial images, on the other hand.

**Assessment of the clean images.** In the first step, we take for $\Omega$ the set $\Omega_{clean}$ of 100 clean ancestors $\mathcal{A}_q^p$ represented in Figure A1 (Appendix A). For $\mathcal{C} = \mathcal{C}$, one computes ShuffleDetect$_\sigma^{\mathcal{C}_k}(\mathcal{A}_q^p)$ for all 100 permutations $\sigma \in \mathcal{P}_{100}$. This leads to the 10 histograms represented in Figure A2 in Appendix C. An example of the outcomes is illustrated in Figure 2a for $\mathcal{C}_1 =$ VGG16, where each vertical bar assesses the number of clean images classified as adversarial for a number of permutations given on the $x$-axis, out of the 100 possible permutations. The notations $[a, b]$ and $(a, b]$ indicate that the number of permutations is between $a$ and $b$, with both included in the former case and $a$ excluded in the latter case. The average outcome (*mutatis mutandis*) for 10 CNNs is shown in Figure 2b.

Over the 100 clean images, on average, over the 10 CNNs, an image is declared adversarial by 34.7% of the 100 considered permutations, as indicated in Figure 2b. Table 5 shows that this percentage varies between 26.4% (for $\mathcal{C}_7$) and 44.4% (for $\mathcal{C}_9$).

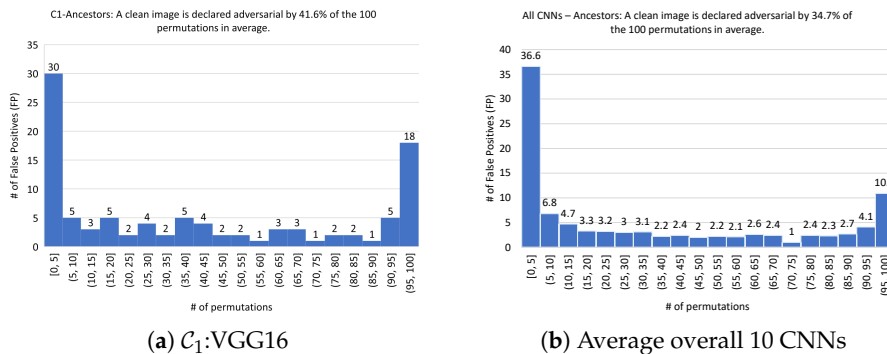

**Figure 2.** ShuffleDetect performed on 100 clean (ancestor) images with 100 permutations.

**Table 5.** For each $\mathcal{C}_k$, the number (=percentage) of clean ancestors $\mathcal{A}_q^p$ declared adversarial for $s$ out of 100 permutations. The first row shows the average number of permutations for which this occurs. The last row, the sum of the two previous ones, provides an estimate of the FPR, which serves as a lower bound for ShuffleDetect per CNN via the assessment of $\Psi_{\mathcal{C}}(100, 91, \Omega_{\mathrm{clean}})$.

| | $\mathcal{C}_1$ | $\mathcal{C}_2$ | $\mathcal{C}_3$ | $\mathcal{C}_4$ | $\mathcal{C}_5$ | $\mathcal{C}_6$ | $\mathcal{C}_7$ | $\mathcal{C}_8$ | $\mathcal{C}_9$ | $\mathcal{C}_{10}$ | **Average** |
|---|---|---|---|---|---|---|---|---|---|---|---|
| $s$ | 41.6 | 40.2 | 31.7 | 35.9 | 32.4 | 26.8 | 26.4 | 26.6 | 44.4 | 40.5 | 34.7 |
| (95–100] | 18 | 11 | 8 | 13 | 10 | 8 | 8 | 6 | 16 | 11 | 10.9 |
| (90–95] | 5 | 2 | 6 | 5 | 2 | 5 | 3 | 2 | 7 | 4 | 4.1 |
| $\Psi_{\mathcal{C}}(100, 91, \Omega_{\mathrm{clean}})$ | 23 | 13 | 14 | 18 | 12 | 13 | 11 | 8 | 23 | 15 | 15 |

The last row of Table 5 provides an estimate of a realistic FPR, which serves as a lower bound, or an "incompressible" FPR, whatever the choice of the parameter $R_{th}$. On average, over the 10 CNNs, $\Psi_{\mathcal{C}}(100, 91, \Omega_{\mathrm{clean}}) = 15\%$, and its value varies between 8% (for $\mathcal{C}_8$) and 23% (for $\mathcal{C}_1$ and $\mathcal{C}_9$). In this context, we noticed that some individual clean images were declared adversarial by ShuffleDetect$_\sigma^{\mathcal{C}}$ for all CNNs $\mathcal{C}$ by a large number (and, therefore, a proportion) of permutations $\sigma$. Indeed, the 7 clean images $\mathcal{A}_3^9$, $\mathcal{A}_6^5$, $\mathcal{A}_6^9$, $\mathcal{A}_9^1$, $\mathcal{A}_9^5$, $\mathcal{A}_9^6$, and $\mathcal{A}_9^7$ are declared adversarial for all CNNs by more than 91 permutations. Whatever the ratio threshold $R_{th}$, these 7 images contribute substantially to the FPR of ShuffleDetect$^{\mathcal{C}, R_{th}, t}$ for a specific CNN individually, and *a fortiori* for the FPR average taken over all CNNs.

**Assessment of the adversarial images.** In the second step, for $\mathcal{C} = \mathcal{C}$, we take for $\Omega$ the set $\Omega_{adv,k}^{scenario}$ of adversarial images $\mathcal{D}_k^{atk,scenario}(\mathcal{A}_q^p)$ as of Table 3. One computes the values of ShuffleDetect$_\sigma^{\mathcal{C}_k}$ for these images for all 100 permutations $\sigma \in \mathcal{P}_{100}$, and one defines

$$s_{min}^{scenario}(k, adv) = \underset{1 \le i \le 100}{Max} \{i; \Psi_{\mathcal{C}_k}(100, i, \Omega_{adv,k}^{scenario}) = M\%\}$$

which captures the optimum index that makes sure that $\Psi_{\mathcal{C}_k}(100, s_{min}^{scenario}(adv, k), \Omega_{adv,k}^{scenario}) = M\%$, where $M\%$ is the maximum possible detection rate of adversarial images created by the given attack on the given CNN. Clearly $M\% = 100\%$ if there are no adversarial images $\mathcal{D}_k^{atk,scenario}(\mathcal{A}_q^p)$ for which

$$\mathrm{Dom}_{\mathcal{C}_k}(\mathcal{D}_k^{atk,scenario}(\mathcal{A}_q^p)) = \mathrm{Dom}_{\mathcal{C}_k}(sh_\sigma(\mathcal{D}_k^{atk,scenario}(\mathcal{A}_q^p), 56))$$

for all 100 permutations $\sigma$. While this eventuality does not occur in our experiments with the target scenario, we shall see that it does for the untargeted scenario for many attacks and many CNNs.

We proceed firstly with the target scenario. This leads to the 40 histograms (obtained from 4 targeted attacks performed on the 10 CNNs) represented in Figures A3–A6 in Appendix C. The following Figure 3 shows the average behavior over the 10 CNNs of the ShuffleDetect method for all 100 permutations in the adversarial images created by each targeted attack. Note that the *y*-axis indicates the average number of adversarial images

for a given attack, and all CNNs are taken together, as derived from Table 3. For example, there are 580/10 = 58 adversarial images on average for the target scenario for BIM.

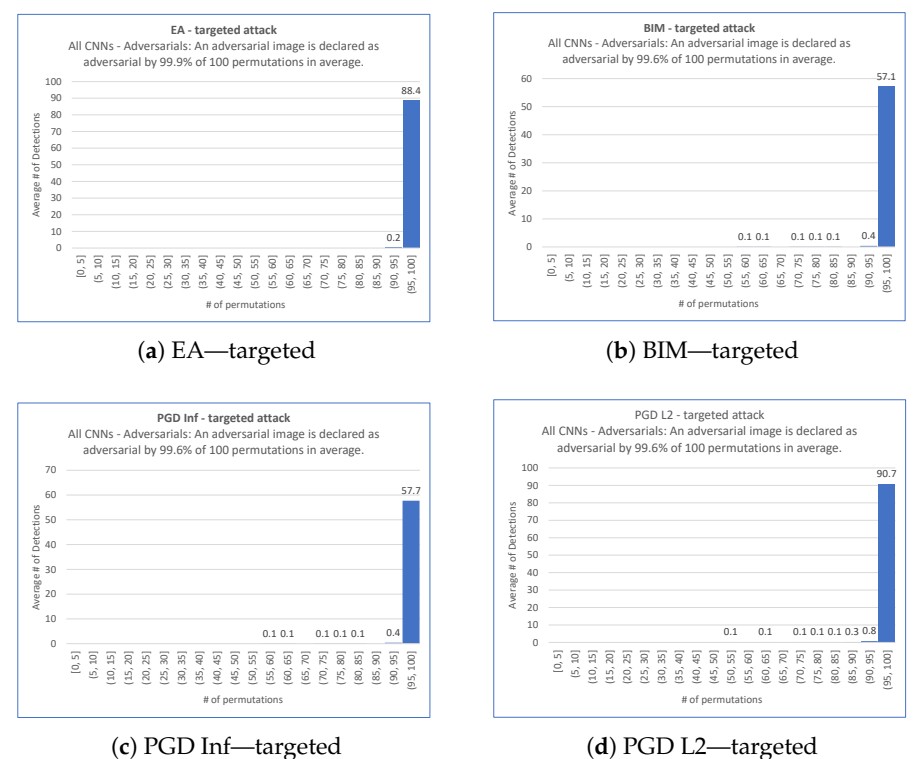

(**a**) EA—targeted

(**b**) BIM—targeted

(**c**) PGD Inf—targeted

(**d**) PGD L2—targeted

**Figure 3.** Average outcome over the 10 CNNs of ShuffleDetect performed with 100 permutations on the adversarial images created for the target scenario by EA, BIM, PGD Inf, and PGD L2.

Table 6 details the outcomes for each CNN individually, and provides an assessment of the detection rate DR of the ShuffleDetect method for the detection of adversarial images for the target scenario created by each of the four attacks. More precisely, for each $\mathcal{C}_k$, for each targeted attack $atk$, the table first provides the percentage $s$ of the 100 permutations $\sigma$ for which the shuffled-by $\sigma$ image of an adversarial image, namely $\text{ShuffleDetect}_\sigma^{\mathcal{C}_k}\left(\mathcal{D}_k^{atk,targeted}(\mathcal{A}_q^p)\right)$, is declared adversarial on average. Consistently with assessments performed on the clean images, Table 6 provides $\Psi_{\mathcal{C}_k}(100,91,\Omega_{adv,k}^{target})$. Note that the number of elements of $\Omega_{adv,k}^{target}$ used to compute these values is equal to the corresponding value $\beta$ from Table 3. Finally, Table 6 provides the values of $M\%$ and of $s_{min}^{target}(k,adv)$.

We proceed with the untargeted scenario. This leads to the 69 histograms (derived from 6 untargeted attacks performed on the 10 CNNs, and from the FGSM untargeted attack performed on 9 CNNs) represented in Figures A7–A13 in Appendix C. The following Figure 4 provides the average behavior over the 10 (or 9 in the case of FGSM) CNNs of the ShuffleDetect method for all 100 permutations on the adversarial images created by each untargeted attack. Note that the $y$-axis indicates the average number of adversarial images for a given attack, all relevant CNNs taken together, derived from Table 3. For instance, there are 750/9 = 83.3 adversarial images on average for the untargeted scenario for FGSM.

With notations consistent with the already handled case of targeted attacks, Table 7 details the outcome for each CNN individually, and provides an assessment of the DR of the ShuffleDetect method applied to adversarial images for the untargeted scenarios created by each of the seven attacks. The number of elements of $\Omega_{adv,k}^{untarget}$ used to compute the values of $\Psi_{\mathcal{C}_k}(100,91,\Omega_{adv,k}^{untarget})$ is equal to the corresponding value $\alpha$ from Table 3.

**Table 6.** For each $\mathcal{C}_k$, for each targeted attack *atk*, percentage *s* of the 100 permutations $\sigma$ for which the shuffled-by $\sigma$ image of an adversarial image, namely ShuffleDetect$_\sigma^{\mathcal{C}_k}\left(\mathcal{D}_k^{atk,targeted}(\mathcal{A}_q^p)\right)$, is declared adversarial on average, assessment of $\Psi_{\mathcal{C}_k}(100,91,\Omega_{adv,k}^{target})$, of the maximum possible detection rate $M\%$, and of $s_{min}^{target}(k,adv)$.

| Targeted Attacks | $\mathcal{C}_1$ | $\mathcal{C}_2$ | $\mathcal{C}_3$ | $\mathcal{C}_4$ | $\mathcal{C}_5$ | $\mathcal{C}_6$ | $\mathcal{C}_7$ | $\mathcal{C}_8$ | $\mathcal{C}_9$ | $\mathcal{C}_{10}$ | Average |
|---|---|---|---|---|---|---|---|---|---|---|---|
| EA-targeted | 100 | 100 | 100 | 99.8 | 99.8 | 99.9 | 99.9 | 100 | 100 | 99.9 | 99.9 |
| $\Psi_{\mathcal{C}_k}(100,91,\Omega_{\text{EA},k}^{target})$ | 100 | 100 | 100 | 100 | 100 | 100 | 100 | 100 | 100 | 100 | 100 |
| $M\%$ | 100 | 100 | 100 | 100 | 100 | 100 | 100 | 100 | 100 | 100 | 100 |
| $s_{min}^{target}(k,EA)$ | 100 | 100 | 100 | 93 | 95 | 96 | 98 | 100 | 100 | 96 | 97.8 |
| BIM-targeted | 100 | 99.7 | 99.8 | 99.4 | 99.2 | 99.6 | 99.6 | 99.9 | 99.9 | 99.2 | 99.6 |
| $\Psi_{\mathcal{C}_k}(100,91,\Omega_{\text{BIM},k}^{target})$ | 100 | 100 | 100 | 98 | 97.8 | 98.2 | 98.6 | 100 | 100 | 98.7 | 99.1 |
| $M\%$ | 100 | 100 | 100 | 100 | 100 | 100 | 100 | 100 | 100 | 100 | 100 |
| $s_{min}^{target}(k,BIM)$ | 100 | 92 | 97 | 72 | 64 | 85 | 77 | 99 | 98 | 58 | 84.2 |
| PGD Inf-targeted | 99.9 | 99.7 | 99.8 | 99.4 | 99.2 | 99.6 | 99.6 | 99.9 | 99.9 | 99.2 | 99.6 |
| $\Psi_{\mathcal{C}_k}(100,91,\Omega_{\text{PGD Inf},k}^{target})$ | 100 | 100 | 100 | 98 | 97.8 | 98.2 | 98.6 | 100 | 100 | 98.7 | 99.1 |
| $M\%$ | 100 | 100 | 100 | 100 | 100 | 100 | 100 | 100 | 100 | 100 | 100 |
| $s_{min}^{target}(k,PGDInf)$ | 99 | 92 | 97 | 72 | 64 | 85 | 77 | 99 | 98 | 58 | 84.1 |
| PGD L2-targeted | 99.7 | 99.6 | 99.8 | 99.6 | 99.5 | 99.6 | 99.5 | 99.7 | 99.9 | 99.2 | 99.6 |
| $\Psi_{\mathcal{C}_k}(100,91,\Omega_{\text{PGD L2},k}^{target})$ | 100 | 100 | 100 | 98.9 | 98.8 | 97.8 | 98.9 | 98.8 | 100 | 97.9 | 99.1 |
| $M\%$ | 100 | 100 | 100 | 100 | 100 | 100 | 100 | 100 | 100 | 100 | 100 |
| $s_{min}^{target}(k,PGDL2)$ | 95 | 92 | 96 | 72 | 62 | 85 | 78 | 87 | 98 | 54 | 81.9 |

**Table 7.** For each $\mathcal{C}_k$, for each untargeted attack *atk*, the percentage *s* of the 100 permutations $\sigma$ for which the shuffled-by $\sigma$ image of an adversarial image, namely ShuffleDetect$_\sigma^{\mathcal{C}_k}\left(\mathcal{D}_k^{atk,untargeted}(\mathcal{A}_q^p)\right)$, is declared adversarial on average, the assessment of $\Psi_{\mathcal{C}_k}(100,91,\Omega_{adv,k}^{untarget})$, of the maximum possible detection rate $M\%$, and of $s_{min}^{untarget}(k,adv)$.

| Untargeted Attacks | $\mathcal{C}_1$ | $\mathcal{C}_2$ | $\mathcal{C}_3$ | $\mathcal{C}_4$ | $\mathcal{C}_5$ | $\mathcal{C}_6$ | $\mathcal{C}_7$ | $\mathcal{C}_8$ | $\mathcal{C}_9$ | $\mathcal{C}_{10}$ | Average |
|---|---|---|---|---|---|---|---|---|---|---|---|
| EA-untargeted | 86.0 | 86.9 | 91.1 | 94.4 | 90.6 | 92.2 | 93.3 | 95.0 | 91.7 | 90.7 | 91.2 |
| $\Psi_{\mathcal{C}_k}(100,91,\Omega_{\text{EA},k}^{untarget})$ | 69.7 | 64.9 | 78.7 | 83.6 | 78.5 | 77.7 | 85.5 | 86.7 | 85.8 | 78.7 | 79.0 |
| $M\%$ | 100 | 100 | 98.9 | 100 | 100 | 100 | 100 | 100 | 98.9 | 98.9 | 99.6 |
| $s_{min}^{untarget}(k,EA)$ | 2 | 15 | 22 | 1 | 1 | 1 | 2 | 4 | 3 | 24 | 7.5 |
| FGSM-untargeted | NA | 75.3 | 84.8 | 89.2 | 82.2 | 81.5 | 84.2 | 86.4 | 82.2 | 84.1 | 83.3 |
| $\Psi_{\mathcal{C}_k}(100,91,\Omega_{\text{EA},k}^{untarget})$ | NA | 45.7 | 63.4 | 74.0 | 53.7 | 56.9 | 68.8 | 58.7 | 63.0 | 61.7 | 60.7 |
| $M\%$ | NA | 97.5 | 98.7 | 98.7 | 98.8 | 98.7 | 98.7 | 98.9 | 97.7 | 99.7 | 98.6 |
| $s_{min}^{untarget}(k,FGSM)$ | NA | 2 | 12 | 4 | 1 | 1 | 16 | 12 | 3 | 17 | 7.5 |
| BIM-untargeted | 67.0 | 68.7 | 84.1 | 90.1 | 83.2 | 79.2 | 86.7 | 86.8 | 81.6 | 77.8 | 80.6 |
| $\Psi_{\mathcal{C}_k}(100,91,\Omega_{\text{EA},k}^{untarget})$ | 40.8 | 40.6 | 64.5 | 75 | 66.6 | 56.1 | 74.7 | 62.1 | 68.4 | 44.6 | 59.3 |
| $M\%$ | 97.8 | 94.5 | 98.9 | 98.9 | 97.8 | 97.9 | 98.9 | 98.9 | 97.8 | 98.9 | 98.0 |
| $s_{min}^{untarget}(k,BIM)$ | 1 | 1 | 3 | 9 | 9 | 3 | 10 | 21 | 1 | 2 | 6.0 |
| PGD Inf-untargeted | 67.0 | 68.6 | 84.1 | 90.0 | 83.2 | 79.2 | 86.7 | 86.8 | 81.6 | 77.8 | 80.6 |
| $\Psi_{\mathcal{C}_k}(100,91,\Omega_{\text{EA},k}^{untarget})$ | 40.8 | 39.5 | 64.5 | 75 | 66.6 | 56.1 | 74.7 | 62.1 | 68.4 | 44.6 | 59.2 |
| $M\%$ | 97.8 | 94.5 | 98.9 | 98.9 | 97.8 | 97.9 | 98.9 | 98.9 | 97.8 | 98.9 | 98.0 |
| $s_{min}^{untarget}(k,PGDInf)$ | 1 | 1 | 3 | 9 | 9 | 3 | 10 | 21 | 1 | 2 | 6.0 |
| PGD L2-untargeted | 66.9 | 59.6 | 78.6 | 87.8 | 80.3 | 74.3 | 82.9 | 81.9 | 75.6 | 69.1 | 75.9 |
| $\Psi_{\mathcal{C}_k}(100,91,\Omega_{\text{EA},k}^{untarget})$ | 43.0 | 30.7 | 52.5 | 68.6 | 59.3 | 51.5 | 66.3 | 53.6 | 51.0 | 37.8 | 51.4 |
| $M\%$ | 96.7 | 92.3 | 97.9 | 98.9 | 97.9 | 97.9 | 98.9 | 98.9 | 96.8 | 97.8 | 97.4 |
| $s_{min}^{untarget}(k,PGDL2)$ | 1 | 1 | 13 | 1 | 8 | 2 | 3 | 8 | 1 | 3 | 4.1 |

**Table 7.** *Cont.*

| Untargeted Attacks | $\mathcal{C}_1$ | $\mathcal{C}_2$ | $\mathcal{C}_3$ | $\mathcal{C}_4$ | $\mathcal{C}_5$ | $\mathcal{C}_6$ | $\mathcal{C}_7$ | $\mathcal{C}_8$ | $\mathcal{C}_9$ | $\mathcal{C}_{10}$ | Average |
|---|---|---|---|---|---|---|---|---|---|---|---|
| CW Inf-untargeted | 79.4 | 82.0 | 90.4 | 91.5 | 88.4 | 86.5 | 90.8 | 91.2 | 83.9 | 87.8 | 87.3 |
| $\Psi_{\mathcal{C}_k}(100, 91, \Omega_{\text{EA},k}^{untarget})$ | 61.7 | 58.9 | 76.5 | 77.7 | 73.4 | 67.0 | 77.3 | 74.7 | 64.5 | 65.9 | 69.7 |
| $M\%$ | 97.8 | 98.9 | 98.9 | 98.9 | 98.9 | 99 | 98.9 | 98.9 | 97.8 | 98.9 | 98.7 |
| $s_{min}^{untarget}(k, CWInf)$ | 7 | 1 | 15 | 18 | 24 | 10 | 24 | 27 | 1 | 25 | 15.2 |
| DeepFool-untargeted | 90.5 | 90.7 | 93.0 | 95.6 | 92.8 | 93.3 | 93.8 | 94.8 | 91.5 | 92.1 | 92.8 |
| $\Psi_{\mathcal{C}_k}(100, 91, \Omega_{\text{EA},k}^{untarget})$ | 76.5 | 78.3 | 81.5 | 88.6 | 82.9 | 83 | 82.9 | 83.5 | 81.2 | 82.9 | 82.1 |
| $M\%$ | 100 | 100 | 98.9 | 100 | 100 | 99 | 100 | 100 | 98.9 | 98.9 | 99.5 |
| $s_{min}^{untarget}(k, DeepFool)$ | 1 | 2 | 35 | 16 | 12 | 28 | 5 | 12 | 1 | 12 | 12.4 |

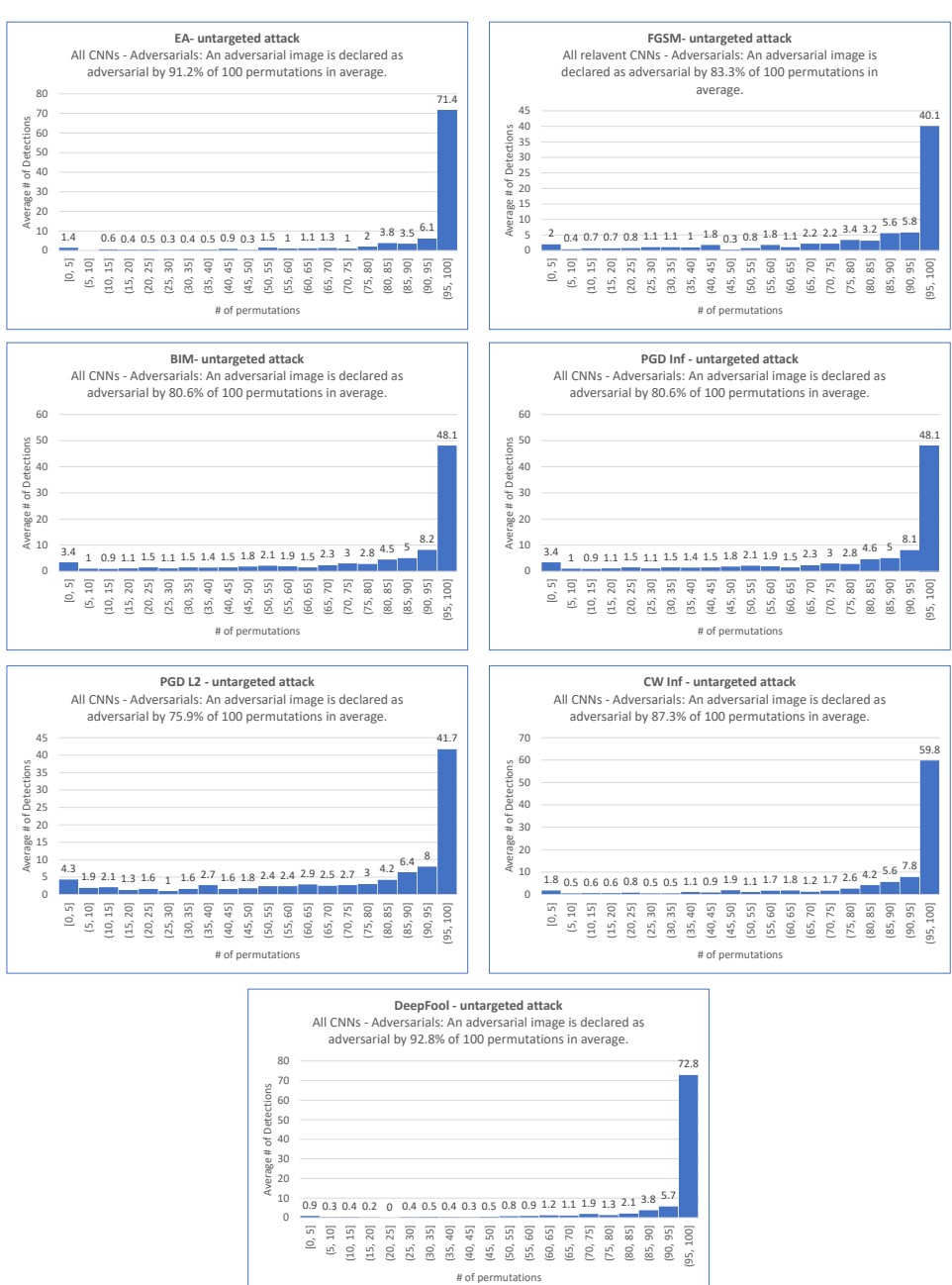

**Figure 4.** Average outcome over all relevant CNNs of ShuffleDetect performed with 100 permutations on the adversarial images created for the untargeted scenario by EA, FGSM, BIM, PGD Inf, PGD L2, CW Inf, and DeepFool.

## 9. Intrinsic Performance of ShuffleDetect$^{\mathcal{C},R_{th},t}$

**Indicators and performance.** Since ShuffleDetect$^{\mathcal{C},R_{th},t}$ is an unsupervised detector, the complexity criterion does not apply. The values of the remaining indicators do depend on the number $t$ of permutations to be considered, and most of them are determined by the selected threshold ratio $R_{th}$.

To assess the inference time latency, the creation of $t = 100$ permutations (Step 1 of Algorithm 1) took 0.064 s using the command SymmetricGroup(16) and 100 calls of the command RandomElement on Maple 2022 (this timing could certainly be optimized). Running ShuffleDetect$^{\mathcal{C}}_{\sigma}$ for a single permutation $\sigma$ (Steps 8 to 12 of Algorithm 1) takes 0.0784 s/permutation on average (over 100 considered permutations, overall 10 CNNs, and over 100 random clean images because considering them is sufficient to assess this average). The time required by Steps 2, 3, 6, and 14 (all of which are outside the loop of the $t$ permutations) is negligible. The overall inference time latency of ShuffleDetect performed on an image with $t = 100$ permutations amounts to $\simeq 0.064 + 1 \times 0.0784 + 100 \times 0.0784 = 0.064 + 7.918 = 7.982$ s/image on average. On the one hand, the prediction process performed by the CNN (one time in Steps 4 and 5 for the unshuffled image, and $t = 100$ times in Steps 9 and 10 for the shuffled images) contributes to $\simeq 98.02\%$ of this time consumption. On the other hand, the shuffling process (Step 8, called $t = 100$ times) contributes to $\simeq 1.98\%$ of this time consumption. See Appendix B, Table A4 for detailed information on all CNNs.

One should take into account two positive aspects of the proposed detector. Firstly, the 0.064 s consumed by the creation of the 100 permutations can be mutualized over several calls of the detector for different input images. Secondly, the tasks performed iteratively (Steps (7) to (13)) can be easily distributed; thus, apart from the time required for the creation of the 100 permutations, the algorithm would require only $\simeq 0.0784$, plus some minor time due to the gathering of the distributed information, and the final computation and comparison.

The Overhead is very limited (and can be optimized). Algorithm 1 shows that the "permanent storage" is limited to the $t = 100$ permutations expressed as products of cycles as in Table A3, Appendix B (which actually can be called upon if the permutations are computed once for all images to handle as we do in our experiments), the integer $s_{th}$, and the extracted dominating category $\mathrm{Dom}_{\mathcal{C}}(\mathcal{I})$ (which amounts to a numbering among the 1000 categories of ImageNet in our case). The "incremental storage" is made of the value 0 or 1 as $\sigma$ progresses throughout the $t$ permutations, hence (at most) $t$ such Boolean values if one wants to keep the whole information. A memory-saving alternative is to keep only the updated $N$ as $\sigma$ progresses throughout the $t$ permutations. The "ephemeral storage" (deleted after each run) is composed of the running images $sh_{\sigma}(\mathcal{I}, s)$ and $\mathrm{Dom}_{\mathcal{C}}(sh_{\sigma}(\mathcal{I}, s))$. The computation resources are essentially limited to the creation of $t$ permutations (to be done once at the beginning, as recommended), to $1 + t$ calls to the CNN for the classification of $\mathcal{I}$ and $sh(\mathcal{I}, \sigma_i)$, and to the creation of the (up to) $t$ shuffled images $sh_{\sigma}(\mathcal{I}, s)$. Finding the dominant category, as is necessary once for $\mathcal{I}$ and (at most) $t$ times for the shuffled images, amounts to looking for the largest value in the classification output category, which is immediate in a set of 1000 values, as is the case here.

Note that in what precedes, we mention "at most" a few times since one could stop the loop before its natural end. This is the case if after some rounds the running threshold reaches such a value that the remaining rounds, whatever happens, cannot ensure that the threshold ratio $R_{th}$ will not be reached.

The specific value chosen for $R_{th}$ clearly impacts the different indicators of the ShuffleDetect algorithm (but, foremost, FPR and DR). To summarize, the smaller the $R_{th}$, the higher the DR and FPR. However, our experiments show that a high $R_{th}$ leads to a very good DR and a moderate FPR. A b-moll to this statement is that the situation differs according to the nature of the "targeted" or "untargeted" attack, as we shall see now.

For targeted attacks, Table 6 together with Figure A2 led us to consider (for $t = 100$ permutations) four choices for the value for $R_{th}$:

- $R_{th}$ = 51% matches the requirement that most permutations declare an image as adversarial.
- $R_{th}$ = 54% is motivated by the fact that the smallest $s_{min}^{target}(k, atk)$ among the 100 permutations and the four targeted attacks is $\geq 54$.
- $R_{th}$ = 87% is motivated by the fact that the average of the $s_{min}^{target}(k, atk)$ among the 100 permutations and the four targeted attacks is =87.
- $R_{th}$ = 91% as a demanding ratio compromise.

For untargeted attacks, Table 7 together with Figures A7–A13 (Appendix C) show that (for $t$ = 100 permutations) using $s_{min}^{untarget}(k, atk)$ is irrelevant for the selection of $R_{th}$. Indeed, $s_{min}^{untarget}(k, atk)$ is usually small. More precisely, $s_{min}^{untarget}(k, atk) \leq 35$ in all cases, and is = 9.8 on average, as opposed to what occurs for the target scenario, where $s_{min}^{target}(k, atk) \geq 54$ in all cases, and is = 87 on average. Therefore, we limit the selection of $R_{th}$ to two values:

- $R_{th}$ = 51% for the same reason as for the target case.
- $R_{th}$ = 91% because it makes sense to keep the same demanding $R_{th}$ value for the detector independently on the scenario of the attack, hence the same value for the targeted attack.

The values of FP and FPR depend only on the value of $R_{th}$ (since no attack is considered for their computation), and on the CNN. Note that FPR = FP/100 for $t$ = 100 permutations. One writes $FP_{avg}$ and $FPR_{avg}$ for their respective average values over the 10 considered CNNs. Table 8 provides the corresponding values for $R_{th}$ = 51%, 54%, 87%, and 91% (the four values used in the context of the target scenario also contain the two values used in the context of the untargeted scenario) for $t$ = 100 permutations.

**Table 8.** Table of FP per CNN for each selected value of $R_{th}$; FPR is deduced from FP by the formula FPR = FP/100.

| $R_{th}$ | $C_1$ | $C_2$ | $C_3$ | $C_4$ | $C_5$ | $C_6$ | $C_7$ | $C_8$ | $C_9$ | $C_{10}$ | $FP_{avg}$ |
|---|---|---|---|---|---|---|---|---|---|---|---|
| 51 | 38 | 38 | 30 | 34 | 33 | 26 | 25 | 26 | 40 | 37 | 32.7 |
| 54 | 37 | 36 | 29 | 33 | 30 | 25 | 23 | 25 | 40 | 36 | 31.4 |
| 87 | 24 | 13 | 16 | 19 | 14 | 15 | 13 | 12 | 29 | 19 | 17.4 |
| 91 | 23 | 13 | 14 | 18 | 12 | 13 | 11 | 8 | 23 | 15 | 15 |

**Remark.** The number of adversarial images against each CNN, created either by targeted or untargeted attacks, is in all cases strictly less than the number of clean ancestor images from which these attacks started. As mentioned at the end of Section 3, this imbalance should be considered to have a fair comparison basis and sound values for the indicators (what we measure is the performance of the indicator, not of the attack). Therefore, in Tables 9–12 the clean images selected are those that correspond to the adversarial images obtained from them. For instance, since the EA-targeted attack succeeded to create "only" 91 images adversarial against $C_1$, we consider only the exact 91 clean images from which these adversarial images were obtained to assess the FP value.

For targeted attacks, Tables 9 and 10 show (for $t$ = 100 permutations) the DR (which coincides with $\Psi_{C_k}(100, 100R_{th}, \Omega_{atk,k}^{target})$), TP, FN, precision, recall, and F1 score values per CNN per targeted attack for each of the four selected values of $R_{th}$, as well as their average values.

For untargeted attacks, Tables 11 and 12 show (for $t$ = 100 permutations) the DR (which coincides with $\Psi_{C_k}(100, 100R_{th}, \Omega_{atk,k}^{untarget})$), TP, FP, FN, precision, recall, F1 scores per CNN per untargeted attack for each selected value of $R_{th}$, as well as their average values.

**Conclusion for the intrinsic performance of ShuffleDetect.** Regarding targeted attacks, Tables 9 and 10 show that the difference in values of the indicators obtained when $R_{th}$ = 0.54 versus 0.51 (respectively, 0.87 versus 0.91) remains marginal.

If one knows the nature of "targeted" and "untargeted" attacks, and/or if one knows which specific attack to expect, one can choose the most appropriate threshold ratio value $R_{th}$. However, one rarely has access to this intelligence in practice.

Consequently, it makes sense to consider *a priori* only $R_{th}$ = 0.51 or $R_{th}$ = 0.91 whatever the attack (hence, *a fortiori* whatever its targeted or untargeted nature). Table 13 provides the values of all indicators per CNN on average over the 4 targeted attacks. It also provides the worst $F_1$ value as a proxy of the worst case for our detector. Similar information for untargeted attacks is given in Table 14.

Tables 13 and 14 show that our detector achieves very good results. For instance (when both $R_{th}$ = 0.51 and 0.91 are considered) for the two highly significant indicators made of the detection rate and the F1 values:

- For all targeted attacks, the detection rate is ≥98.55, the F1 value is ≥0.76, and the average values of these indicators are 99.67 and 0.87, respectively.
- For untargeted attacks, the detection rate is ≥51.23, the F1 value is ≥0.60, and the average values are 76.77 and 0.75, respectively.

Recall that a defender does not know the nature (targeted or untargeted) of an attack he is exposed to. For the sake of completeness, Table 15 provides the values of all indicators per CNN in the average overall attacks, targeted and not targeted, for the two values $R_{th}$ = 0.51 and 0.91.

Now, as a defender, it is wise to consider the values of the indicators given in Table 14 for untargeted attacks, since then one is also "on the safe side" for targeted attacks as well.

A remaining issue is whether one can achieve results as those given in Table 14 (allowing one to be "on the safe side", as pointed out above), say for DR, precision, recall, and F1, with less than 100 permutations. For instance for $C_1$, can one achieve (DR, precision, recall, F1) = (77.5, 0.7, 0.8, 0.7) in less than 100 permutations? Indeed, doing so would clearly speed up the process (see Table A4 to assess time savings per spared permutation).

We performed a series of tests with increasing values of the number of permutations, aimed at indicator values, as those of Table 14. More precisely, we fixed the indicator values as those of Table 14, and we added permutations one by one (following their numbering, as given in Table A3), and stopped when we achieved those fixed indicator values. Note that the minimal number of permutations, with which it makes sense, from a mathematical point of view, to start this process, depends on the value of $R_{th}$.

For $R_{th}$ = 0.51, it makes sense to consider $t \geq 3$, while for $R_{th}$ = 0.91, it makes sense to consider $t \geq 12$. Therefore, for each CNN $C$, for each attack *atk*, targeted or untargeted accordingly, starting with the first 3 permutations for $R_{th}$ = 0.51 (respectively, the first 12 permutations for $R_{th}$ = 0.91), we added the subsequent permutations whenever appropriate, and stopped the process when the minimal number $t_{optimal,C,atk}$ of permutations fulfilling the above criteria was achieved. Table 16 provides the outcome of this experiment.

Finally, which value for $R_{th}$ do we privilege? We considered the DR indicator as the most significant one to make our choice. With this indicator, we concluded that the "democratic" value $R_{th}$ = 0.51 is an appropriate and reasonable choice for most applications of ShuffleDetect. In terms of the number of permutations, one can use the number $t_{optimal,C}$ = $\text{Max}_{atk}\{t_{optimal,C,atk}\}$, defined for $R_{th}$ = 0.51 in Table 16, according to the CNN $C$ considered. This value is convenient for the relevant 4 targeted attacks and 7 untargeted attacks studied here. However, especially in view of the low time and memory price to pay for additional permutations, we consider that a defender who uses 100 permutations is better prepared against unknown attacks. Refinements in this regard are still possible, especially since ShuffleDetect is on the defender's side, the defender knows which CNNs to protect so that he can adapt accordingly.

**Table 9.** Targeted attacks—DR as a percentage, and TP, FP, FN, precision, recall, F1 scores for CNN, per attack, for selected values of $R_{th}$ = 0.51 and 0.54, and their corresponding averages.

| $R_{th}$ | Targeted Attack | Metrics | $\mathcal{C}_1$ | $\mathcal{C}_2$ | $\mathcal{C}_3$ | $\mathcal{C}_4$ | $\mathcal{C}_5$ | $\mathcal{C}_6$ | $\mathcal{C}_7$ | $\mathcal{C}_8$ | $\mathcal{C}_9$ | $\mathcal{C}_{10}$ | Avg |
|---|---|---|---|---|---|---|---|---|---|---|---|---|---|
| | | DR | 100 | 100 | 100 | 100 | 100 | 100 | 100 | 100 | 100 | 100 | **100** |
| | | TP | 91 | 90 | 88 | 84 | 79 | 85 | 89 | 86 | 97 | 97 | **88.6** |
| | | FP | 37 | 37 | 28 | 32 | 29 | 25 | 23 | 25 | 39 | 37 | **31.2** |
| | EA | FN | 0 | 0 | 0 | 0 | 0 | 0 | 0 | 0 | 0 | 0 | **0** |
| | | Precision | 0.71 | 0.70 | 0.75 | 0.72 | 0.73 | 0.77 | 0.79 | 0.77 | 0.71 | 0.72 | **0.73** |
| | | Recall | 1 | 1 | 1 | 1 | 1 | 1 | 1 | 1 | 1 | 1 | **1** |
| | | F1 | 0.83 | 0.82 | 0.85 | 0.83 | 0.84 | 0.87 | 0.88 | 0.87 | 0.83 | 0.83 | **0.84** |
| | | DR | 100 | 100 | 100 | 100 | 100 | 100 | 100 | 100 | 100 | 100 | **100** |
| | | TP | 43 | 38 | 57 | 52 | 46 | 56 | 73 | 50 | 87 | 78 | **58** |
| | | FP | 23 | 23 | 20 | 19 | 21 | 18 | 17 | 15 | 35 | 31 | **22.2** |
| | BIM | FN | 0 | 0 | 0 | 0 | 0 | 0 | 0 | 0 | 0 | 0 | **0** |
| | | Precision | 0.65 | 0.62 | 0.74 | 0.73 | 0.68 | 0.75 | 0.81 | 0.76 | 0.71 | 0.71 | **0.71** |
| | | Recall | 1 | 1 | 1 | 1 | 1 | 1 | 1 | 1 | 1 | 1 | **1** |
| | | F1 | 0.78 | 0.76 | 0.85 | 0.84 | 0.8 | 0.85 | 0.89 | 0.86 | 0.83 | 0.83 | **0.82** |
| 0.51 | | DR | 100 | 100 | 100 | 100 | 100 | 100 | 100 | 100 | 100 | 100 | **100** |
| | | TP | 49 | 38 | 57 | 52 | 46 | 56 | 73 | 50 | 87 | 78 | **58.6** |
| | | FP | 23 | 23 | 20 | 19 | 21 | 18 | 17 | 15 | 35 | 31 | **22.2** |
| | PGD Inf | FN | 0 | 0 | 0 | 0 | 0 | 0 | 0 | 0 | 0 | 0 | **0** |
| | | Precision | 0.68 | 0.62 | 0.74 | 0.73 | 0.68 | 0.75 | 0.81 | 0.76 | 0.71 | 0.71 | **0.71** |
| | | Recall | 1 | 1 | 1 | 1 | 1 | 1 | 1 | 1 | 1 | 1 | **1** |
| | | F1 | 0.80 | 0.76 | 0.85 | 0.84 | 0.80 | 0.85 | 0.89 | 0.86 | 0.83 | 0.83 | **0.83** |
| | | DR | 100 | 100 | 100 | 100 | 100 | 100 | 100 | 100 | 100 | 100 | **100** |
| | | TP | 90 | 88 | 94 | 92 | 89 | 94 | 94 | 86 | 97 | 99 | **92.3** |
| | | FP | 23 | 23 | 20 | 31 | 29 | 26 | 24 | 24 | 38 | 36 | **27.4** |
| | PGD L2 | FN | 0 | 0 | 0 | 0 | 0 | 0 | 0 | 0 | 0 | 0 | **0** |
| | | Precision | 0.79 | 0.79 | 0.82 | 0.74 | 0.75 | 0.78 | 0.79 | 0.78 | 0.71 | 0.73 | **0.76** |
| | | Recall | 1 | 1 | 1 | 1 | 1 | 1 | 1 | 1 | 1 | 1 | **1** |
| | | F1 | 0.88 | 0.88 | 0.90 | 0.85 | 0.85 | 0.87 | 0.88 | 0.87 | 0.83 | 0.84 | **0.86** |
| | | DR | 100 | 100 | 100 | 100 | 100 | 100 | 100 | 100 | 100 | 100 | **100** |
| | | TP | 91 | 90 | 88 | 84 | 79 | 85 | 89 | 86 | 97 | 97 | **88.6** |
| | | FP | 36 | 35 | 27 | 31 | 26 | 24 | 21 | 24 | 39 | 36 | **29.9** |
| | EA | FN | 0 | 0 | 0 | 0 | 0 | 0 | 0 | 0 | 0 | 0 | **0** |
| | | Precision | 0.71 | 0.72 | 0.76 | 0.73 | 0.75 | 0.77 | 0.80 | 0.78 | 0.71 | 0.72 | **0.74** |
| | | Recall | 1 | 1 | 1 | 1 | 1 | 1 | 1 | 1 | 1 | 1 | **1** |
| | | F1 | 0.83 | 0.83 | 0.86 | 0.84 | 0.85 | 0.87 | 0.88 | 0.87 | 0.83 | 0.83 | **0.84** |
| | | DR | 100 | 100 | 100 | 100 | 100 | 100 | 100 | 100 | 100 | 100 | **100** |
| | | TP | 43 | 38 | 57 | 52 | 46 | 56 | 73 | 50 | 87 | 78 | **58** |
| | | FP | 22 | 22 | 19 | 19 | 19 | 18 | 16 | 14 | 35 | 30 | **21.4** |
| | BIM | FN | 0 | 0 | 0 | 0 | 0 | 0 | 0 | 0 | 0 | 0 | **0** |
| | | Precision | 0.66 | 0.63 | 0.75 | 0.73 | 0.70 | 0.75 | 0.82 | 0.78 | 0.71 | 0.72 | **0.72** |
| | | Recall | 1 | 1 | 1 | 1 | 1 | 1 | 1 | 1 | 1 | 1 | **1** |
| | | F1 | 0.79 | 0.77 | 0.85 | 0.84 | 0.82 | 0.85 | 0.90 | 0.87 | 0.83 | 0.83 | **0.83** |
| 0.54 | | DR | 100 | 100 | 100 | 100 | 100 | 100 | 100 | 100 | 100 | 100 | **100** |
| | | TP | 49 | 38 | 57 | 52 | 46 | 56 | 73 | 50 | 87 | 78 | **58.6** |
| | | FP | 22 | 22 | 19 | 19 | 19 | 18 | 16 | 14 | 35 | 30 | **21.4** |
| | PGD Inf | FN | 0 | 0 | 0 | 0 | 0 | 0 | 0 | 0 | 0 | 0 | **0** |
| | | Precision | 0.69 | 0.63 | 0.75 | 0.73 | 0.70 | 0.75 | 0.82 | 0.78 | 0.71 | 0.72 | **0.72** |
| | | Recall | 1 | 1 | 1 | 1 | 1 | 1 | 1 | 1 | 1 | 1 | **1** |
| | | F1 | 0.81 | 0.77 | 0.85 | 0.84 | 0.82 | 0.85 | 0.9 | 0.87 | 0.83 | 0.83 | **0.83** |
| | | DR | 100 | 100 | 100 | 100 | 100 | 100 | 100 | 100 | 100 | 100 | **100** |
| | | TP | 90 | 88 | 94 | 92 | 89 | 94 | 94 | 86 | 97 | 99 | **92.3** |
| | | FP | 22 | 22 | 19 | 30 | 26 | 25 | 22 | 23 | 38 | 35 | **26.2** |
| | PGD L2 | FN | 0 | 0 | 0 | 0 | 0 | 0 | 0 | 0 | 0 | 0 | **0** |
| | | Precision | 0.80 | 0.80 | 0.83 | 0.75 | 0.77 | 0.78 | 0.81 | 0.78 | 0.71 | 0.73 | **0.77** |
| | | Recall | 1 | 1 | 1 | 1 | 1 | 1 | 1 | 1 | 1 | 1 | **1** |
| | | F1 | 0.88 | 0.88 | 0.90 | 0.85 | 0.87 | 0.87 | 0.89 | 0.87 | 0.83 | 0.84 | **0.86** |

**Table 10.** Targeted attacks—DR as a percentage, and TP, FP, FN, precision, recall, F1 scores per CNN per attack for each selected value of $R_{th}$ = 0.87 and 0.91, and their corresponding averages.

| $R_{th}$ | Targeted Attack | Metrics | $C_1$ | $C_2$ | $C_3$ | $C_4$ | $C_5$ | $C_6$ | $C_7$ | $C_8$ | $C_9$ | $C_{10}$ | Avg |
|---|---|---|---|---|---|---|---|---|---|---|---|---|---|
| 0.87 | EA | DR | 100 | 100 | 100 | 100 | 100 | 100 | 100 | 100 | 100 | 100 | **100** |
| | | TP | 91 | 90 | 88 | 84 | 79 | 85 | 89 | 86 | 97 | 97 | **88.6** |
| | | FP | 23 | 13 | 16 | 18 | 12 | 15 | 12 | 12 | 29 | 19 | **16.9** |
| | | FN | 0 | 0 | 0 | 0 | 0 | 0 | 0 | 0 | 0 | 0 | **0** |
| | | Precision | 0.79 | 0.87 | 0.84 | 0.82 | 0.86 | 0.85 | 0.88 | 0.87 | 0.76 | 0.83 | **0.83** |
| | | Recall | 1 | 1 | 1 | 1 | 1 | 1 | 1 | 1 | 1 | 1 | **1** |
| | | F1 | 0.88 | 0.93 | 0.91 | 0.90 | 0.92 | 0.91 | 0.93 | 0.93 | 0.86 | 0.90 | **0.9** |
| | BIM | DR | 100 | 100 | 100 | 98.1 | 97.8 | 98.2 | 98.6 | 100 | 100 | 98.7 | **99.1** |
| | | TP | 43 | 38 | 57 | 51 | 45 | 55 | 72 | 50 | 87 | 77 | **57.5** |
| | | FP | 17 | 9 | 11 | 15 | 11 | 11 | 11 | 9 | 27 | 18 | **13.9** |
| | | FN | 0 | 0 | 0 | 1 | 1 | 1 | 1 | 0 | 0 | 1 | **0.5** |
| | | Precision | 0.71 | 0.80 | 0.83 | 0.77 | 0.80 | 0.83 | 0.86 | 0.84 | 0.76 | 0.81 | **0.8** |
| | | Recall | 1 | 1 | 1 | 0.98 | 0.978 | 0.982 | 0.986 | 1 | 1 | 0.987 | **0.99** |
| | | F1 | 0.83 | 0.88 | 0.90 | 0.86 | 0.88 | 0.89 | 0.91 | 0.91 | 0.86 | 0.88 | **0.87** |
| | PGD Inf | DR | 100 | 100 | 100 | 98 | 97.8 | 98.2 | 98.6 | 100 | 100 | 98.7 | **99.1** |
| | | TP | 49 | 38 | 57 | 51 | 45 | 55 | 72 | 50 | 87 | 77 | **58.1** |
| | | FP | 17 | 9 | 11 | 15 | 11 | 11 | 11 | 9 | 27 | 18 | **13.9** |
| | | FN | 0 | 0 | 0 | 1 | 1 | 1 | 1 | 0 | 0 | 1 | **0.5** |
| | | Precision | 0.74 | 0.80 | 0.83 | 0.77 | 0.80 | 0.83 | 0.86 | 0.84 | 0.76 | 0.81 | **0.8** |
| | | Recall | 1 | 1 | 1 | 0.98 | 0.978 | 0.982 | 0.986 | 1 | 1 | 0.987 | **0.99** |
| | | F1 | 0.85 | 0.88 | 0.90 | 0.86 | 0.88 | 0.89 | 0.91 | 0.91 | 0.86 | 0.88 | **0.88** |
| | PGD L2 | DR | 100 | 100 | 100 | 98.9 | 98.8 | 98.9 | 98.9 | 100 | 100 | 98.9 | **99.4** |
| | | TP | 90 | 88 | 94 | 91 | 88 | 93 | 93 | 86 | 97 | 98 | **91.8** |
| | | FP | 17 | 9 | 11 | 19 | 12 | 15 | 12 | 11 | 28 | 19 | **15.3** |
| | | FN | 0 | 0 | 0 | 1 | 1 | 1 | 1 | 0 | 0 | 1 | **0.5** |
| | | Precision | 0.84 | 0.90 | 0.89 | 0.82 | 0.88 | 0.86 | 0.88 | 0.88 | 0.77 | 0.83 | **0.85** |
| | | Recall | 1 | 1 | 1 | 0.98 | 0.98 | 0.98 | 0.98 | 1 | 1 | 0.98 | **0.99** |
| | | F1 | 0.91 | 0.94 | 0.94 | 0.89 | 0.92 | 0.91 | 0.92 | 0.93 | 0.87 | 0.89 | **0.91** |
| 0.91 | EA | DR | 100 | 100 | 100 | 100 | 100 | 100 | 100 | 100 | 100 | 100 | **100** |
| | | TP | 91 | 90 | 88 | 84 | 79 | 85 | 89 | 86 | 97 | 97 | **88.6** |
| | | FP | 23 | 13 | 14 | 17 | 11 | 13 | 10 | 8 | 23 | 15 | **14.7** |
| | | FN | 0 | 0 | 0 | 0 | 0 | 0 | 0 | 0 | 0 | 0 | **0** |
| | | Precision | 0.79 | 0.87 | 0.86 | 0.83 | 0.87 | 0.86 | 0.89 | 0.91 | 0.80 | 0.86 | **0.85** |
| | | Recall | 1 | 1 | 1 | 1 | 1 | 1 | 1 | 1 | 1 | 1 | **1** |
| | | F1 | 0.88 | 0.93 | 0.92 | 0.90 | 0.93 | 0.92 | 0.94 | 0.95 | 0.88 | 0.92 | **0.91** |
| | BIM | DR | 100.00 | 100.00 | 100.00 | 98.00 | 97.80 | 98.20 | 98.60 | 100.00 | 100.00 | 98.70 | **99.10** |
| | | TP | 43 | 38 | 57 | 51 | 45 | 55 | 72 | 50 | 87 | 77 | **57.5** |
| | | FP | 17 | 9 | 9 | 15 | 10 | 9 | 9 | 7 | 22 | 14 | **12.1** |
| | | FN | 0 | 0 | 0 | 1 | 1 | 1 | 1 | 0 | 0 | 1 | **0.5** |
| | | Precision | 0.71 | 0.80 | 0.86 | 0.77 | 0.81 | 0.85 | 0.88 | 0.87 | 0.79 | 0.84 | **0.81** |
| | | Recall | 1 | 1 | 1 | 0.98 | 0.978 | 0.982 | 0.986 | 1 | 1 | 0.987 | **0.99** |
| | | F1 | 0.83 | 0.88 | 0.92 | 0.86 | 0.88 | 0.91 | 0.92 | 0.93 | 0.88 | 0.90 | **0.89** |
| | PGD Inf | DR | 100 | 100 | 100 | 98 | 97.8 | 98.2 | 98.6 | 100 | 100 | 98.7 | **99.1** |
| | | TP | 49 | 38 | 57 | 51 | 45 | 55 | 72 | 50 | 87 | 77 | **58.1** |
| | | FP | 17 | 9 | 9 | 15 | 10 | 9 | 9 | 7 | 22 | 14 | **12.1** |
| | | FN | 0 | 0 | 0 | 1 | 1 | 1 | 1 | 0 | 0 | 1 | **0.5** |
| | | Precision | 0.74 | 0.80 | 0.86 | 0.77 | 0.81 | 0.85 | 0.88 | 0.87 | 0.79 | 0.84 | **0.82** |
| | | Recall | 1 | 1 | 1 | 0.98 | 0.97 | 0.98 | 0.98 | 1 | 1 | 0.98 | **0.99** |
| | | F1 | 0.85 | 0.88 | 0.92 | 0.86 | 0.88 | 0.91 | 0.92 | 0.93 | 0.88 | 0.90 | **0.89** |
| | PGD L2 | DR | 100 | 100 | 100 | 98.9 | 98.8 | 97.8 | 98.9 | 98.8 | 100 | 97.9 | **99.1** |
| | | TP | 90 | 88 | 94 | 91 | 88 | 92 | 93 | 85 | 97 | 97 | **91.5** |
| | | FP | 17 | 9 | 9 | 18 | 11 | 13 | 10 | 8 | 22 | 15 | **13.2** |
| | | FN | 0 | 0 | 0 | 1 | 1 | 2 | 1 | 1 | 0 | 2 | **0.8** |
| | | Precision | 0.84 | 0.90 | 0.91 | 0.83 | 0.88 | 0.87 | 0.90 | 0.91 | 0.81 | 0.86 | **0.87** |
| | | Recall | 1 | 1 | 1 | 0.989 | 0.988 | 0.978 | 0.989 | 0.988 | 1 | 0.979 | **0.99** |
| | | F1 | 0.91 | 0.94 | 0.95 | 0.9 | 0.93 | 0.92 | 0.94 | 0.94 | 0.89 | 0.91 | **0.91** |

**Table 11.** Untargeted attacks—DR as a percentage, and TP, FP, FN, precision, recall, F1 scores per CNN per attack for each selected value of $R_{th}$ = 0.51, and their corresponding averages.

| $R_{th}$ | Untargeted Attack | Metrics | $C_1$ | $C_2$ | $C_3$ | $C_4$ | $C_5$ | $C_6$ | $C_7$ | $C_8$ | $C_9$ | $C_{10}$ | Avg |
|---|---|---|---|---|---|---|---|---|---|---|---|---|---|
| | EA | DR | 88.5 | 90.7 | 93.9 | 96.9 | 96.9 | 95.9 | 96.9 | 97.9 | 93.9 | 93.9 | **94.5** |
| | | TP | 85 | 88 | 93 | 95 | 95 | 95 | 94 | 96 | 93 | 93 | **92.7** |
| | | FP | 38 | 38 | 29 | 34 | 33 | 26 | 25 | 26 | 39 | 36 | **32.4** |
| | | FN | 11 | 9 | 6 | 3 | 3 | 4 | 3 | 2 | 6 | 6 | **5.3** |
| | | Precision | 0.69 | 0.69 | 0.76 | 0.73 | 0.74 | 0.78 | 0.78 | 0.78 | 0.70 | 0.72 | **0.74** |
| | | Recall | 0.88 | 0.90 | 0.93 | 0.96 | 0.96 | 0.95 | 0.96 | 0.97 | 0.93 | 0.93 | **0.94** |
| | | F1 | 0.77 | 0.78 | 0.83 | 0.82 | 0.83 | 0.85 | 0.86 | 0.86 | 0.79 | 0.81 | **0.82** |
| | FGSM | DR | | 80.7 | 89.0 | 92.5 | 86.2 | 87.2 | 85.7 | 92.5 | 85.8 | 86.5 | **87.3** |
| | | TP | | 67 | 73 | 75 | 69 | 75 | 66 | 74 | 79 | 77 | **72.7** |
| | | FP | | 33 | 28 | 31 | 29 | 23 | 22 | 23 | 34 | 30 | **28.1** |
| | | FN | | 16 | 9 | 6 | 11 | 11 | 11 | 6 | 13 | 12 | **10.5** |
| | | Precision | | 0.67 | 0.72 | 0.70 | 0.70 | 0.76 | 0.75 | 0.76 | 0.69 | 0.71 | **0.72** |
| | | Recall | | 0.80 | 0.89 | 0.92 | 0.86 | 0.87 | 0.85 | 0.92 | 0.85 | 0.86 | **0.87** |
| | | F1 | | 0.72 | 0.79 | 0.79 | 0.77 | 0.81 | 0.79 | 0.83 | 0.76 | 0.77 | **0.78** |
| | BIM | DR | 66.6 | 72.5 | 86.4 | 93.7 | 86.0 | 83.6 | 88.4 | 92.6 | 84.2 | 84.0 | **83.8** |
| | | TP | 62 | 66 | 83 | 90 | 80 | 82 | 84 | 88 | 80 | 79 | **79.4** |
| | | FP | 34 | 32 | 28 | 32 | 29 | 24 | 22 | 24 | 33 | 31 | **28.9** |
| | | FN | 31 | 25 | 13 | 6 | 13 | 16 | 11 | 7 | 15 | 15 | **15.2** |
| | | Precision | 0.64 | 0.67 | 0.74 | 0.73 | 0.73 | 0.77 | 0.79 | 0.78 | 0.70 | 0.71 | **0.73** |
| | | Recall | 0.66 | 0.72 | 0.86 | 0.93 | 0.86 | 0.83 | 0.88 | 0.92 | 0.84 | 0.84 | **0.83** |
| | | F1 | 0.64 | 0.69 | 0.79 | 0.81 | 0.78 | 0.79 | 0.83 | 0.84 | 0.76 | 0.76 | **0.77** |
| 0.51 | PGD Inf | DR | 66.6 | 72.5 | 86.4 | 93.7 | 86.0 | 83.6 | 88.4 | 92.6 | 84.2 | 84.0 | **83.8** |
| | | TP | 62 | 66 | 83 | 90 | 80 | 82 | 84 | 88 | 80 | 79 | **79.4** |
| | | FP | 34 | 32 | 28 | 32 | 29 | 24 | 22 | 24 | 33 | 31 | **28.9** |
| | | FN | 31 | 25 | 13 | 6 | 13 | 16 | 11 | 7 | 15 | 15 | **15.2** |
| | | Precision | 0.64 | 0.67 | 0.74 | 0.73 | 0.73 | 0.77 | 0.79 | 0.78 | 0.70 | 0.71 | **0.73** |
| | | Recall | 0.66 | 0.72 | 0.86 | 0.93 | 0.86 | 0.83 | 0.88 | 0.92 | 0.84 | 0.84 | **0.83** |
| | | F1 | 0.64 | 0.69 | 0.79 | 0.81 | 0.78 | 0.79 | 0.83 | 0.84 | 0.76 | 0.76 | **0.77** |
| | PGD L2 | DR | 68.8 | 61.5 | 80.4 | 91.9 | 83.3 | 77.7 | 86.7 | 88.6 | 78.1 | 73.6 | **79.0** |
| | | TP | 64 | 56 | 78 | 91 | 80 | 77 | 85 | 86 | 75 | 70 | **76.2** |
| | | FP | 34 | 31 | 27 | 32 | 29 | 24 | 22 | 24 | 34 | 32 | **28.9** |
| | | FN | 29 | 35 | 19 | 8 | 16 | 22 | 13 | 11 | 21 | 25 | **19.9** |
| | | Precision | 0.65 | 0.64 | 0.74 | 0.73 | 0.73 | 0.76 | 0.79 | 0.78 | 0.68 | 0.68 | **0.72** |
| | | Recall | 0.68 | 0.61 | 0.80 | 0.91 | 0.83 | 0.77 | 0.86 | 0.88 | 0.78 | 0.73 | **0.79** |
| | | F1 | 0.66 | 0.62 | 0.76 | 0.81 | 0.77 | 0.76 | 0.82 | 0.82 | 0.72 | 0.7 | **0.75** |
| | CW Inf | DR | 79.7 | 86.3 | 93.8 | 93.9 | 91.8 | 90.0 | 93.8 | 96.9 | 88.1 | 89.3 | **90.3** |
| | | TP | 75 | 82 | 92 | 93 | 90 | 90 | 91 | 96 | 82 | 84 | **87.5** |
| | | FP | 34 | 33 | 28 | 32 | 30 | 25 | 21 | 24 | 31 | 30 | **28.8** |
| | | FN | 19 | 13 | 6 | 6 | 8 | 10 | 6 | 3 | 11 | 10 | **9.2** |
| | | Precision | 0.68 | 0.71 | 0.76 | 0.74 | 0.75 | 0.78 | 0.81 | 0.80 | 0.72 | 0.73 | **0.75** |
| | | Recall | 0.79 | 0.86 | 0.93 | 0.93 | 0.91 | 0.90 | 0.93 | 0.96 | 0.88 | 0.89 | **0.90** |
| | | F1 | 0.73 | 0.77 | 0.83 | 0.82 | 0.82 | 0.83 | 0.86 | 0.87 | 0.79 | 0.8 | **0.81** |
| | DeepFool | DR | 94.6 | 93.8 | 95.6 | 96.9 | 97.8 | 96.0 | 97.8 | 97.9 | 94.7 | 93.6 | **95.8** |
| | | TP | 89 | 91 | 88 | 94 | 92 | 96 | 92 | 95 | 91 | 88 | **91.6** |
| | | FP | 36 | 37 | 27 | 33 | 28 | 25 | 22 | 25 | 37 | 32 | **30.2** |
| | | FN | 5 | 6 | 4 | 3 | 2 | 4 | 2 | 2 | 5 | 6 | **3.9** |
| | | Precision | 0.71 | 0.71 | 0.76 | 0.74 | 0.76 | 0.79 | 0.80 | 0.79 | 0.71 | 0.73 | **0.75** |
| | | Recall | 0.94 | 0.93 | 0.95 | 0.96 | 0.97 | 0.96 | 0.97 | 0.97 | 0.94 | 0.93 | **0.95** |
| | | F1 | 0.8 | 0.8 | 0.84 | 0.83 | 0.85 | 0.86 | 0.87 | 0.87 | 0.8 | 0.81 | **0.83** |

**Table 12.** Untargeted attacks—DR as a percentage, and TP, FP, FN, precision, recall, F1 scores per CNN per attack for each selected value of $R_{th}$ = 0.91, and their corresponding averages.

| $R_{th}$ | Untargeted Attack | Metrics | $\mathcal{C}_1$ | $\mathcal{C}_2$ | $\mathcal{C}_3$ | $\mathcal{C}_4$ | $\mathcal{C}_5$ | $\mathcal{C}_6$ | $\mathcal{C}_7$ | $\mathcal{C}_8$ | $\mathcal{C}_9$ | $\mathcal{C}_{10}$ | Avg |
|---|---|---|---|---|---|---|---|---|---|---|---|---|---|
| | EA | DR | 69.7 | 64.9 | 78.7 | 83.6 | 78.5 | 77.7 | 85.5 | 86.7 | 85.8 | 78.7 | **78.9** |
| | | TP | 67 | 63 | 78 | 82 | 77 | 77 | 83 | 85 | 85 | 78 | **77.5** |
| | | FP | 23 | 13 | 13 | 18 | 12 | 13 | 11 | 8 | 22 | 14 | **14.7** |
| | | FN | 29 | 34 | 21 | 16 | 21 | 22 | 14 | 13 | 14 | 21 | **20.5** |
| | | Precision | 0.74 | 0.82 | 0.85 | 0.82 | 0.86 | 0.85 | 0.88 | 0.91 | 0.79 | 0.84 | **0.83** |
| | | Recall | 0.69 | 0.64 | 0.78 | 0.83 | 0.78 | 0.77 | 0.85 | 0.86 | 0.85 | 0.78 | **0.78** |
| | | F1 | 0.71 | 0.71 | 0.81 | 0.82 | 0.81 | 0.8 | 0.86 | 0.88 | 0.81 | 0.8 | **0.8** |
| | FGSM | DR | | 45.7 | 63.4 | 74.0 | 53.7 | 56.9 | 68.8 | 58.7 | 63.0 | 61.7 | **60.7** |
| | | TP | | 38 | 52 | 60 | 43 | 49 | 53 | 47 | 58 | 55 | **50.6** |
| | | FP | | 11 | 13 | 16 | 9 | 12 | 10 | 8 | 17 | 11 | **11.9** |
| | | FN | | 45 | 30 | 21 | 37 | 37 | 24 | 33 | 34 | 34 | **32.8** |
| | | Precision | | 0.77 | 0.80 | 0.78 | 0.82 | 0.80 | 0.84 | 0.85 | 0.77 | 0.83 | **0.8** |
| | | Recall | | 0.45 | 0.63 | 0.74 | 0.53 | 0.56 | 0.68 | 0.58 | 0.63 | 0.61 | **0.6** |
| | | F1 | | 0.56 | 0.7 | 0.75 | 0.64 | 0.65 | 0.75 | 0.68 | 0.69 | 0.7 | **0.68** |
| | BIM | DR | 40.8 | 40.6 | 64.5 | 75.0 | 66.6 | 56.1 | 74.7 | 62.1 | 68.4 | 44.6 | **59.3** |
| | | TP | 38 | 37 | 62 | 72 | 62 | 55 | 71 | 59 | 65 | 42 | **56.3** |
| | | FP | 20 | 11 | 13 | 16 | 9 | 12 | 10 | 8 | 16 | 11 | **12.6** |
| | | FN | 55 | 54 | 34 | 24 | 31 | 43 | 25 | 36 | 30 | 52 | **38.4** |
| | | Precision | 0.65 | 0.77 | 0.82 | 0.81 | 0.87 | 0.82 | 0.87 | 0.88 | 0.80 | 0.79 | **0.8** |
| | | Recall | 0.40 | 0.40 | 0.64 | 0.75 | 0.66 | 0.56 | 0.73 | 0.62 | 0.68 | 0.44 | **0.58** |
| | | F1 | 0.49 | 0.52 | 0.71 | 0.77 | 0.75 | 0.66 | 0.79 | 0.72 | 0.73 | 0.56 | **0.67** |
| 0.91 | PGD Inf | DR | 40.8 | 39.5 | 64.5 | 75.0 | 66.6 | 56.1 | 74.7 | 62.1 | 68.4 | 44.6 | **59.2** |
| | | TP | 38 | 36 | 62 | 72 | 62 | 55 | 71 | 59 | 65 | 42 | **56.2** |
| | | FP | 20 | 13 | 13 | 16 | 9 | 12 | 10 | 8 | 16 | 11 | **12.8** |
| | | FN | 55 | 55 | 34 | 24 | 31 | 43 | 24 | 36 | 30 | 52 | **38.4** |
| | | Precision | 0.65 | 0.73 | 0.82 | 0.81 | 0.87 | 0.82 | 0.87 | 0.88 | 0.80 | 0.79 | **0.8** |
| | | Recall | 0.40 | 0.39 | 0.64 | 0.75 | 0.66 | 0.56 | 0.74 | 0.62 | 0.68 | 0.44 | **0.58** |
| | | F1 | 0.49 | 0.5 | 0.71 | 0.77 | 0.75 | 0.66 | 0.79 | 0.72 | 0.73 | 0.56 | **0.66** |
| | PGD L2 | DR | 43.0 | 30.7 | 52.5 | 68.6 | 59.3 | 51.5 | 66.3 | 53.6 | 51.0 | 37.8 | **51.4** |
| | | TP | 40 | 28 | 51 | 68 | 57 | 51 | 65 | 52 | 49 | 36 | **49.7** |
| | | FP | 20 | 11 | 12 | 16 | 9 | 12 | 10 | 8 | 17 | 12 | **12.7** |
| | | FN | 53 | 63 | 46 | 31 | 39 | 48 | 33 | 45 | 47 | 59 | **46.4** |
| | | Precision | 0.66 | 0.71 | 0.80 | 0.80 | 0.86 | 0.80 | 0.86 | 0.86 | 0.74 | 0.75 | **0.78** |
| | | Recall | 0.43 | 0.30 | 0.52 | 0.68 | 0.59 | 0.51 | 0.66 | 0.53 | 0.51 | 0.37 | **0.51** |
| | | F1 | 0.52 | 0.42 | 0.63 | 0.73 | 0.69 | 0.62 | 0.74 | 0.65 | 0.6 | 0.49 | **0.6** |
| | CW Inf | DR | 61.7 | 58.9 | 76.5 | 77.7 | 73.4 | 67.0 | 77.3 | 74.7 | 64.5 | 65.9 | **69.7** |
| | | TP | 58 | 56 | 75 | 77 | 72 | 67 | 75 | 74 | 60 | 62 | **67.6** |
| | | FP | 20 | 11 | 13 | 16 | 10 | 12 | 9 | 8 | 14 | 11 | **12.4** |
| | | FN | 36 | 39 | 23 | 22 | 26 | 33 | 22 | 25 | 33 | 32 | **29.1** |
| | | Precision | 0.74 | 0.83 | 0.85 | 0.82 | 0.87 | 0.84 | 0.89 | 0.90 | 0.81 | 0.84 | **0.83** |
| | | Recall | 0.61 | 0.58 | 0.76 | 0.77 | 0.73 | 0.67 | 0.77 | 0.74 | 0.64 | 0.65 | **0.69** |
| | | F1 | 0.66 | 0.68 | 0.8 | 0.79 | 0.79 | 0.74 | 0.82 | 0.81 | 0.71 | 0.73 | **0.75** |
| | DeepFool | DR | 76.5 | 78.3 | 81.5 | 88.6 | 82.9 | 83.0 | 82.9 | 83.5 | 81.2 | 82.9 | **82.1** |
| | | TP | 72 | 76 | 75 | 86 | 78 | 83 | 78 | 81 | 78 | 78 | **78.5** |
| | | FP | 22 | 12 | 12 | 17 | 12 | 12 | 9 | 8 | 20 | 12 | **13.6** |
| | | FN | 22 | 21 | 17 | 11 | 16 | 17 | 16 | 16 | 18 | 16 | **17** |
| | | Precision | 0.76 | 0.86 | 0.86 | 0.83 | 0.86 | 0.87 | 0.89 | 0.91 | 0.79 | 0.86 | **0.84** |
| | | Recall | 0.76 | 0.78 | 0.81 | 0.88 | 0.82 | 0.83 | 0.82 | 0.83 | 0.81 | 0.82 | **0.81** |
| | | F1 | 0.76 | 0.81 | 0.83 | 0.85 | 0.83 | 0.84 | 0.85 | 0.86 | 0.79 | 0.83 | **0.82** |

**Table 13.** Average for all indicators (worst case for F1) per CNN over all 4 targeted attacks.

| $R_{th}$ | | $\mathcal{C}_1$ | $\mathcal{C}_2$ | $\mathcal{C}_3$ | $\mathcal{C}_4$ | $\mathcal{C}_5$ | $\mathcal{C}_6$ | $\mathcal{C}_7$ | $\mathcal{C}_8$ | $\mathcal{C}_9$ | $\mathcal{C}_{10}$ | Avg |
|---|---|---|---|---|---|---|---|---|---|---|---|---|
| | | | | | | Targeted Attacks | | | | | | |
| | DR | 100 | 100 | 100 | 100 | 100 | 100 | 100 | 100 | 100 | 100 | 100 |
| | TP | 68.3 | 63.5 | 74.0 | 70.0 | 65.0 | 72.8 | 82.3 | 68.0 | 92.0 | 88.0 | 74.4 |
| | FP | 26.5 | 26.5 | 22.0 | 25.3 | 25.0 | 21.8 | 20.3 | 19.8 | 36.8 | 33.8 | 25.8 |
| 0.51 | FN | 0 | 0 | 0 | 0 | 0 | 0 | 0 | 0 | 0 | 0 | 0 |
| | Precision | 0.71 | 0.68 | 0.76 | 0.73 | 0.71 | 0.76 | 0.80 | 0.77 | 0.71 | 0.72 | 0.73 |
| | Recall | 1 | 1 | 1 | 1 | 1 | 1 | 1 | 1 | 1 | 1 | 1 |
| | F1 | 0.82 | 0.81 | 0.86 | 0.84 | 0.82 | 0.86 | 0.89 | 0.87 | 0.83 | 0.83 | 0.84 |
| | $\text{F1}_{Worst}$ | 0.78 | 0.76 | 0.85 | 0.83 | 0.80 | 0.85 | 0.88 | 0.86 | 0.83 | 0.83 | 0.82 |
| | DR | 100 | 100 | 100 | 98.73 | 98.6 | 98.55 | 99.03 | 99.7 | 100 | 98.83 | 99.34 |
| | TP | 68.25 | 63.5 | 74 | 69.25 | 64.25 | 71.75 | 81.5 | 67.75 | 92 | 87 | 73.93 |
| | FP | 18.5 | 10 | 10.25 | 16.25 | 10.5 | 11 | 9.5 | 7.5 | 22.25 | 14.5 | 13.03 |
| 0.91 | FN | 0 | 0 | 0 | 0.75 | 0.75 | 1 | 0.75 | 0.25 | 0 | 1 | 0.45 |
| | Precision | 0.77 | 0.84 | 0.87 | 0.8 | 0.84 | 0.86 | 0.89 | 0.89 | 0.79 | 0.85 | 0.84 |
| | Recall | 1 | 1 | 1 | 0.99 | 0.98 | 0.99 | 0.99 | 0.99 | 1 | 0.99 | 0.99 |
| | F1 | 0.87 | 0.91 | 0.93 | 0.88 | 0.91 | 0.92 | 0.93 | 0.94 | 0.88 | 0.91 | 0.90 |
| | $\text{F1}_{Worst}$ | 0.83 | 0.88 | 0.92 | 0.86 | 0.88 | 0.91 | 0.92 | 0.93 | 0.88 | 0.9 | 0.89 |

**Table 14.** Average for all indicators (worst case for F1) per CNN over all 7 untargeted attacks.

| $R_{th}$ | | $\mathcal{C}_1$ | $\mathcal{C}_2$ | $\mathcal{C}_3$ | $\mathcal{C}_4$ | $\mathcal{C}_5$ | $\mathcal{C}_6$ | $\mathcal{C}_7$ | $\mathcal{C}_8$ | $\mathcal{C}_9$ | $\mathcal{C}_{10}$ | Avg |
|---|---|---|---|---|---|---|---|---|---|---|---|---|
| | | | | | | Untargeted Attacks | | | | | | |
| | DR | 77.5 | 79.7 | 89.4 | 94.2 | 89.7 | 87.7 | 91.1 | 94.1 | 87.0 | 86.4 | 87.7 |
| | TP | 72.8 | 73.7 | 84.3 | 89.7 | 83.7 | 85.3 | 85.1 | 89.0 | 82.9 | 81.4 | 82.8 |
| | FP | 35.0 | 33.7 | 27.9 | 32.3 | 29.6 | 24.4 | 22.3 | 24.3 | 34.4 | 31.7 | 29.6 |
| 0.51 | FN | 21.0 | 18.4 | 10.0 | 5.4 | 9.4 | 11.9 | 8.1 | 5.4 | 12.3 | 12.7 | 11.5 |
| | Precision | 0.7 | 0.7 | 0.7 | 0.7 | 0.7 | 0.8 | 0.8 | 0.8 | 0.7 | 0.7 | 0.7 |
| | Recall | 0.8 | 0.8 | 0.9 | 0.9 | 0.9 | 0.9 | 0.9 | 0.9 | 0.9 | 0.9 | 0.9 |
| | F1 | 0.7 | 0.7 | 0.8 | 0.8 | 0.8 | 0.8 | 0.8 | 0.8 | 0.8 | 0.8 | 0.8 |
| | $\text{F1}_{Worst}$ | 0.64 | 0.62 | 0.76 | 0.81 | 0.77 | 0.76 | 0.82 | 0.82 | 0.72 | 0.70 | 0.7 |
| | DR | 55.42 | 51.23 | 68.8 | 77.5 | 68.71 | 64.04 | 75.74 | 68.77 | 68.9 | 59.46 | 65.86 |
| | TP | 52.17 | 47.71 | 65 | 73.86 | 64.43 | 62.43 | 70.86 | 65.29 | 65.71 | 56.14 | 62.36 |
| | FP | 20.83 | 11.71 | 12.71 | 16.43 | 10 | 12.14 | 9.857 | 8 | 17.43 | 11.71 | 13.08 |
| 0.91 | FN | 41.67 | 44.43 | 29.29 | 21.29 | 28.71 | 34.71 | 22.57 | 29.14 | 29.43 | 38 | 31.92 |
| | Precision | 0.70 | 0.78 | 0.83 | 0.81 | 0.86 | 0.83 | 0.87 | 0.88 | 0.79 | 0.81 | 0.82 |
| | Recall | 0.55 | 0.51 | 0.68 | 0.77 | 0.68 | 0.64 | 0.75 | 0.68 | 0.69 | 0.59 | 0.65 |
| | F1 | 0.61 | 0.60 | 0.74 | 0.78 | 0.75 | 0.71 | 0.80 | 0.76 | 0.72 | 0.67 | 0.71 |
| | $\text{F1}_{Worst}$ | 0.49 | 0.42 | 0.63 | 0.73 | 0.69 | 0.62 | 0.74 | 0.65 | 0.6 | 0.49 | 0.61 |

**Table 15.** Average for all indicators (worst case for F1) per CNN over all attacks.

| $R_{th}$ | | $\mathcal{C}_1$ | $\mathcal{C}_2$ | $\mathcal{C}_3$ | $\mathcal{C}_4$ | $\mathcal{C}_5$ | $\mathcal{C}_6$ | $\mathcal{C}_7$ | $\mathcal{C}_8$ | $\mathcal{C}_9$ | $\mathcal{C}_{10}$ | Avg |
|---|---|---|---|---|---|---|---|---|---|---|---|---|
| | | | | | | All Attacks | | | | | | |
| | DR | 88.73 | 89.86 | 94.68 | 97.11 | 94.86 | 93.86 | 95.55 | 97.07 | 93.50 | 93.21 | 93.84 |
| | TP | 70.54 | 68.61 | 79.14 | 79.86 | 74.36 | 79.02 | 83.70 | 78.50 | 87.43 | 84.71 | 78.59 |
| | FP | 30.75 | 30.11 | 24.93 | 28.77 | 27.29 | 23.09 | 21.27 | 22.02 | 35.59 | 32.73 | 27.65 |
| 0.51 | FN | 10.50 | 9.21 | 5.00 | 2.71 | 4.71 | 5.93 | 4.07 | 2.71 | 6.14 | 6.36 | 5.74 |
| | Precision | 0.69 | 0.68 | 0.75 | 0.73 | 0.72 | 0.77 | 0.79 | 0.77 | 0.71 | 0.72 | 0.73 |
| | Recall | 0.88 | 0.90 | 0.94 | 0.97 | 0.95 | 0.94 | 0.95 | 0.97 | 0.93 | 0.93 | 0.94 |
| | F1 | 0.76 | 0.76 | 0.83 | 0.83 | 0.81 | 0.84 | 0.86 | 0.86 | 0.80 | 0.80 | 0.82 |
| | $\text{F1}_{Worst}$ | 0.71 | 0.69 | 0.81 | 0.82 | 0.79 | 0.81 | 0.85 | 0.84 | 0.78 | 0.77 | 0.78 |

**Table 15.** *Cont.*

| $R_{th}$ | | $\mathcal{C}_1$ | $\mathcal{C}_2$ | $\mathcal{C}_3$ | $\mathcal{C}_4$ | $\mathcal{C}_5$ | $\mathcal{C}_6$ | $\mathcal{C}_7$ | $\mathcal{C}_8$ | $\mathcal{C}_9$ | $\mathcal{C}_{10}$ | Avg |
|---|---|---|---|---|---|---|---|---|---|---|---|---|
| | | | | | | All Attacks | | | | | | |
| | DR | 77.71 | 75.61 | 84.40 | 88.11 | 83.66 | 81.30 | 87.38 | 84.24 | 84.45 | 79.14 | 82.60 |
| | TP | 60.21 | 55.61 | 69.50 | 71.55 | 64.34 | 67.09 | 76.18 | 66.52 | 78.86 | 71.57 | 68.14 |
| | FP | 19.67 | 10.86 | 11.48 | 16.34 | 10.25 | 11.57 | 9.68 | 7.75 | 19.84 | 13.11 | 13.05 |
| 0.91 | FN | 20.83 | 22.21 | 14.64 | 11.02 | 14.73 | 17.86 | 11.66 | 14.70 | 14.71 | 19.50 | 16.19 |
| | Precision | 0.74 | 0.81 | 0.85 | 0.81 | 0.85 | 0.84 | 0.88 | 0.89 | 0.79 | 0.83 | 0.83 |
| | Recall | 0.77 | 0.75 | 0.84 | 0.88 | 0.83 | 0.81 | 0.87 | 0.84 | 0.84 | 0.79 | 0.82 |
| | F2 | 0.74 | 0.75 | 0.83 | 0.83 | 0.83 | 0.81 | 0.87 | 0.85 | 0.80 | 0.79 | 0.81 |
| | F1$_{Worst}$ | 0.66 | 0.65 | 0.78 | 0.80 | 0.79 | 0.77 | 0.83 | 0.79 | 0.74 | 0.70 | 0.75 |

**Table 16.** For $R_{th}$ = 0.51 and 0.91, the optimal number of permutations $t_{optimal,\mathcal{C},atk}$ per CNN and attack, and the optimal number of permutations $t_{optimal,\mathcal{C}}$ per CNN valid for all tested attacks (potentially relevant to assess unknown attacks).

| $R_{th}$ | Scenario | Attacks | $\mathcal{C}_1$ | $\mathcal{C}_2$ | $\mathcal{C}_3$ | $\mathcal{C}_4$ | $\mathcal{C}_5$ | $\mathcal{C}_6$ | $\mathcal{C}_7$ | $\mathcal{C}_8$ | $\mathcal{C}_9$ | $\mathcal{C}_{10}$ |
|---|---|---|---|---|---|---|---|---|---|---|---|---|
| | | | Optimal Number $t_{optimal,\mathcal{C},atk}$ of Permutations per CNN and Attack | | | | | | | | | |
| 0.51 | Untargeted | EA | 3 | 3 | 3 | 3 | 3 | 3 | 3 | 3 | 3 | 3 |
| | | FGSM | | 3 | 19 | 13 | 3 | 9 | 9 | 7 | 13 | 5 |
| | | BIM | 3 | 3 | 11 | 5 | 11 | 19 | 3 | 17 | 13 | 5 |
| | | PGD Inf | 3 | 3 | 11 | 5 | 11 | 19 | 3 | 17 | 13 | 5 |
| | | PGD L2 | 3 | 7 | 3 | 7 | 5 | 15 | 66 | 27 | 7 | 9 |
| | | CW Inf | 3 | 5 | 25 | 17 | 5 | 3 | 5 | 23 | 7 | 3 |
| | | DeepFool | 5 | 3 | 7 | 3 | 37 | 9 | 68 | 3 | 3 | 3 |
| | Targeted | EA | 12 | 12 | 12 | 12 | 12 | 12 | 12 | 12 | 12 | 12 |
| | | BIM | 12 | 12 | 12 | 12 | 12 | 12 | 12 | 12 | 12 | 12 |
| | | PGD Inf | 12 | 12 | 12 | 12 | 12 | 12 | 12 | 12 | 12 | 12 |
| | | PGD L2 | 12 | 12 | 12 | 12 | 12 | 12 | 12 | 12 | 12 | 12 |
| | | $t_{optimal,\mathcal{C}}$ per CNN | 12 | 12 | 25 | 17 | 37 | 19 | 68 | 27 | 13 | 12 |
| 0.91 | Untargeted | EA | 12 | 12 | 12 | 12 | 12 | 12 | 12 | 12 | 12 | 12 |
| | | FGSM | | 12 | 12 | 12 | 12 | 12 | 12 | 12 | 12 | 12 |
| | | BIM | 12 | 12 | 12 | 12 | 12 | 12 | 12 | 12 | 100 | 12 |
| | | PGD Inf | 12 | 12 | 12 | 12 | 12 | 12 | 12 | 12 | 100 | 12 |
| | | PGD L2 | 12 | 12 | 12 | 12 | 12 | 12 | 12 | 12 | 12 | 12 |
| | | CW Inf | 12 | 12 | 23 | 12 | 34 | 12 | 12 | 12 | 12 | 12 |
| | | DeepFool | 12 | 34 | 12 | 12 | 12 | 12 | 12 | 12 | 12 | 12 |
| | Targeted | EA | 12 | 12 | 12 | 12 | 12 | 12 | 12 | 12 | 12 | 12 |
| | | BIM | 12 | 12 | 12 | 12 | 12 | 12 | 12 | 12 | 12 | 12 |
| | | PGD Inf | 12 | 12 | 12 | 12 | 12 | 12 | 12 | 12 | 12 | 12 |
| | | PGD L2 | 12 | 67 | 12 | 12 | 12 | 12 | 12 | 12 | 12 | 12 |
| | | $t_{optimal,\mathcal{C}}$ per CNN | 12 | 67 | 23 | 12 | 34 | 12 | 12 | 12 | 100 | 12 |

## 10. Performance Comparison of ShuffleDetect and Feature Squeezer (FS)

To assess the extrinsic performance of ShuffleDetect, we compared it with the FS detector [13]. We selected this detector since, similar to ShuffleDetect, it is an unsupervised detector, which also presents no significant complexity issues. The comparison between ShuffleDetect and FS is performed only according to the detection rate, and not according to the other indicators mentioned in Section 3, since, for instance, the value of FPR in [13] is determined by the behavior of FS as compared to another detector (MagNet [35]); hence, it is not an intrinsic value, to the difference of what we do in Section 9 for ShuffleDetect. Therefore, the comparisons of the detectors are performed on the 9480 images of Table 3, adversarial against the 10 considered CNNs (Section 7).

In our experiments, we used multiple squeezers for FS as suggested in [13] (we keep their notations in what follows). The $L_1$ norm is used to measure the difference between the prediction by the CNN of the input image and the prediction of the squeezed input image:

$$score^{x,x_{squeezed}} = \|g(x) - g(x_{squeezed})\|_{L_1},\tag{12}$$

where $x$ is the input image and $g(x)$ is the classification vector of the CNN according to the different categories. Multiple feature squeezers are combined in the FS detector. In practice, one computes the maximum distance:

$$score^{joint}(x) = max(score^{x,x_{sq1}}, score^{x,x_{sq2}}, score^{x,x_{sq3}})\tag{13}$$

The values of the parameters of the FS squeezers are chosen as the optimal values recommended in [13]:

- Color depth reduction: the image color depth is decreased to 5 bits.
- Median smoothing: the filter size is set to $2 \times 2$.
- Non-local means: the search window size is set to $11 \times 11$, the patch size is set to $3 \times 3$, and the filter strength is set to 4.
- The threshold is set to 1.2128.

The image is declared by the FS detector as adversarial if $score^{joint}(x) \geq 1.2128$ and is declared clean otherwise.

For ShuffleDetect$^{C,R_{th},t}$, consistently with the outcomes of Section 9, we set $t = 100$, $R_{th} = 0.51$ for all CNNs in the experiments (note that the size $s \times s$ of the patches is kept to $56 \times 56$ for the images considered here).

Table 17 compares the detection rates of ShuffleDetect and FS for the 9480 adversarial images referred to. For the 2975 adversarial images for the targeted scenario, both detectors demonstrate high success rates. Even if FS achieves DR over 92%, it is outperformed by ShuffleDetect, which achieves 100% in all cases. For the 6505 adversarial images for the untargeted scenario, the success rates of both detectors experience a decline. FS achieves slightly better results than ShuffleDetect for DeepFool and CW Inf, and significantly better results for PGD Inf, BIM, PGD, and L2; it is outperformed by ShuffleDetect, slightly for FGSM, and highly significant for EA. Regarding the overall performance (see the last row of Table 17), ShuffleDetect achieves a higher success rate than FS on average (both scenarios and all CNNs considered).

**Table 17.** Performance comparison of ShuffleDetect and FS regarding detection rates.

| Scenario | Attacks | Detectors | $C_1$ | $C_2$ | $C_3$ | $C_4$ | $C_5$ | $C_6$ | $C_7$ | $C_8$ | $C_9$ | $C_{10}$ | AVG |
|---|---|---|---|---|---|---|---|---|---|---|---|---|---|
| Targeted | EA | ShuffleDetect | 100 | 100 | 100 | 100 | 100 | 100 | 100 | 100 | 100 | 100 | 100.0 |
| | | FS | 89.0 | 100.0 | 90.9 | 94.0 | 91.1 | 90.6 | 89.9 | 91.9 | 93.8 | 95.9 | 92.7 |
| | BIM | ShuffleDetect | 100 | 100 | 100 | 100 | 100 | 100 | 100 | 100 | 100 | 100 | 100.0 |
| | | FS | 100.0 | 94.7 | 98.2 | 100.0 | 97.8 | 98.2 | 97.3 | 98.0 | 96.6 | 97.4 | 97.8 |
| | PGD Inf | ShuffleDetect | 100 | 100 | 100 | 100 | 100 | 100 | 100 | 100 | 100 | 100 | 100.0 |
| | | FS | 98.0 | 94.7 | 98.2 | 100.0 | 97.8 | 98.2 | 97.3 | 98.0 | 96.6 | 97.4 | 97.6 |
| | PGD L2 | ShuffleDetect | 100 | 100 | 100 | 100 | 100 | 100 | 100 | 100 | 100 | 100 | 100.0 |
| | | FS | 100.0 | 100.0 | 100.0 | 100.0 | 97.8 | 100.0 | 100.0 | 100.0 | 100.0 | 100.0 | 99.8 |
| Untargeted | EA | ShuffleDetect | 88.5 | 90.7 | 93.9 | 96.9 | 96.9 | 95.9 | 96.9 | 97.9 | 93.9 | 93.9 | 94.5 |
| | | FS | 53.1 | 43.3 | 45.5 | 39.8 | 40 | 33.3 | 33 | 35.7 | 47.5 | 35.4 | 40.6 |
| | FGSM | ShuffleDetect | | 80.7 | 89.0 | 92.5 | 86.2 | 87.2 | 85.7 | 92.5 | 85.8 | 86.5 | 87.3 |
| | | FS | | 81.9 | 86.6 | 86.4 | 85 | 76.7 | 83.1 | 77.5 | 88 | 85.4 | 83.4 |
| | BIM | ShuffleDetect | 66.6 | 72.5 | 86.4 | 93.7 | 86.0 | 83.6 | 88.4 | 92.6 | 84.2 | 84.0 | 83.8 |
| | | FS | 96.8 | 90.1 | 94.8 | 96.9 | 98 | 90.8 | 96.8 | 94.7 | 94.7 | 93.6 | 94.7 |
| | PGD Inf | ShuffleDetect | 66.6 | 72.5 | 86.4 | 93.7 | 86.0 | 83.6 | 88.4 | 92.6 | 84.2 | 84.0 | 83.8 |
| | | FS | 96.8 | 90.1 | 94.8 | 96.9 | 98 | 90.8 | 96.8 | 94.7 | 94.7 | 93.6 | 94.7 |

**Table 17.** *Cont.*

| Scenario | Attacks | Detectors | $c_1$ | $c_2$ | $c_3$ | $c_4$ | $c_5$ | $c_6$ | $c_7$ | $c_8$ | $c_9$ | $c_{10}$ | AVG |
|---|---|---|---|---|---|---|---|---|---|---|---|---|---|
| Untargeted | PGD L2 | ShuffleDetect | 68.8 | 61.5 | 80.4 | 91.9 | 83.3 | 77.7 | 86.7 | 88.6 | 78.1 | 73.6 | 79.1 |
| | | FS | 97.8 | 90.1 | 94.8 | 97 | 97 | 90.9 | 94.9 | 93.8 | 95.8 | 91.6 | 94.4 |
| | CW Inf | ShuffleDetect | 79.7 | 86.3 | 93.8 | 93.9 | 91.8 | 90.0 | 93.8 | 96.9 | 88.1 | 89.3 | 90.4 |
| | | FS | 93.6 | 92.6 | 92.9 | 91.9 | 92 | 89 | 96.9 | 91.9 | 95.7 | 91.5 | 92.8 |
| | DeepFool | ShuffleDetect | 94.6 | 93.8 | 95.6 | 96.9 | 97.8 | 96.0 | 97.8 | 97.9 | 94.7 | 93.6 | 95.9 |
| | | FS | 98.9 | 96.9 | 97.8 | 96.9 | 95 | 95 | 97.9 | 96.9 | 99 | 100 | 97.4 |
| | **Overall** | **ShuffleDetect** | **86.5** | **87.1** | **93.2** | **96.3** | **93.5** | **92.2** | **94.3** | **96.3** | **91.7** | **91.4** | **92.2** |
| | | **FS** | **92.4** | **88.6** | **90.4** | **90.9** | **89.9** | **86.7** | **89.4** | **88.5** | **91.1** | **89.3** | **89.7** |

## 11. Conclusions

In this paper, we presented ShuffleDetect as a new unsupervised method for the detection of image adversarials against trained CNNs. We provided a complete design and recommendations for the selection of the values of its parameters. Given a CNN and an image potentially resized to fit the CNN's input size, the steps that essentially compose this new detection method are fairly simple. During the initiation phase, the dominant category in which the CNN sorts the input image is required, the image is split into non-overlapping patches (of fixed sizes, depending on the CNN's own input size), and a fixed set of appropriate permutations is selected at random. Then a loop is performed according to the successive permutations, where the patches are shuffled with the running permutation, and the dominant category in which the CNN sorts the shuffled image is compared with the outcome for the unshuffled image, leading to a Boolean value. Finally, one assesses the proportion of permutations for which the CNN classifies the shuffled image into a different category than the unshuffled input image. ShuffleDetect declares the image as adversarial if this proportion exceeds a threshold value $R_{th}$ and declares the image clean otherwise.

Our extensive experiments with 10 diverse and state-of-the-art CNNs, trained on ImageNet with images usually resized to $224 \times 224$, with 8 attacks (one 'black-box' and seven 'white-box'), and with 9500 clean and adversarial images for the targeted or untargeted scenario led us to recommend a size of $56 \times 56$ for the altogether 16 patches, and the "democratic" value $R_{th} = 0.51$. Although running ShuffleDetect with 100 permutations is perfectly feasible and could be considered a safe option, a smaller number of permutations, varying between 12 and 68 according to the considered CNN, may also lead to a satisfactory detection rate. Additionally, if the defender has more information about the type of attack expected, the number of permutations can be fine-tuned accordingly. This said, and since this type of knowledge occurs rarely, we recommend taking at least 100 permutations.

Apart from the time needed for the creation of a fixed set of permutations, namely 0.064 s to obtain 100 permutations, which can be performed once for all (at least in our implementation), the intrinsic performance of ShuffleDetect on our computers shows an inference time latency of $\simeq 0.0784$ s per image per permutation. The algorithm can be easily parallelized so that the required time for a complete run of the algorithm can be significantly less than the $\simeq 7.982$ s/image when this task is not distributed. Out of this time-consuming process, the classification process of an image by the CNN is $\simeq 98.02\%$; therefore, the shuffling process itself requires only $\simeq 1.98\%$. The overhead is very limited since its main part, the "permanent storage", is required essentially only for the 100 permutations (the storage per permutation is a sequence of groups of distinct integers between 1 and 16 in our case), and the dominating category of the unshuffled image. Among the main indicators of a detector, the most relevant ones are the false positive rate and the detection rate. With $R_{th} = 0.51$ and 100 permutations, as well as the average overall considered CNNs and images, ShuffleDetect achieves an average FPR of 32.7%, an average DR of 100% for the adversarial images obtained by targeted attacks, and 87.79% of those obtained by

untargeted attacks. Our study also provides the scores of the other relevant indicators, i.e., TP, FP, FN, precision, recall, and F1.

While performing a thorough comparison with other detectors requires overcoming the difficult challenges outlined in Section 3, in order to make sound comparisons under the same conditions, we performed this task for one detector, namely FS, and showed that, on average, ShuffleDetect achieves better detection rates than FS.

Independent of the outcome of any comparison process with other detectors, our ShuffleDetect method could be used as a first line of defense before applying more sophisticated, time-consuming, and overhead-consuming detection methods than ShuffleDetect.

As a potential area for future research, it would be worthwhile to assess the effectiveness of ShuffleDetect using CNNs trained on Cifar10 and MNIST datasets. This could result in determining the optimal patch size as a ratio to the image size. Additionally, it would be beneficial to explore the optimal patch size for images containing very small or very large objects.

**Author Contributions:** All authors: writing—review and editing. All authors have read and agreed to the published version of the manuscript.

**Funding:** This research received no external funding.

**Institutional Review Board Statement:** Not applicable.

**Informed Consent Statement:** Not applicable.

**Data Availability Statement:** Not applicable.

**Acknowledgments:** The authors thank Uli Sorger for the fruitful discussions on this subject.

**Conflicts of Interest:** The authors declare no conflict of interest.

## Appendix A

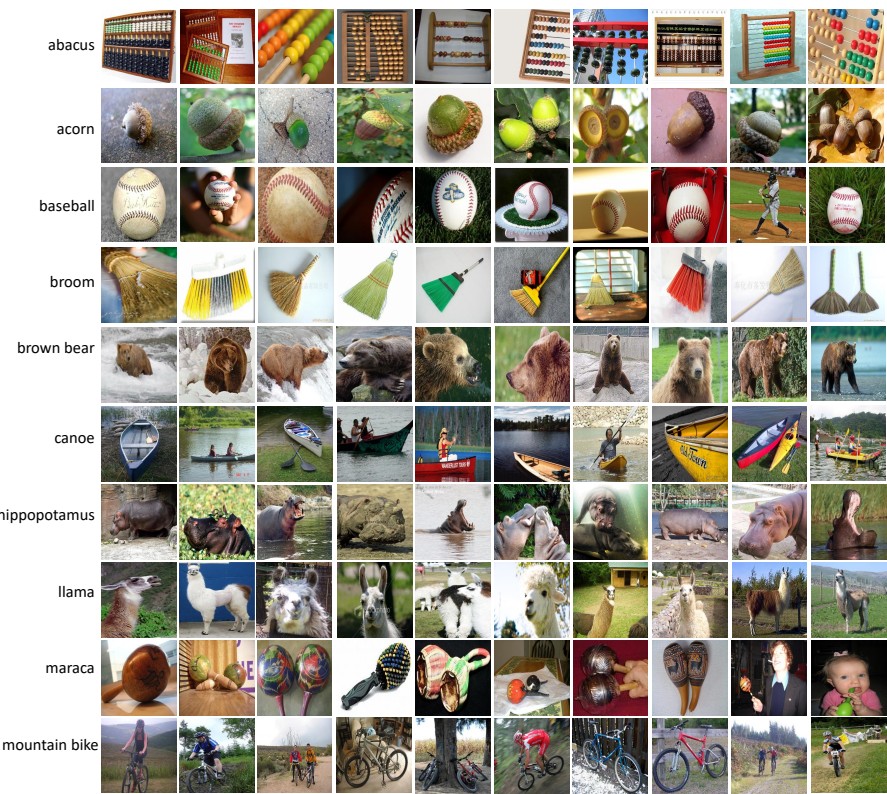

**Figure A1.** The 100 ancestor images $\mathcal{A}_q^p$ used in the experiments. $\mathcal{A}_q^p$ pictured in the $q$th row and $q$th column ($1 \leq p, q \leq 10$) is randomly chosen from the ImageNet validation set of the ancestor category $c_{a_q}$ specified on the left of the $q$th row.

**Table A1.** The original sizes ($h \times w$) of the 100 ancestor images $\mathcal{A}_q^p$ before resizing with the bilinear interpolation function.

| | | | | | | | | | | | |
|---|---|---|---|---|---|---|---|---|---|---|---|
| | | | | | Ancestor Images $\mathcal{A}_q^p$ and Their Original Size ($h \times w$) | | | | | | |
| $c_{a_q}$ | $q$ $\diagdown$ $p$ | 1 | 2 | 3 | 4 | 5 | 6 | 7 | 8 | 9 | 10 |
| abacus | 1 | (206, 250) | (960, 1280) | (262, 275) | (598, 300) | (377, 500) | (501, 344) | (375, 500) | (448, 500) | (500, 500) | (150, 200) |
| acorn | 2 | (374, 500) | (500, 469) | (375, 500) | (500, 375) | (500, 500) | (500, 500) | (375, 500) | (374, 500) | (461, 500) | (333, 500) |
| baseball | 3 | (398, 543) | (240, 239) | (180, 240) | (333, 500) | (262, 350) | (310, 310) | (404, 500) | (344, 500) | (375, 500) | (285, 380) |
| broom | 4 | (113, 160) | (150, 150) | (333, 500) | (500, 333) | (497, 750) | (336, 500) | (188, 250) | (375, 500) | (334, 500) | (419, 640) |
| brown bear | 5 | (500, 333) | (286, 490) | (360, 480) | (298, 298) | (413, 550) | (366, 500) | (400, 400) | (348, 500) | (346, 500) | (640, 480) |
| canoe | 6 | (500, 332) | (450, 600) | (500, 375) | (375, 500) | (406, 613) | (600, 400) | (1067, 1600) | (333, 500) | (1536, 2048) | (375, 500) |
| hippopotamus | 7 | (375, 500) | (1200, 1600) | (333, 500) | (450, 291) | (525, 525) | (375, 500) | (500, 457) | (424, 475) | (500, 449) | (339, 500) |
| llama | 8 | (500, 333) | (618, 468) | (500, 447) | (253, 380) | (500, 333) | (333, 500) | (375, 500) | (375, 500) | (290, 345) | (375, 500) |
| maraca | 9 | (375, 500) | (375, 500) | (470, 627) | (151, 220) | (250, 510) | (375, 500) | (99, 104) | (375, 500) | (375, 500) | (500, 375) |
| mountain bike | 10 | (375, 500) | (500, 375) | (375, 500) | (333, 500) | (500, 375) | (300, 402) | (375, 500) | (446, 500) | (375, 500) | (500, 333) |

## Appendix B

**Table A2.** For a $224 \times 224$ image, grid of its 16 patches of size $56 \times 56$, represented as $P_{i,j}$ (left grid), and as $P_1, \cdots, P_{16}$ (right grid) as used by the permutations $\sigma_k$.

| Left Grid | | | | Right Grid | | | |
|---|---|---|---|---|---|---|---|
| $P_{1,1}$ | $P_{1,2}$ | $P_{1,3}$ | $P_{1,4}$ | $P_1$ | $P_2$ | $P_3$ | $P_4$ |
| $P_{2,1}$ | $P_{2,2}$ | $P_{2,3}$ | $P_{2,4}$ | $P_5$ | $P_6$ | $P_7$ | $P_8$ |
| $P_{3,1}$ | $P_{3,2}$ | $P_{3,3}$ | $P_{3,4}$ | $P_9$ | $P_{10}$ | $P_{11}$ | $P_{12}$ |
| $P_{4,1}$ | $P_{4,2}$ | $P_{4,3}$ | $P_{4,4}$ | $P_{13}$ | $P_{14}$ | $P_{15}$ | $P_{16}$ |

**Table A3.** For $t$ up to 100 rounds, the list of random permutations $\sigma_r$ for $1 \le r \le 100$. Each $\sigma_r$ is represented as the product of cycles operating on 16 patches of a $224 \times 224$ image.

| | | | |
|---|---|---|---|
| | $t = 100$ | | |
| Round $r$ | Permutation $\sigma_r$ | Round $r$ | Permutation $\sigma_r$ |
| 1 | (1,13,4,6)(2,14,10)(3,11,12,5,9)(7,16,15,8) | 51 | (1,9,10,8,13,6,2,15,5,14,4,7,11)(3,16) |
| 2 | (1,8,9,2,6,11,15,12)(3,4,5,10,7)(14,16) | 52 | (1,11,15,4,10,2,3,5,12,9,13,8,16,7)(6,14) |
| 3 | (1,2,10,12)(3,11,16,15)(4,6,13,7,14,9,5) | 53 | (1,12,3,7,2,5,6,15,16,14,4,10)(8,13)(9,11) |
| 4 | (1,12,7,6,9,5,13,16)(2,4,14,10)(3,11,15,8) | 54 | (1,2,13,12,7)(3,6,4,8)(9,10)(11,15) |
| 5 | (3,5,14,16,7,4,12,6,13,11)(8,15,9,10) | 55 | (1,6,3)(2,12,14,4,15,7)(5,9)(8,13,10,11,16) |
| 6 | (1,7,4,9,2,5)(3,14)(6,16,8,13,10,15,12,11) | 56 | (1,8,3,4,13,10,9,16,5,2,7,11,12)(6,15) |
| 7 | (1,7,15,5,10,4,2,13,14,12,6,9)(3,11,16,8) | 57 | (1,5,12,9,15,4,7,11,2,10,6,16,8,3,14) |
| 8 | (1,2,5)(3,11,16,10,12,9,7,6,15,4,13,8) | 58 | (1,12)(2,6,13,10,7,8)(3,15,5,16,11,9)(4,14) |
| 9 | (1,7,15,8,13,5,9,11)(2,12)(3,16,14,4)(6,10) | 59 | (2,11,13,6)(3,12,10,7,16,4)(5,8)(9,15,14) |
| 10 | (1,16,8,15,4,5,6)(3,14,13)(7,12)(9,10,11) | 60 | (1,13,15,8,4,14,5,9,12,7,10,11,16,3,6,2) |
| 11 | (1,8,10,13,9,6,2)(3,12,5,15,14,4,7)(11,16) | 61 | (1,2,14,6,10,7)(4,5,12,9,8,16,11) |
| 12 | (1,4,14,16,5,6,11,13,15,9)(2,12,10,3,8) | 62 | (1,11)(2,7,4,5,10,12,14,9)(3,6,8,13,15,16) |
| 13 | (1,5,14,13,10)(2,6,7,4,8)(3,15,11,9,16,12) | 63 | (1,9,14,15,11,5,8,10,2,4,3,12,16,13,6,7) |
| 14 | (1,16,9,4,3,2,5,7,6,11,12,10,8,15,14,13) | 64 | (2,11,12,10,5)(3,16,14,13,4,8,6,15)(7,9) |
| 15 | (1,16,5,13,8,6)(2,15,14,10,11,12,9,3,7,4) | 65 | (1,5,12,3,2,6,11,13,16,14)(7,10,15) |
| 16 | (1,14,12,2,13,7,10,8,3,15,11,6,16,4) | 66 | (1,15,7,11,12,2)(3,10,4,14,5,8,6,16,9,13) |
| 17 | (1,2,5,13)(4,11,8,10,16,14,15)(6,7)(9,12) | 67 | (1,4)(2,6,15,11,12,16)(3,5,14)(7,8)(9,10) |
| 18 | (1,12,13,16,3,8,10,2,11,14,7,4,15,6) | 68 | (1,13,6,14,2,10,5,15,11,9,4,12,8,3,7,16) |
| 19 | (1,8,4,16,3,13,6,7,15)(2,12)(5,14,11)(9,10) | 69 | (1,9,15,6,8,10,11,2,12,16,4,13,14,7)(3,5) |
| 20 | (1,14,15,5)(2,4,12,13)(3,8,16,11)(6,7)(9,10) | 70 | (1,2,6,8,3)(4,12)(5,7,13,10,15)(9,11,14,16) |
| 21 | (1,2,6)(3,8,14,10,13,12)(5,9,16,15) | 71 | (2,10,16,6,13,3,14,12)(4,5,8,15,7,9,11) |

**Table A3.** *Cont.*

| | t = 100 | | |
|---|---|---|---|
| **Round** $r$ | **Permutation** $\sigma_r$ | **Round** $r$ | **Permutation** $\sigma_r$ |
| 22 | (1,3,11,14,2,10)(4,12,6,7,15,5,16,9)(8,13) | 72 | (1,8,13,7)(2,10,15,6,14,9,3,16,5,11) |
| 23 | (1,4,11,9,14,7,2,5,3,8,6)(10,15) | 73 | (1,5,16,12,6,2,8,11,4,10,9,13,14)(7,15) |
| 24 | (1,12,8,7)(2,4,5,14,6,9,3,13,16)(10,15,11) | 74 | (1,6,16,13,11,5,14,4,3,9,15,2,8,10,7) |
| 25 | (1,14,6,4,10,16,5,13,12,2,8,15,9,3,7,11) | 75 | (1,12,9,6,15,4,5,14,2,3)(7,11,16)(8,13) |
| 26 | (1,15,5)(2,13,4,9,16,8,11,12,3,6,10,14,7) | 76 | (1,11,15,16,9)(2,12,5,3,8,13,6)(7,10,14) |
| 27 | (1,10,8,12,14,7,2)(3,13,11,5,6)(4,15) | 77 | (2,8,15,10,16,9,12,7,4)(3,5,11,14)(6,13) |
| 28 | (1,8,11,7,16,5,6,12,4,14)(2,15,3,10,9) | 78 | (1,15,11,8,16,5,2,12,3,13,6,10,14)(4,9) |
| 29 | (1,5,3,12,15,11)(2,14,10,6,8,9,7,13,16) | 79 | (1,16,13,5,3,10,6,4,15,2,11)(7,14,9) |
| 30 | (1,10,8,15)(3,9,7,4,12)(5,11,6) | 80 | (2,10,13,11,15,6,5,8,3,16,4,7,9,14) |
| 31 | (1,3,10,6,9,7,16,2,8)(4,5,14)(12,13) | 81 | (1,5,15,2,16,10,9,14,11,4,12,6,3) |
| 32 | (1,2,11,16,10,15)(3,14,6,5,9)(4,7,13,12) | 82 | (1,13,5,10,2,15,11,4,16,7,12,9,14,3,8,6) |
| 33 | (1,16,14,13,10,7,12,3,6,11,9,5)(2,4) | 83 | (1,14,8,9,15,3,5,2,7,10,4,12,6,11,16) |
| 34 | (1,6,14)(2,10,3,15,9,12,11,4,16,13,8,7) | 84 | (1,15,8,9,4,3,16,6,7,14,5,12,2,10,13,11) |
| 35 | (1,2)(3,15,16,13,12,4,5,6,7,9,10,11)(8,14) | 85 | (1,9,3,13)(4,11,15,12)(5,16,6,10,7,8,14) |
| 36 | (3,12,8,6,7,10,16,5,15,13)(4,9) | 86 | (1,9,15,8,13,14,6,11,7)(2,10,12,3,16,5,4) |
| 37 | (1,3,10,4,15,8,16,12,13,7,14,9,2) | 87 | (1,3,9,7,6,4,5)(10,12,14,16,11)(13,15) |
| 38 | (1,4,16)(2,9,5,13,10,14,3,11,8,7)(12,15) | 88 | (1,9,8,12,14,5,10,6,15,4,3)(2,11,16,13) |
| 39 | (1,4,5,2,11,10,12,9,14,15,3,16,13)(7,8) | 89 | (1,13,2,9,16)(3,14,11,8,7,15,6)(5,12,10) |
| 40 | (1,9,8,15,5,10,11,12,4,14,2,3,13,16,6,7) | 90 | (1,8,16,2,6,3,10,14,7,13,4,9,12,5,11) |
| 41 | (1,8,9,11,16,4)(2,13,14,15,7,12)(3,10,5) | 91 | (1,5,16,6,10,3,11,15,9,12,14,8,7,2,4) |
| 42 | (1,9,4,15,14,5)(2,11,12,3,6,10,13)(7,8,16) | 92 | (1,10,16,11,4,8,5,12,13,3,14,9)(2,7,15) |
| 43 | (2,8,14,9,7,16,12,10,13,6,15,3,11,4,5) | 93 | (1,4,2,13,6,9,14,3,10,8,16,11,15,7) |
| 44 | (1,11,12,14,2,13,8,9,3,10,6)(5,15,16) | 94 | (1,16,15,3,9,2,6,7,11,4)(5,8,14,12)(10,13) |
| 45 | (1,3,16,4)(2,5,6,15,7,11)(8,9,10)(13,14) | 95 | (3,10,13,15,12,9,14,16,7,5,4,6,8,11) |
| 46 | (1,6,12,10,8,15,5)(2,4,16,3,13)(7,14,9,11) | 96 | (1,6,15,4,5,3,16,13,9,10,12,2,8,7) |
| 47 | (1,7,14,3,4,16,8,13)(2,9)(5,12,6,11)(10,15) | 97 | (1,14,2,7,3,13,8,16,5,11,15,4,6,10,9,12) |
| 48 | (1,8,14,6,11,13,3,10,12,16,2,15,5,7,4) | 98 | (1,13,3,16)(2,11,6,14,5)(4,9,10,7,12,8,15) |
| 49 | (1,8,12,10,11,6,9,15)(3,13,4,7)(5,16,14) | 99 | (1,6,13,5,12,15,2)(3,14,8)(7,11)(9,16,10) |
| 50 | (1,5,16,2,11,4,13,15,12,3,8,7,14,6) | 100 | (1,12,11,8,2,3)(4,14,16,7,10,6)(5,13,15) |

**Table A4.** The duration, in s, of each of the main steps of Algorithm 1 for each CNN.

| | | **Per Permutation** | | | |
|---|---|---|---|---|---|
| | | **Shuffling** | **Predicting** | | |
| $\mathcal{C}$ | **Steps: 8–12** | **Step: 8** | **Steps: 9–10** | **Shuff%** | **Pred%** |
| $C_1$ | 0.0955 | 0.0014 | 0.0941 | 1.483 | 98.511 |
| $C_2$ | 0.1176 | 0.0014 | 0.1162 | 1.228 | 98.767 |
| $C_3$ | 0.0578 | 0.0016 | 0.0563 | 2.727 | 97.264 |
| $C_4$ | 0.0933 | 0.0016 | 0.0917 | 1.664 | 98.330 |
| $C_5$ | 0.1262 | 0.0016 | 0.1246 | 1.233 | 98.763 |
| $C_6$ | 0.0660 | 0.0016 | 0.0644 | 2.449 | 97.542 |
| $C_7$ | 0.0844 | 0.0017 | 0.0828 | 1.975 | 98.018 |
| $C_8$ | 0.1017 | 0.0017 | 0.1000 | 1.678 | 98.316 |
| $C_9$ | 0.0213 | 0.0015 | 0.0198 | 6.930 | 93.047 |
| $C_{10}$ | 0.0196 | 0.0015 | 0.0182 | 7.563 | 92.411 |
| **AVG** | **0.0784** | **0.0015** | **0.077** | **1.978** | **98.015** |

# Appendix C

(**a**) $\mathcal{C}_1$

(**b**) $\mathcal{C}_2$

(**c**) $\mathcal{C}_3$

(**d**) $\mathcal{C}_4$

(**e**) $\mathcal{C}_5$

(**f**) $\mathcal{C}_6$

(**g**) $\mathcal{C}_7$

(**h**) $\mathcal{C}_8$

(**i**) $\mathcal{C}_9$

(**j**) $\mathcal{C}_{10}$

**Figure A2.** Shuffling test results of 100 clean (ancestor) images on $\mathcal{C} = \mathcal{C}$ for $1 \le k \le 10$ over 100 permutations.

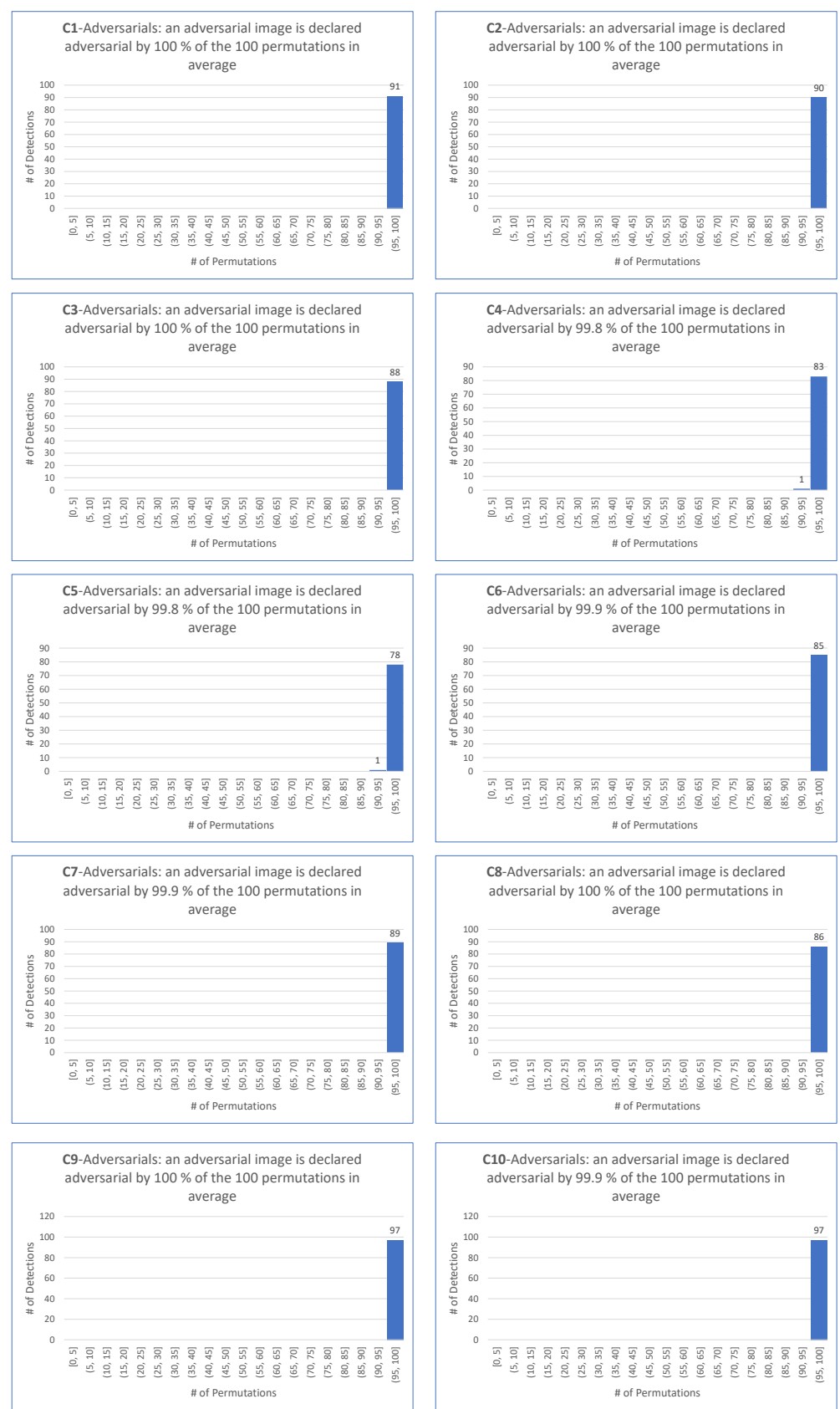

**Figure A3.** ShuffleDetect results for adversarial images generated by the **EA-targeted** attack on $\mathcal{C} = \mathcal{C}$ for $1 \leq k \leq 10$ over 100 permutations.

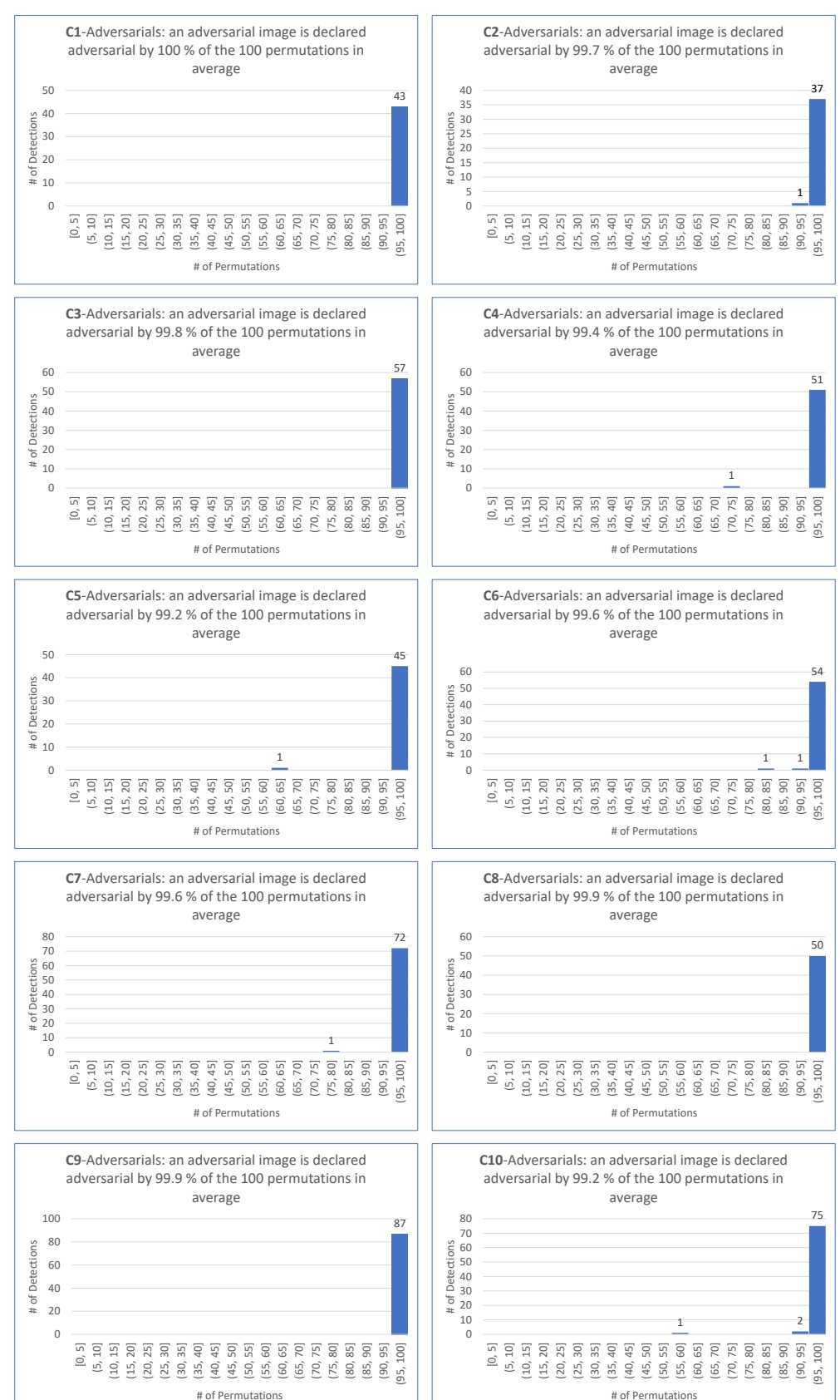

**Figure A4.** ShuffleDetect results for adversarial images generated by the **BIM-targeted** attack on $\mathcal{C} = \mathcal{C}$ for $1 \leq k \leq 10$ over 100 permutations.

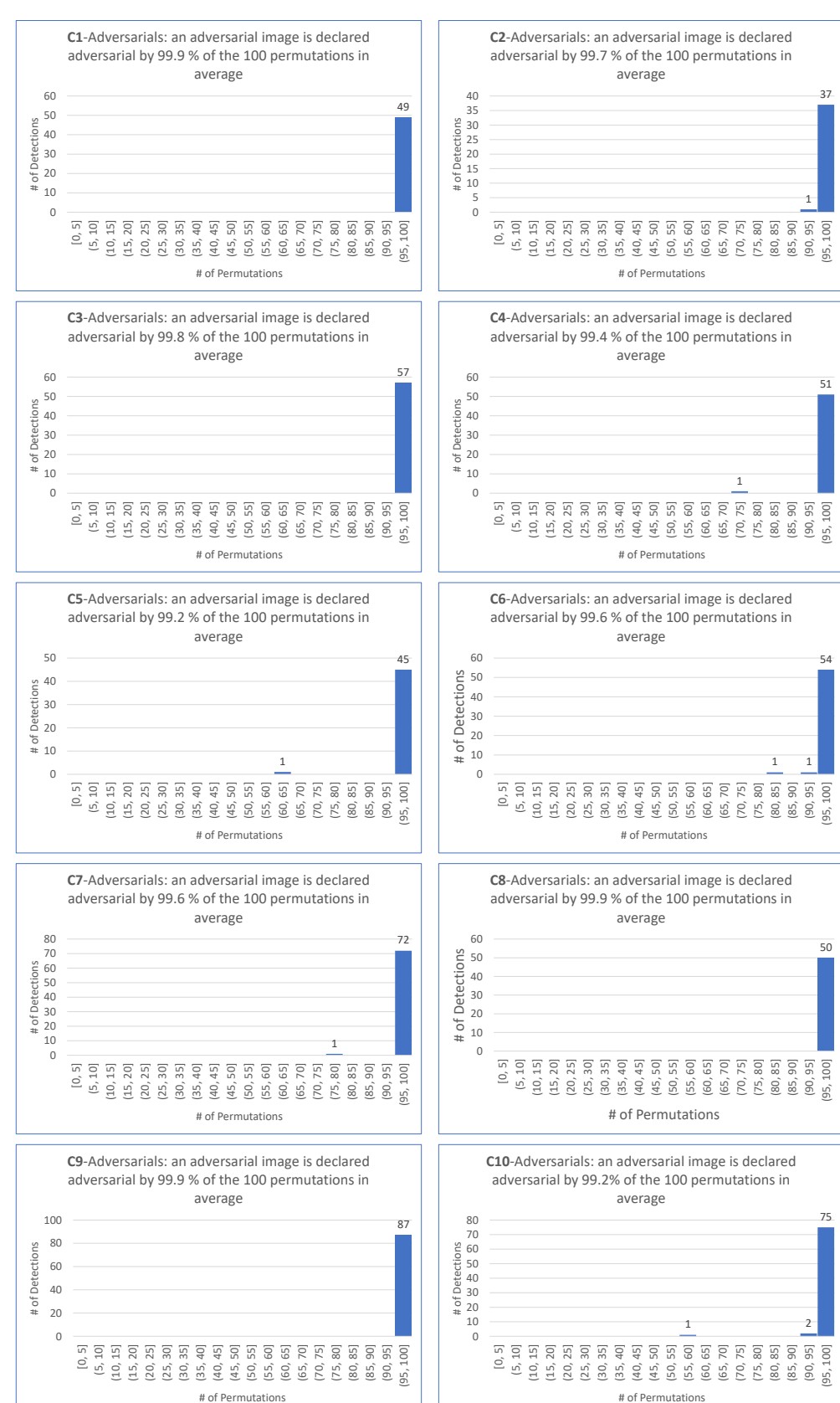

**Figure A5.** ShuffleDetect results for adversarial images generated by the **PGD Inf-targeted** attack on $\mathcal{C} = \mathcal{C}$ for $1 \leq k \leq 10$ over 100 permutations.

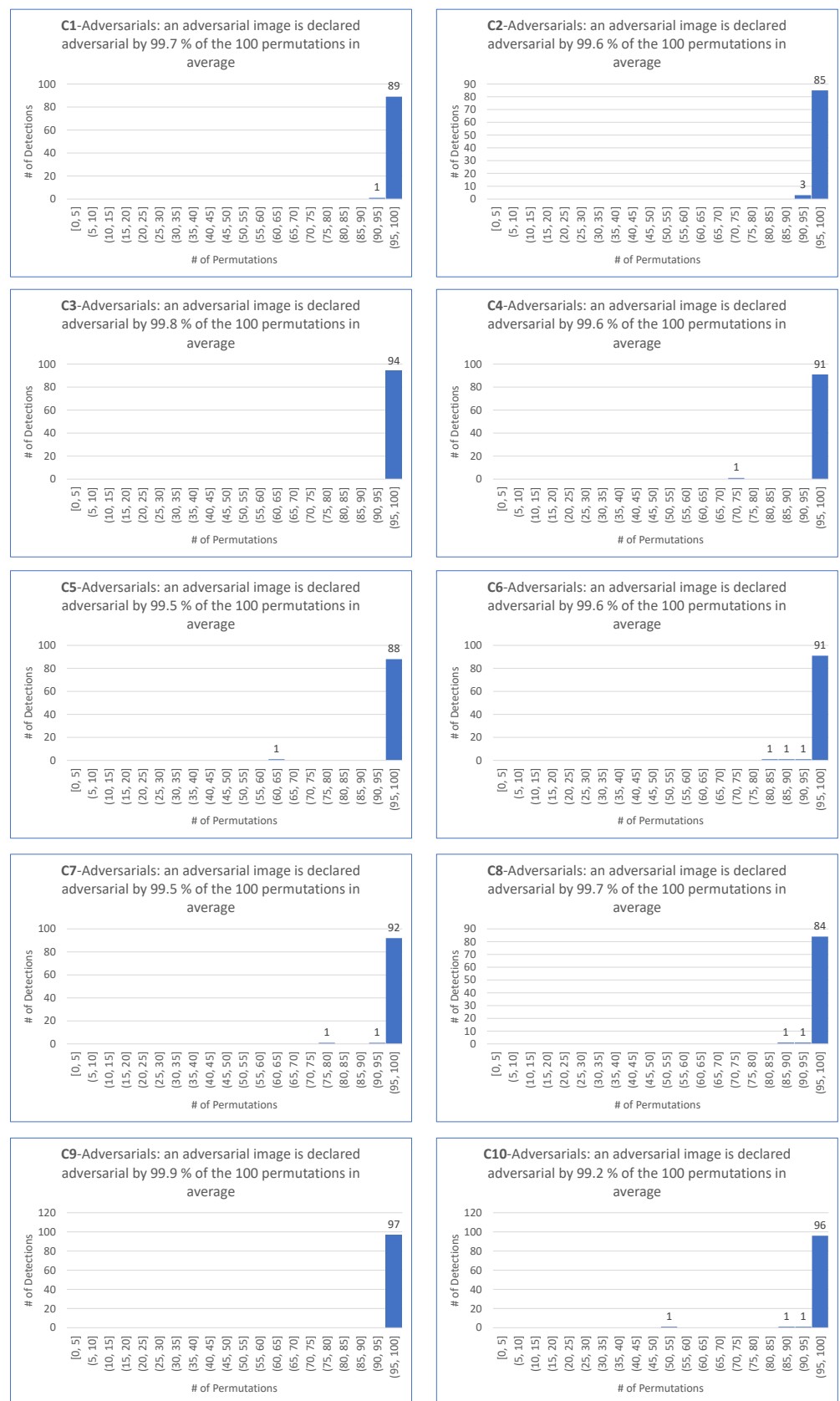

**Figure A6.** ShuffleDetect results for adversarial images generated by the **PGD L2-targeted** attack on $\mathcal{C} = \mathcal{C}$ for $1 \leq k \leq 10$ over 100 permutations.

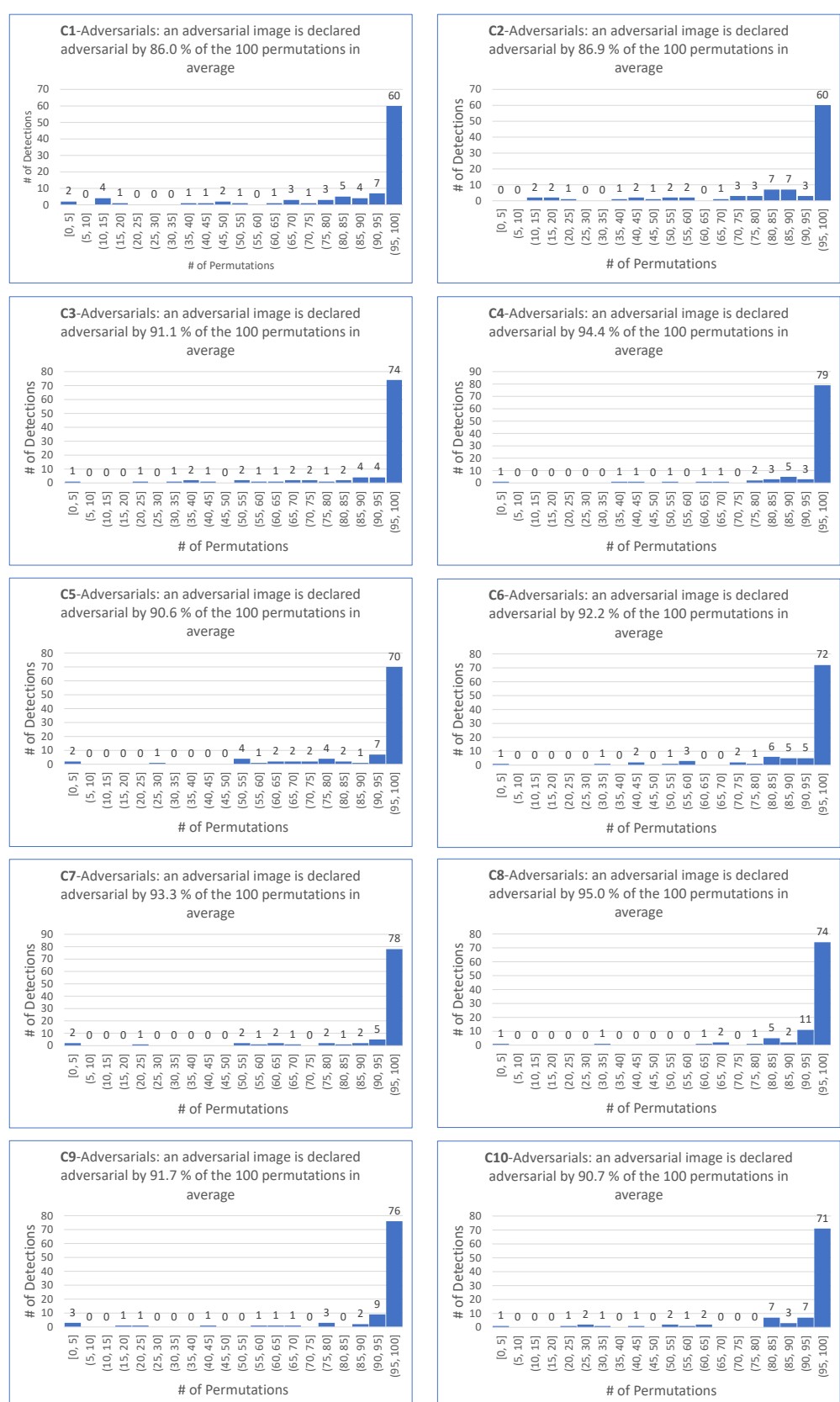

**Figure A7.** ShuffleDetect results for adversarial images generated by the **EA-untargeted** attack on $\mathcal{C} = \mathcal{C}$ for $1 \leq k \leq 10$ over 100 permutations.

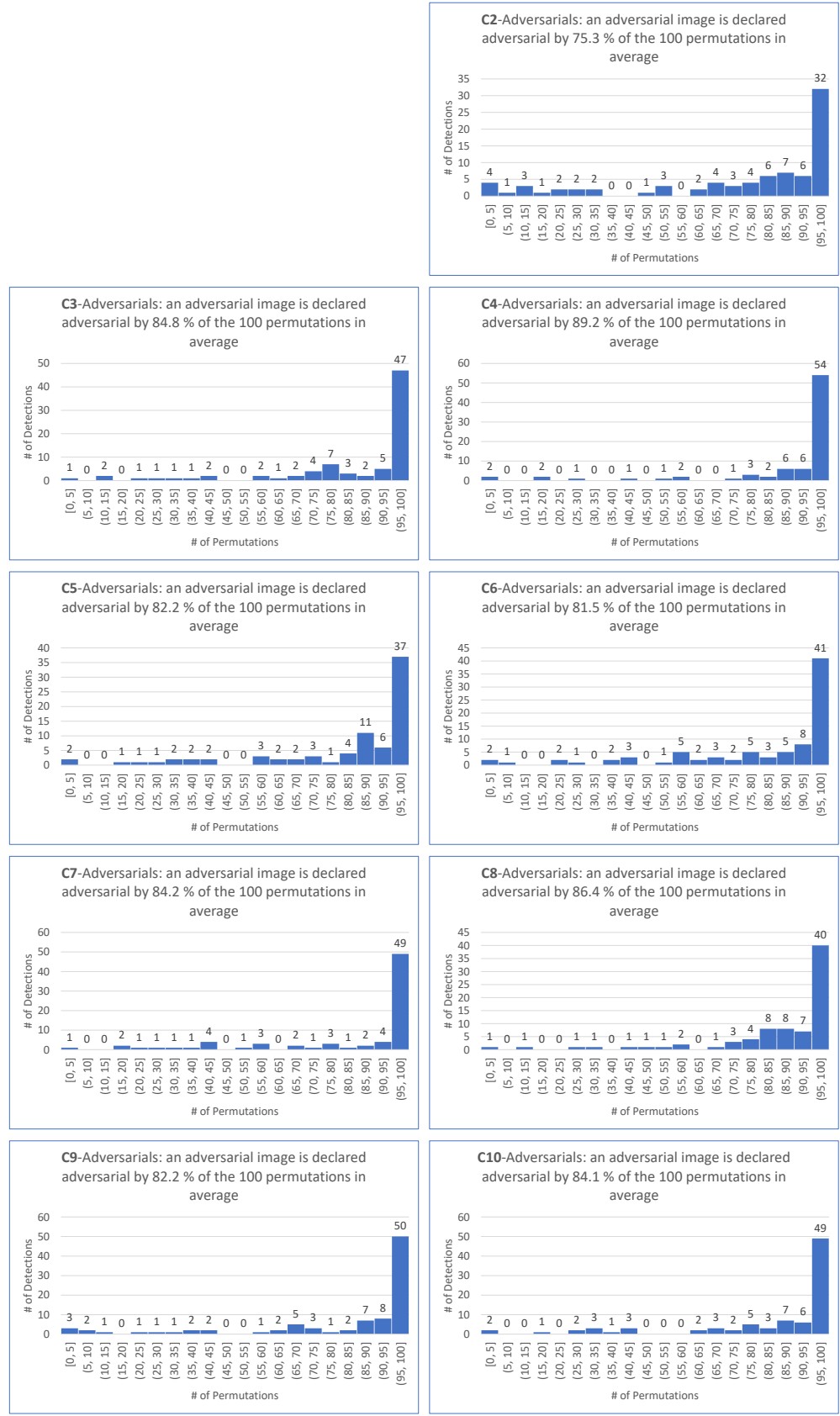

**Figure A8.** ShuffleDetect results for adversarial images generated by the **FGSM-untargeted** attack on $\mathcal{C} = \mathcal{C}$ for $2 \leq k \leq 10$ over 100 permutations.

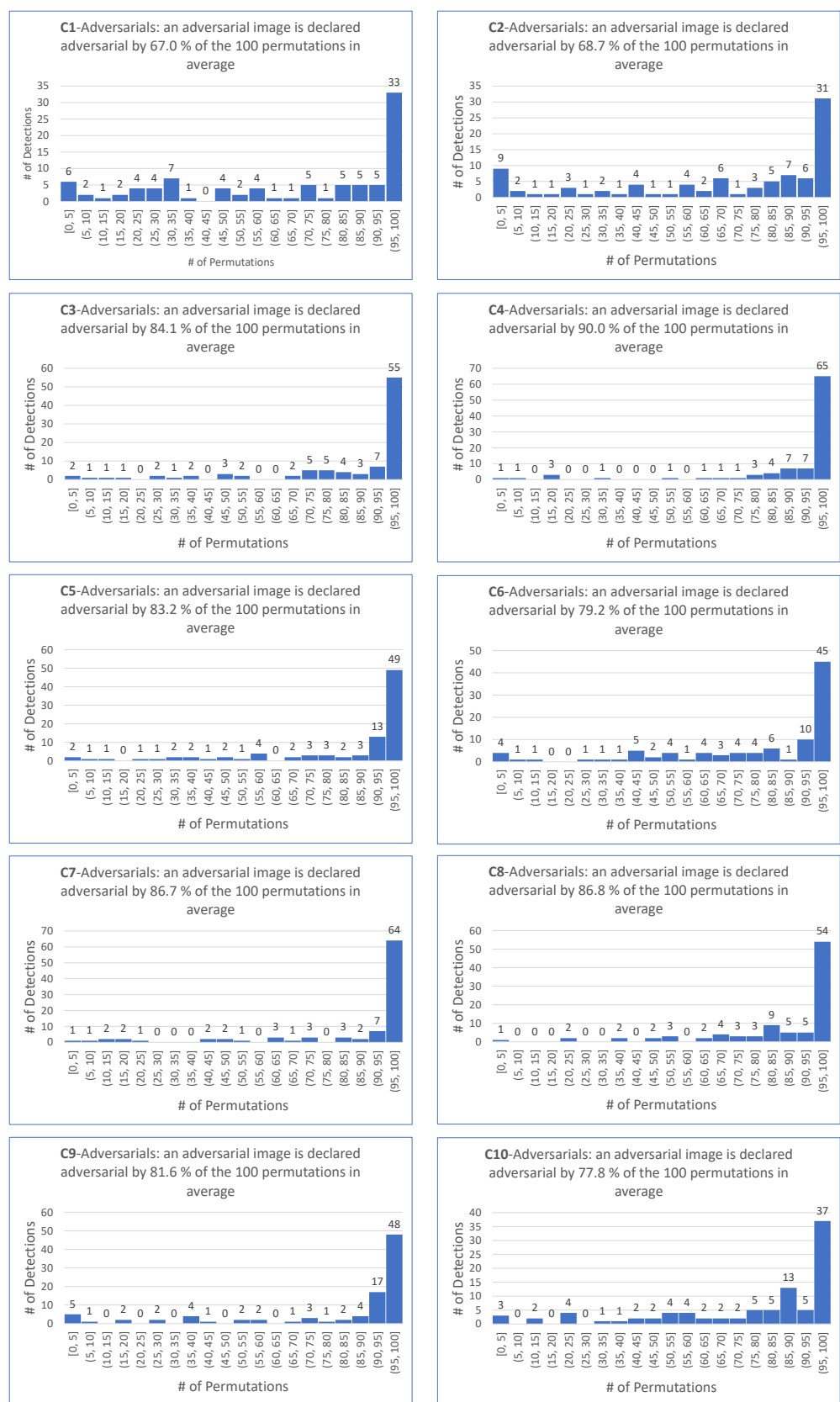

**Figure A9.** ShuffleDetect results for adversarial images generated by the **BIM-untargeted** attack on $\mathcal{C} = \mathcal{C}$ for $1 \leq k \leq 10$ over 100 permutations.

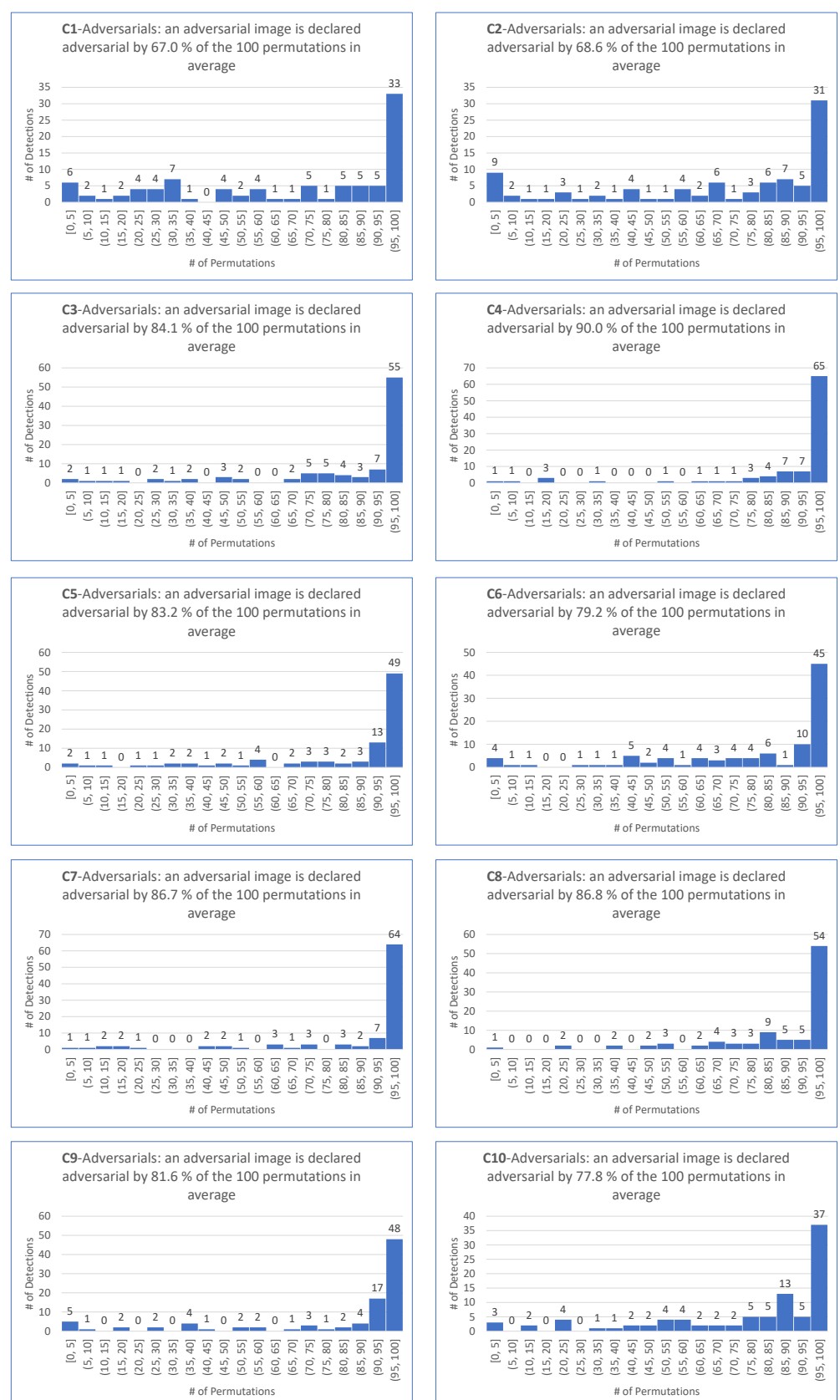

**Figure A10.** ShuffleDetect results for adversarial images generated by the **PGD Inf-untargeted** attack on $\mathcal{C} = \mathcal{C}$ for $1 \leq k \leq 10$ over 100 permutations.

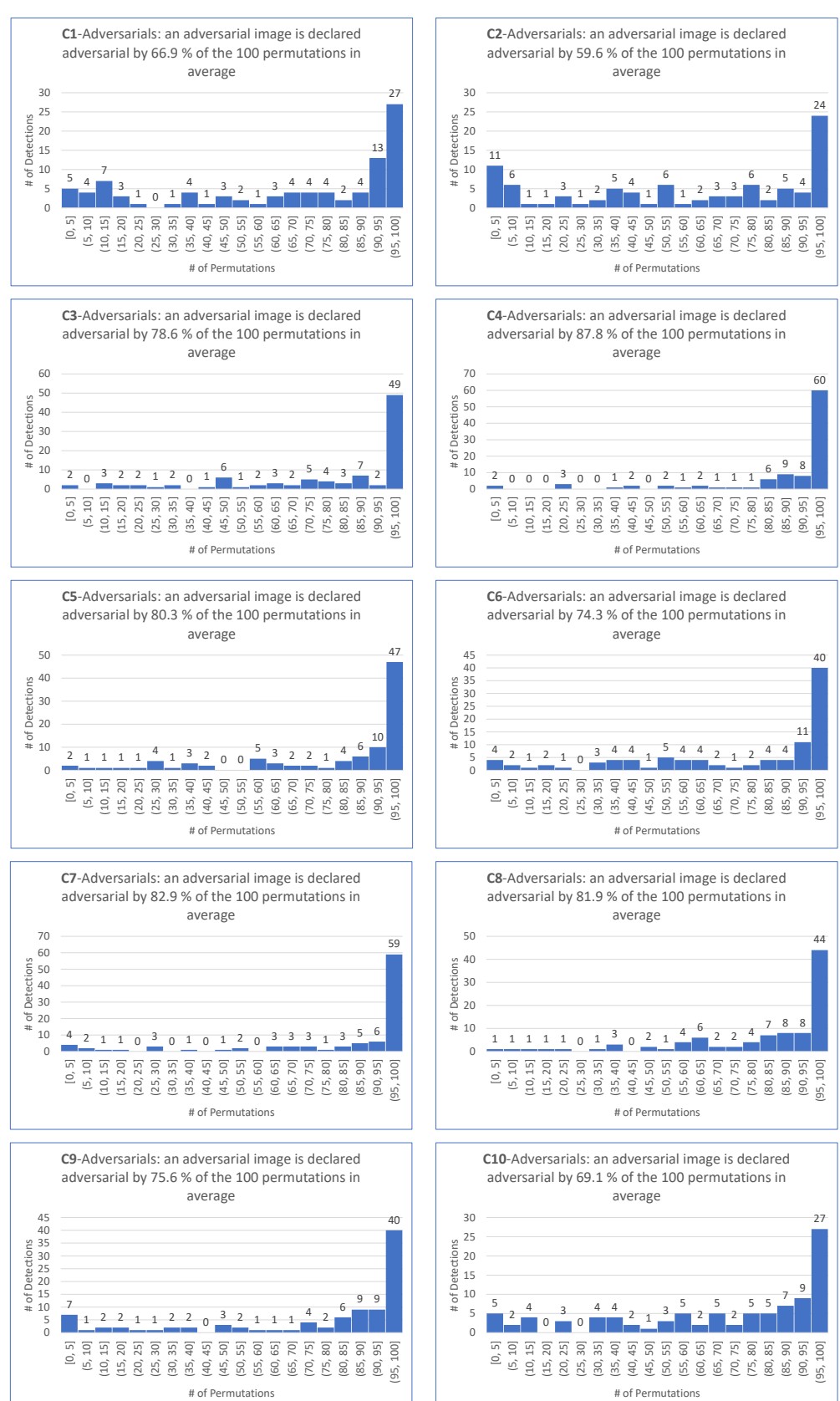

**Figure A11.** ShuffleDetect results for adversarial images generated by the **PGD L2-untargeted** attack on $\mathcal{C} = \mathcal{C}$ for $1 \leq k \leq 10$ over 100 permutations.

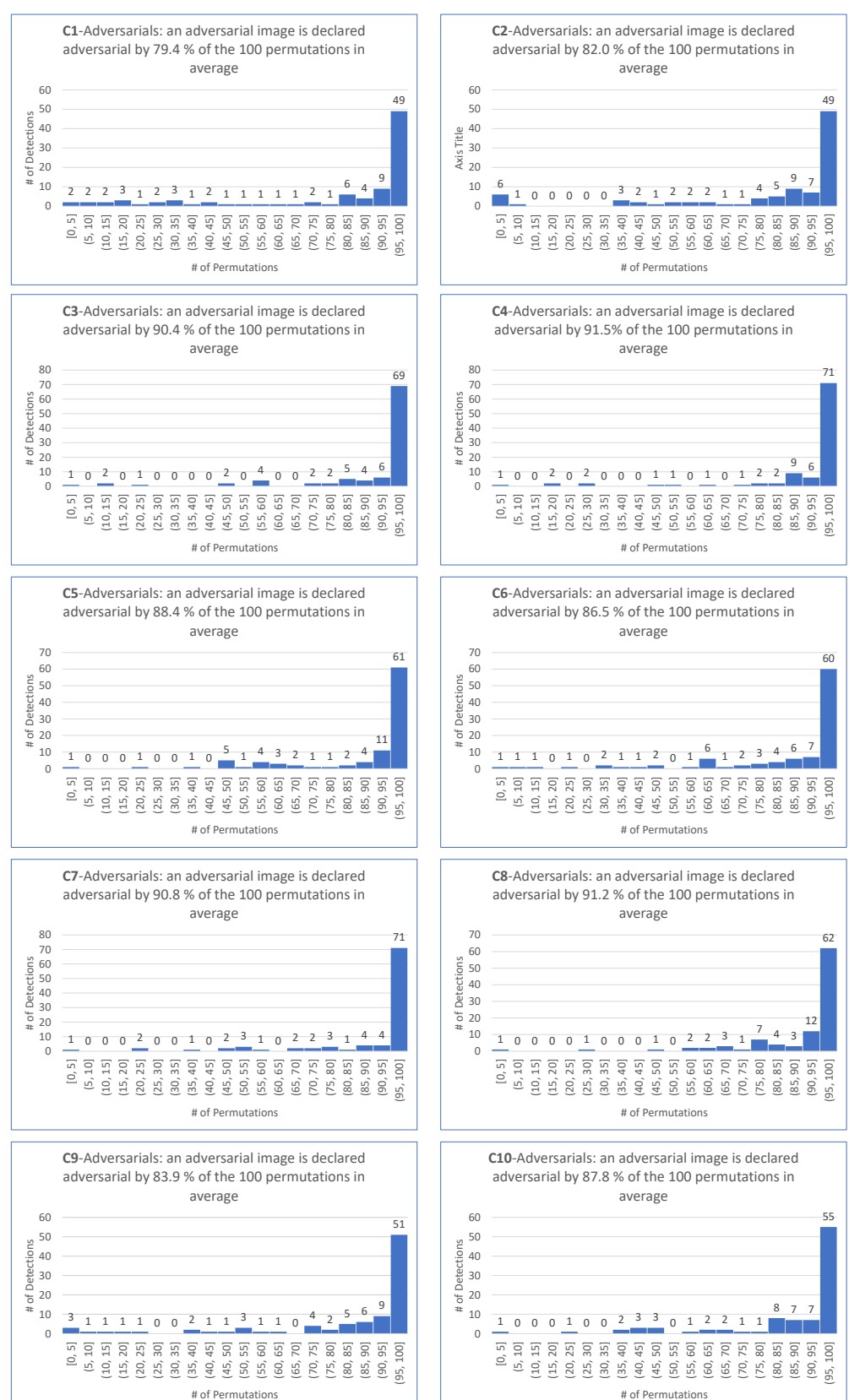

**Figure A12.** ShuffleDetect results for adversarial images generated by the **CW Inf-untargeted** attack on $\mathcal{C} = \mathcal{C}$ for $1 \leq k \leq 10$ over 100 permutations.

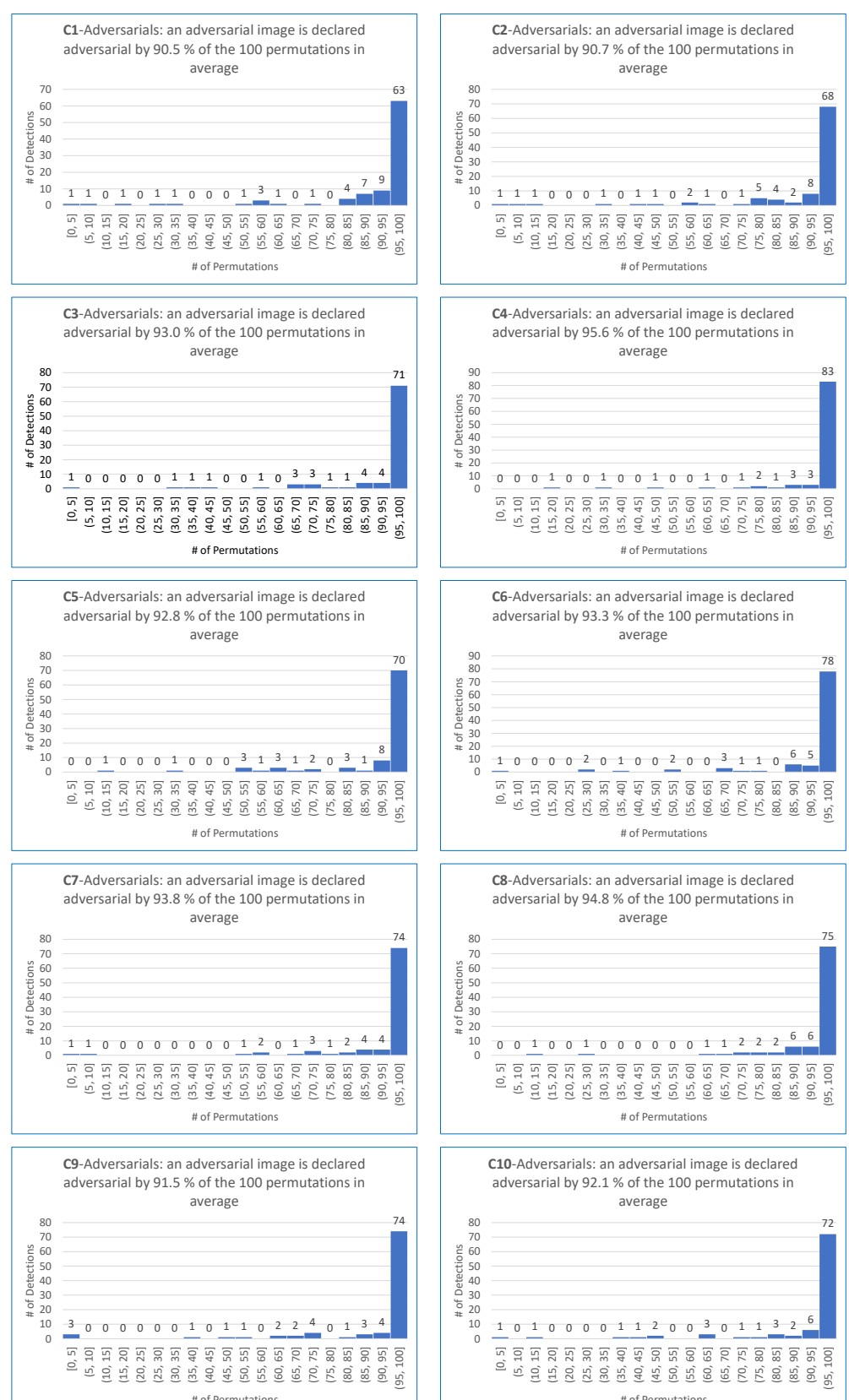

**Figure A13.** ShuffleDetect results for adversarial images generated by the **Deep Fool-untargeted** attack on $\mathcal{C} = \mathcal{C}$ for $1 \le k \le 10$ over 100 permutations.

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
