# Peer review of "ShuffleDetect: Detecting Adversarial Images against Convolutional Neural Networks"

_applsci, doi:10.3390/app13064068_

Round 1
Reviewer 1 Report
1. The paper's title is "Detecting Adversarial Images against Convolutional Neural Networks". Detection is an important aspect that should be measured by accuracy. In the entire manuscript, accuracy has not been measured. This dilutes the verification of the proposed work.
2. The comparison with existing related work must be tabulated based on accuracy.
3. Introduction section is extremely poor. sufficient literature must be included.
4. In abstract, the authors have not included the problem statement.
5. authors have taken the Evaluation criteria from [16] and not compared which not justifying.
6. As claimed in conclusion section, proposed ShuffleDetect achieves an average FPR of 32.7%, an average DR of 100% for the adversarial 684 images obtained by targeted attacks. It is doubtful results, clarification is required.
7. Revision of the language of the manuscript is required.
8. "the performance of 8 ShuffleDetect is intrinsically". which parameter is responsible for intrinsic performace evaluation and why? also include extrinsic parameters like mean opinion score
9. Experiments show that ShuffleDetect is an easy-to-implement, very fast, 9 and close to the memory-free detector, that achieves high detection rates and low false positive rates.
a) How "easy-to-implement" ? Include into the manuscript
b) very fast? compare with literature
c)high detection rates? compare with literature
d) low false positive rates? compare with literature
Also make sure to compare the results on same dataset as in literature
10. Rth = 0.51 is selected for most applications. which is dataset baised. Test the same for other dataset as well
Reviewer 2 Report
This paper proposes a novel unsupervised approach for detecting adversarial image attacks against Convolutional Neural Networks (CNNs) called "ShuffleDetect". The method involves dividing an input image into non-overlapping patches and shuffling them according to permutations. The proportion of permutations for which the CNN classifies the unshuffled input image and the shuffled image into different categories is counted. If this proportion exceeds a threshold value, the image is declared adversarial.
The paper shows that the method is efficient, taking fewer than 100 "shuffling" permutations to get good results, with fairly low memory and computational overheads and moreover easy-to-implement with high detection rates and low false positive rates.
The paper is generally well-written but it ought to be proof-read by a native speaker to correct a few syntactic errors which occasionally hamper intelligibility. Even the abstract is not quite error-free "close to the memory-free detector" should be "close to a memory-free detector" or "close to being a memory-free detector". Some sentences are unclear and seem to have words or symbols missing. For example, "...by first extracting the dominating category Dom(I) in which sorts I" and "... requires from the dominating category Dom(sh(I, σ)) in which sorts the shuffled image" (both on page 2), which are neither grammatical nor meaningful as written (in which what sorts what?).
The key assumptions are that "shuffling" affects adversarial noise more than an image's intrinsic content, and moreover that the permutation of image patches should tend not to cause a CNN to change the "dominating" classification of that image unless the original image was an adversarial example. These assumptions are validated using a set of 8 untargeted and 7 targeted attacks on 10 ImageNet-trained CNNs using 10 randomly chosen "ancestor" classes (with 10 images each) and target classes taken from ImageNet. The experiments consisted of 15000 "attack runs" (15 attack variations applied to 10 CNNs, each with 10 classes and 10 images per class) that yielded approximately 9700 successful adversarial images, some of which were ruled out to due low success rates of the respective methods. The authors find that ShuffleDetect achieves high detection rates and low false positive rates on the remaining set of about 9500 adversarial images. The method is limited to CNNs with square input sizes, although this is not a major constraint.
However, the algorithm requires careful consideration of hyperparameters, namely the patch size, the number of permutations, and the threshold ratio. The authors acknowledge that "Therefore, for the time being, our choice of parameters is purely experimental", so there is some uncertainty regarding the degree to which these parameters would have to be adapted for CNN architectures, classification tasks, and adversarial attacks that differ significantly from those considered in this paper.
Moreover, the compositional complexity and relative scale of objects within an image, together with the sort of classification task that a given CNN has been trained on (e.g. object classification vs scene classification) will likely result in significant differing impact of the classification ratio depending upon patch size.
ShuffleDetect's sensitivity to parameter choice needs to be investigated further. Some such work was done by the authors elsewhere and is cited ("Empirical Perturbation Analysis of Two Adversarial Attacks: Black Box versus White Box") but the selected value s=56 is very unlikely to be universally optimal (the parameter value Rth = 0.51 seems more convincing). It would be interesting to see how much hyperparameter tuning would be required to make ShuffleDetect work well on significantly different classification tasks. However, the paper presents extensive performance analysis of thresholds and permutation counts for the chosen CNNs and data sets.
Unfortunately, the authors do not show clear comparative results with other adversarial detection techniques. Instead, they say that "Ideally, we would like to compare ShuffleDetect with well-known detectors (...) We do not undertake this task here and keep it for future work". However, many of these alternative methods are not that complex, so implementing and testing at least one of them would have been expected. The lack of this comparative evaluation is a major weakness of the current paper.
So while the method is interesting has the potential to be widely used in practical applications for detecting adversarial images against CNNs, the paper (or a follow-up study) should evaluate the choice and sensitivity of its parameters in more detail and explicitly compare it against other related techniques. Overall, the authors make a convincing case that ShuffleDetect appears to be an efficient method that could be used on its own or in conjunction with other approaches.
